

# Asymmetric, biraphid diatoms from the Laurentian Great Lakes

Euan D. Reavie

Natural Resources Research Institute, University of Minnesota Duluth, Duluth, MN,
United States

## ABSTRACT

This taxonomic account of light micrographs from the coastal Laurentian Great lakes contains taxa from the diatom genera *Amphora, Halamphora, Cymbella, Cymbopleura, Delicatophycus, Encyonema, Encyonopsis, Reimeria, Gomphonema, Gomphosphenia, Gomphonella, Gomphosinica*, and *Gomphoneis*. A total of 207 samples of surface sediment and periphyton collected from 106 wetland, high-energy, embayment, and deeper nearshore locales are represented. Light micrographs of 154 taxa are presented. Of these, 76 could not be fully identified as known taxa from the existing literature and so are given tentative names, numbers or conferred assignments. Lake and habitat specificity, modeled autecological optima for phosphorus and chloride, and tolerance to anthropogenic stressors are described for 39 of the more common taxa.

## INTRODUCTION

Despite being the largest surface freshwater system in the world, The Laurentian Great Lakes diatom flora is poorly known. Great Lakes diatom studies before the mid-20th century were rare (*Thomas & Chase, 1886*; *Vorce, 1881*; *Skvortzow, 1937*), though diatom accounts increased in the late 1960s with Eugene Stoermer and his colleagues' ecological assessments supporting water quality conservation. The earliest work focused on Lake Michigan (*Stoermer & Yang, 1969*; *Stoermer & Ladewski, 1976*) and eventually led to more detailed reports from extensive field efforts (*Stoermer & Yang, 1971*; *Stoermer, 1978*; *Stevenson & Stoermer, 1978*; *Kreis & Stoermer, 1979*; *Kociolek & Stoermer, 1990, 1991*; *Theriot & Stoermer, 1984*). Eventually this work led to retrospective interpretations through paleoecological assessments of fossil diatom records (*Stoermer, Wolin & Schelske, 1993*). Stoermer's use of diatoms as powerful inferential tools in the Great Lakes paved the way for their use in large environmental programs, including the Great Lakes Environmental Indicators (GLEI) project (*Niemi et al., 2006*), and the multi-decadal Great Lakes Biological Monitoring Program led by the USEPA's Great Lakes National Program Office (*Barbiero et al., 2018*). In 2000, GLEI was initiated to focused on development of indicators of coastal conditions in the lakes. Diatom collections from coastal habitats were used to create diatom-based indicators of ecological stress (*Danz et al., 2005*; *Reavie et al., 2006*). Diatom evaluations using light microscopy (LM) were based on ornamentation of

Corresponding author
Euan D. Reavie, ereavie@d.umn.edu

the siliceous cell wall. To date, applied works from these taxonomic assemblages have included inference models for phosphorus and coastal stressors, multimetric indices, and habitat assessments (*Reavie et al., 2006*; *Sgro et al., 2007*; *Reavie, 2007*; *Brazner et al., 2007*; *Reavie et al., 2008*; *Kireta et al., 2007*). So far, iterative publications of Laurentian Great Lakes diatoms have covered 'centric' and 'araphid' taxa (*Reavie & Kireta, 2015*), monoraphid taxa (*Reavie, 2020*), *Navicula* Bory (*Reavie & Andresen, 2020*) and the non-*Navicula* symmetric taxa (*Reavie, 2022*). In this article I aim to illustrate the asymmetric, biraphid taxa observed during the GLEI project, summarizing their autecology when possible.

## MATERIALS AND METHODS

Detailed collection, preparation and analytical methods are closely paraphrased from previous publications (*Reavie, 2020*, *2022*; *Reavie & Kireta, 2015*; *Reavie & Andresen, 2020*), and they are included here for clarity.

Field sites (Fig. 1) were sampled from June to September 2002 and May to August 2003 (*Reavie & Kireta, 2015*). In addition to diatom samples, a suite of environmental measurements was collected at each sample location, and a detailed account of these parameters is provided by *Reavie et al. (2006)*. Benthic and sedimented diatoms were sampled from natural substrates from 0.5 to 3 m depth. Additional surface sediment samples were collected from nearshore locations at a 30-m depth from the USEPA's research vessel Lake Explorer. Surface sediments were sampled using a 6.5 cm diameter push corer and core tube. Sediments were extruded in the boat or on shore, and the top 1 cm of sediment was carefully removed using a spoon and/or spatula. In areas where coring was not feasible, a 'petite' PONAR sampler was used to collect unconsolidated bottom substrates, or rocks were carefully collected by hand. Approximately 1 cm of surface sediments from PONAR samples was removed using a spoon and/or spatula. The surfaces of rocks and pebbles were scrubbed clean with a small brush or plastic knife and collected in vials as epilithic samples. All samples were iced at 4–6 °C until processing. Approximately 75% of sites were cored, 13% required PONAR grab samples and 12% relied on epilithic samples collected by hand. The full set of sample locations, sample types and associated environmental data are provided in Table S1.

### Sample preparation and analysis (Reavie & Kireta, 2015)

In the lab, subsamples were taken from homogenized sediment samples, and the diatom remains were cleaned using concentrated nitric or hydrochloric acid, or 30% hydrogen peroxide. Samples were digested in a water bath (85 °C) for 1 h. Samples were allowed to cool and settle at room temperature for 24 h and then were centrifuged at 1,800 RPM for 10 min. The tubes were aspirated, refilled with deionized water and shaken to break up the pellet. This centrifugation process was repeated five times. Four microscope slides were prepared for each sample using the *Battarbee (1986)* method. Diatom assessments for the GLEI project relied on light microscopy for timely data collection. For each sample, 400 diatom valves were counted along random transects at 1,000× magnification using oil immersion microscopy. Counts were made continuously along transects as wide as the

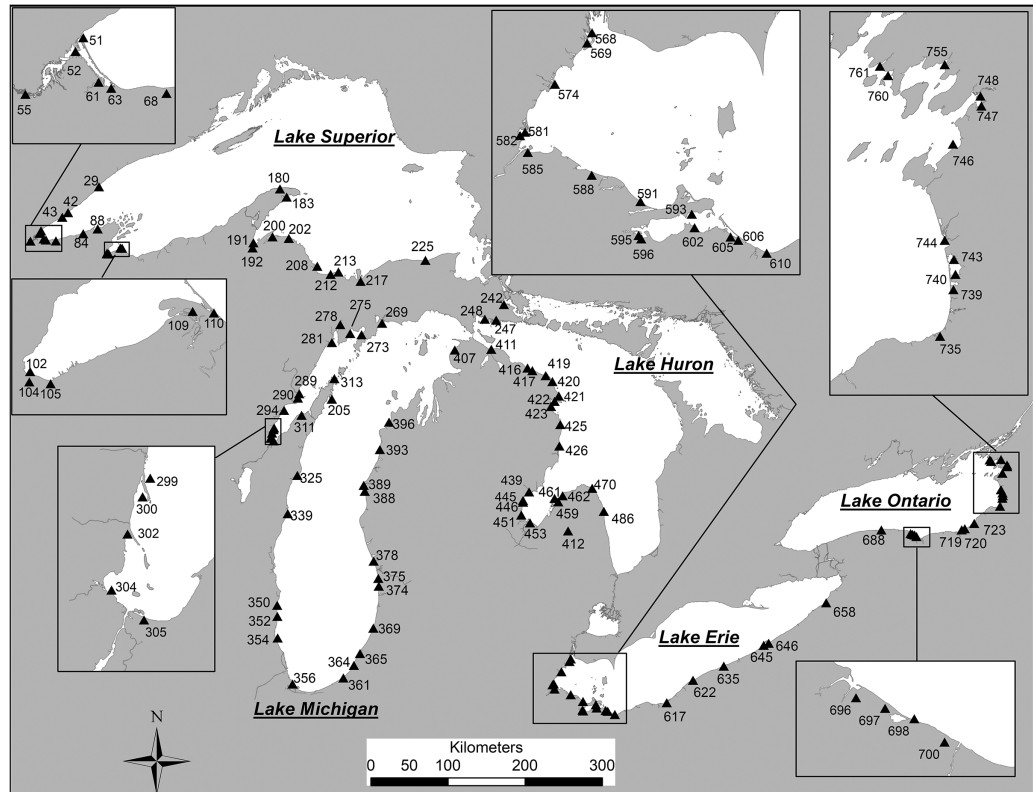

**Figure 1 Coastal sample locations for diatom samples from the Laurentian Great Lakes.** Station numbers match those for each specimen photograph in the plate captions (in square parentheses). Modified from *Reavie & Kireta (2015)*.

field of view until sufficient valves were counted. Specimen photography did not follow a regimented protocol; digital photographs of diatom valves generally occurred as new taxa, or variations thereof, were encountered. Species were identified to the lowest taxonomic level possible using numerous diatom checklists, journal articles and iconographs. The most recently published nomenclature was used as much as possible for identifications (*e.g.*, *Guiry & Guiry, 2022*; *Spaulding et al., 2022*).

Photographic sessions involving scanning and photographing specific genera also occurred in preparation for the many taxonomic workshops held throughout the project period (*Reavie & Kireta, 2015*; *Reavie & Andresen, 2020*). Photographs were always collected at the highest magnification possible (1,000× and higher) using standard brightfield, or employing differential interference contrast or oblique light path interference. Photography occurred at three primary locations: The University of Minnesota Duluth, John Carroll University, and the University of Michigan. Because of the variation in locations and equipment, a range of gray tones are observed across images. Although this article does not provide high-resolution diagnostic images that would be acquired through scanning electron microscopy, it is hoped that it will be used by taxonomists interested in species occurrences and distributions in the Great Lakes.

Preserved material and prepared microscope slides are stored in the Natural Resources Research Institute's diatom collection at the University of Minnesota Duluth.

## RESULTS, DISCUSSION AND TAXONOMY

As portions of the diatom flora for the coastal Laurentian Great Lakes are iteratively published, it has become clear that at least one-third of the species have never been adequately described. Even dominant diatom species in the Great Lakes phytoplankton have been described only recently after decades of incorrect identifications (*Alexson et al., 2018*, *2022*; *Van de Vijver et al., 2021*); a similar lack of understanding is observed in the highly diverse benthic communities from the coastal Great Lakes. For this manuscript suitable photos were acquired of 154 of the ~2,200 diatom taxa encountered in GLEI samples. Unlike most existing diatom monographs, imperfect specimens such as broken valves or those with plane-of-focus issues are sometimes presented to account for occurrence of a taxon and at least some morphological information. Taxonomic accounts herein vary based on their abundance and need for documentation. Species that sufficiently meet the parameters of previously published accounts are afforded fewer details than uncertain or unknown taxa, which require greater details on how they differ from known taxa. Diagnostic information for most taxa includes the observed ranges of valve length, width and stria density. We acknowledge that the measured specimens represent those that were observed during the environmental assessment, and taxon-focused assessments (such as collection and analysis of additional material where a taxon was present) were not performed. Hence, the diagnostic information provided for less common taxa likely does not capture the full ranges present in the Great Lakes, and instead these values were used to provide evidence for linking identifications to previously published accounts. The 'cf.' qualifier (="confer", meaning "compare with") is used frequently because many taxa did not adequately fit published accounts. I do not establish new taxa because of uncertainty about whether a given difference reflects variability in an existing species, and so additional specimen observations are recommended. Despite decades of diatom work in the Laurentian Great Lakes, the lack of viable assignment to known species is a result of past reliance on (mostly) Eurasian concepts, such as the many published accounts from European authors. Now, we are fortunate to have an accelerating publication of North American diatoms, such as through the open-access Diatoms of North America portal (*Spaulding et al., 2022*; cited multiple times herein). Despite the many uncertainties in my species descriptions, this specific account of the asymmetric diatoms should provide a floral basis for future diatom-based assessments in the Great Lakes. Many of the diatom taxa described herein are not found in existing literature.

A total of 39 taxa were sufficiently abundant to generate environmental characteristics (Figs. 2–5). These environmental data are called out in the following taxonomic accounts. Slides and freeze-dried materials are retained at the Natural Resources Research Institute (NRRI).
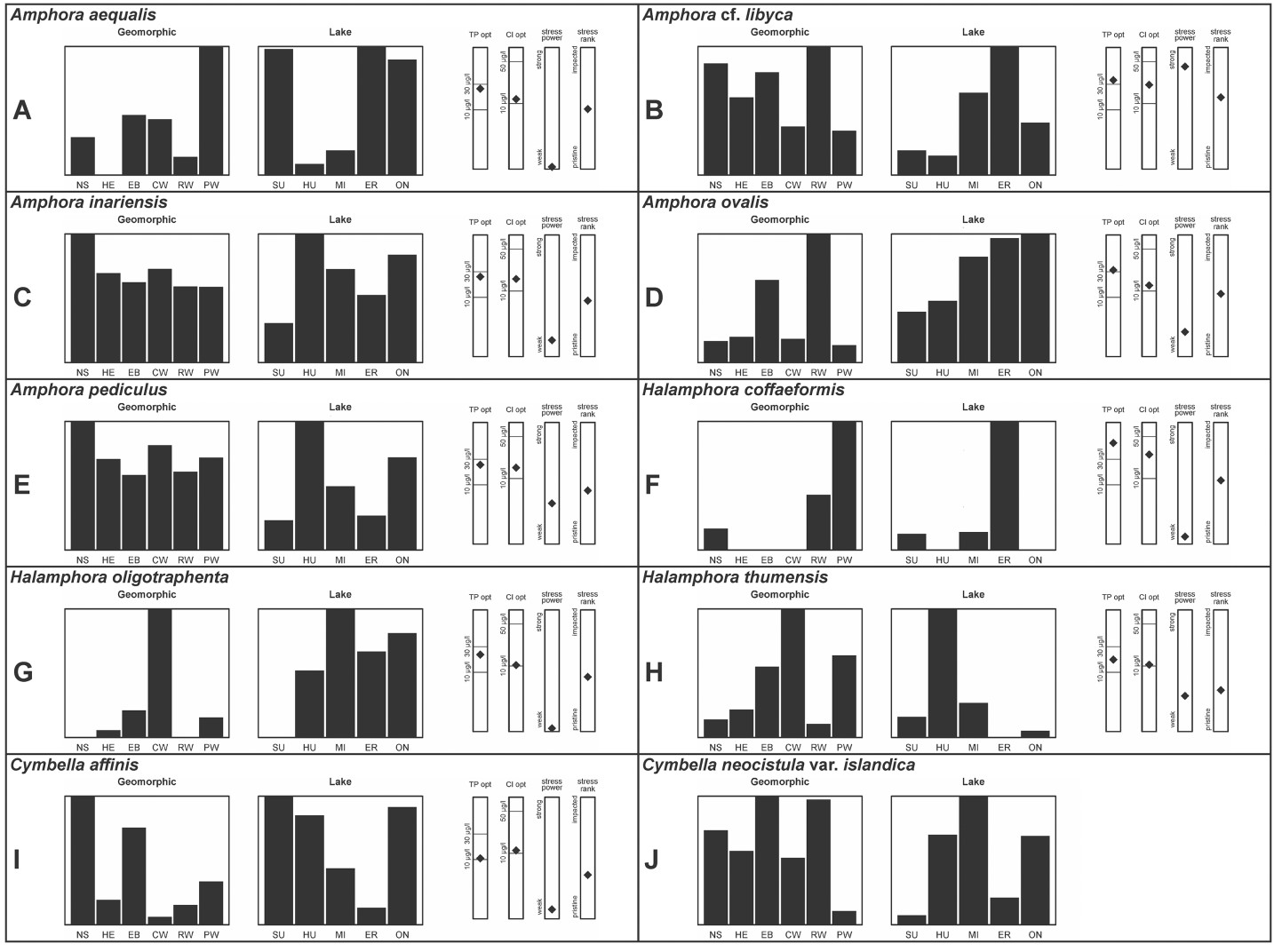

**Figure 2** (A–J) Part 1: geomorphic habitat distribution, lake specificity and environmental characteristics for several of the more common taxa in the United States Great Lakes coastlines.

Systematic account

As for the methods, the following is again closely paraphrased from *Reavie (2022)*. Photographs are linked to taxonomic and autecological descriptions. In cases where specimens could not be reconciled with existing literature, a few approaches were followed (*Reavie, 2022*). For species resembling known taxa but differing slightly (*e.g.*, slightly lower or higher stria density, slightly smaller or larger valve), a 'cf.' qualifier is applied. Sometimes a question mark (?) is associated with ambiguous photographs with uncertain affinity but enough similarity to the described taxon for tentative consideration. Sufficiently unique taxa with poor correspondence to existing published records were assigned 'sp.' or a numeric identifier given at the time of GLEI sample assessment. In many uncertain cases I include recommendations for future scanning electron microscopy (SEM) assessment. Ranges of morphological parameters in Great Lakes specimens are

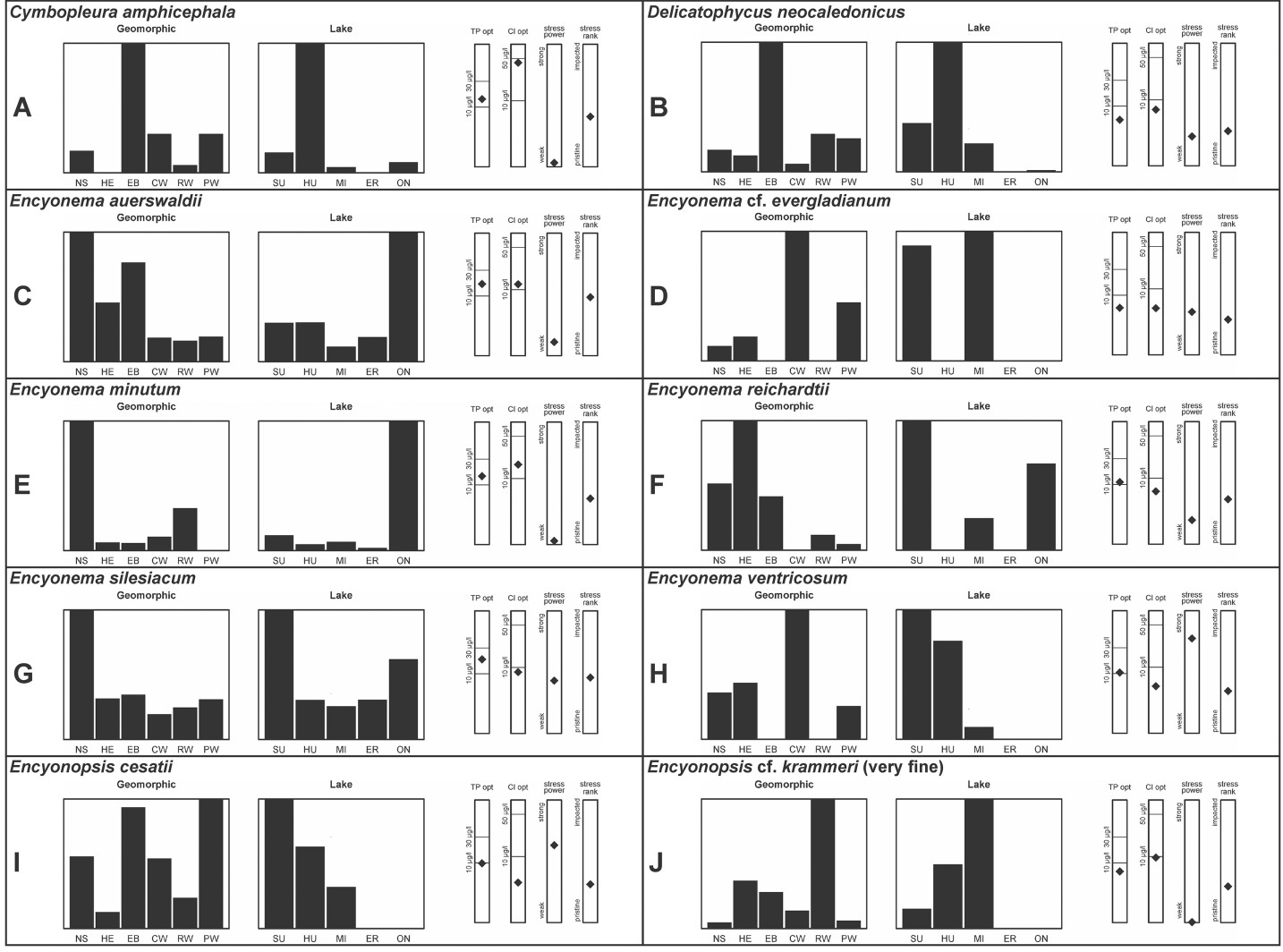

**Figure 3** (A–J) Part 2: geomorphic habitat distribution, lake specificity and environmental characteristics for several of the more common taxa in the United States Great Lakes coastlines.

provided relative to published accounts; ranges of those parameters from previously published account are provided and cited.

Environmental information for taxa: Environmental characteristics were quantified for common taxa (*Reavie, 2020*, *2022*; *Reavie & Kireta, 2015*; *Reavie & Andresen, 2020*). Taxa were considered common if: they occurred in at least five samples with greater than 1% relative abundance in at least one of those samples; or they represented more than 5% relative abundance in at least one sample. Autecology for each of these common taxa is presented in a summary diagram (Figs. 2–5). A histogram presents habitat affinity according to the relative frequency the taxon was encountered in each geomorphic habitat (high-energy (HE), embayment (EM), riverine wetland (RW), protected wetland (PW), coastal wetland (CW), open water nearshore (NS); *Kireta et al., 2007*), based on samples in which the species was observed. This diagram is intended to depict whether taxa have habitat specificity in our samples or if one should expect to encounter a taxon across a wide

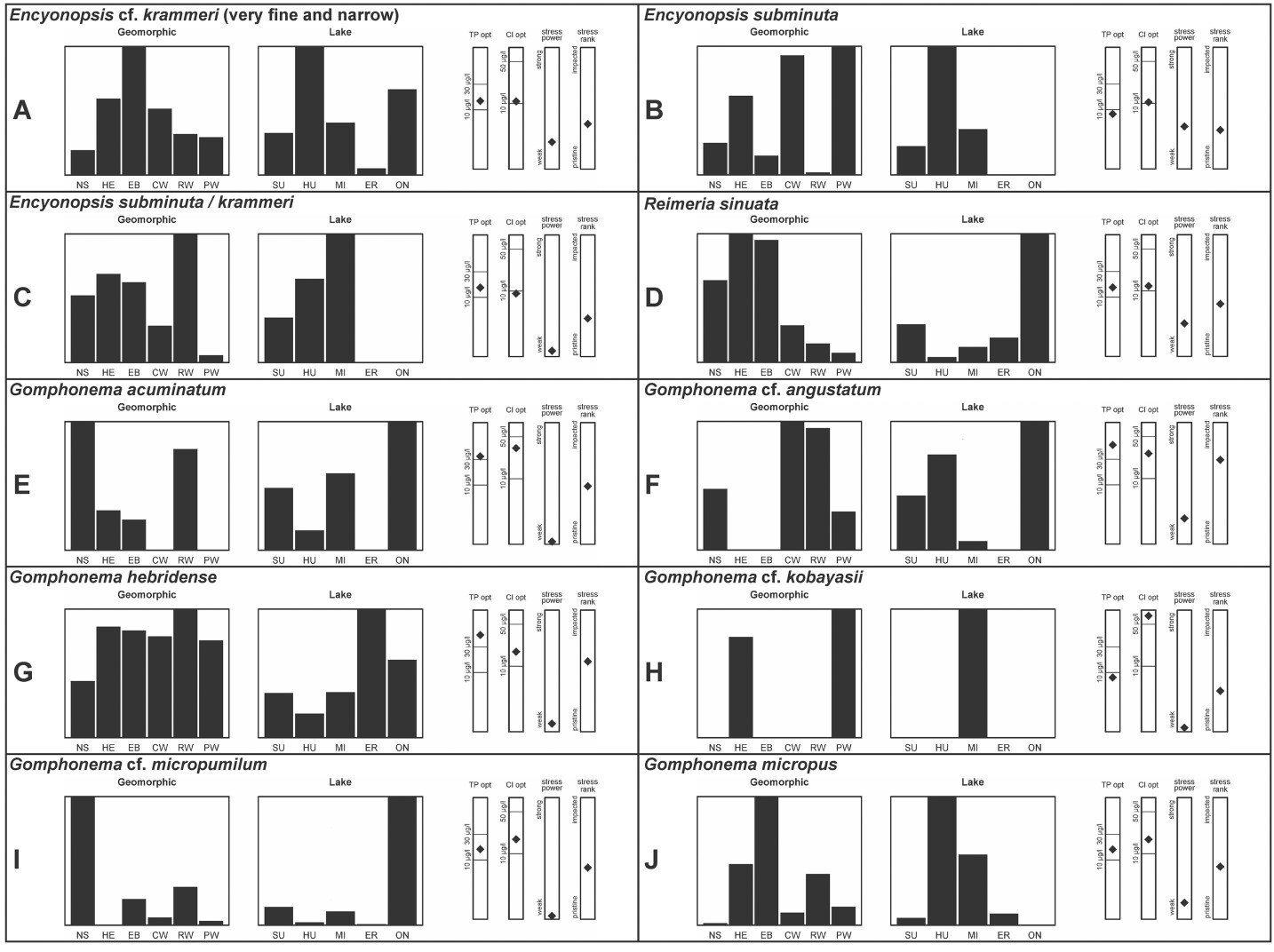

**Figure 4 (A–J) Part 3:** Geomorphic habitat distribution, lake specificity and environmental characteristics for several of the more common taxa in the United States Great Lakes coastlines.

range of physical conditions. A second histogram illustrates the relative occurrence of each taxon in the five lakes (Superior (SU), Michigan (MI), Huron (HU), Erie (ER), Ontario (ON); *Kireta et al., 2007*) incorporating standardized weighting by the number of samples collected in each lake. Again, this relative representation of occurrence is based on samples in which the given species was observed.

When a taxon was observed in at least 10 locations, water quality optima for total phosphorus (TP) and chloride (Cl) were presented along vertical bars representing the measured water quality gradient for all GLEI samples. These autecological data for common taxa were based on statistical evaluations of species-environmental relationships (covered in greater detail by *Reavie et al. (2006)* and *Kireta et al. (2007)*). Species optima for these variables were estimated using weighted averaging regression and calibration, implemented by the rioja package (*Juggins, 2015*) using the R statistical program. Diatom assemblages were related to water chemistry assuming unimodal species responses along

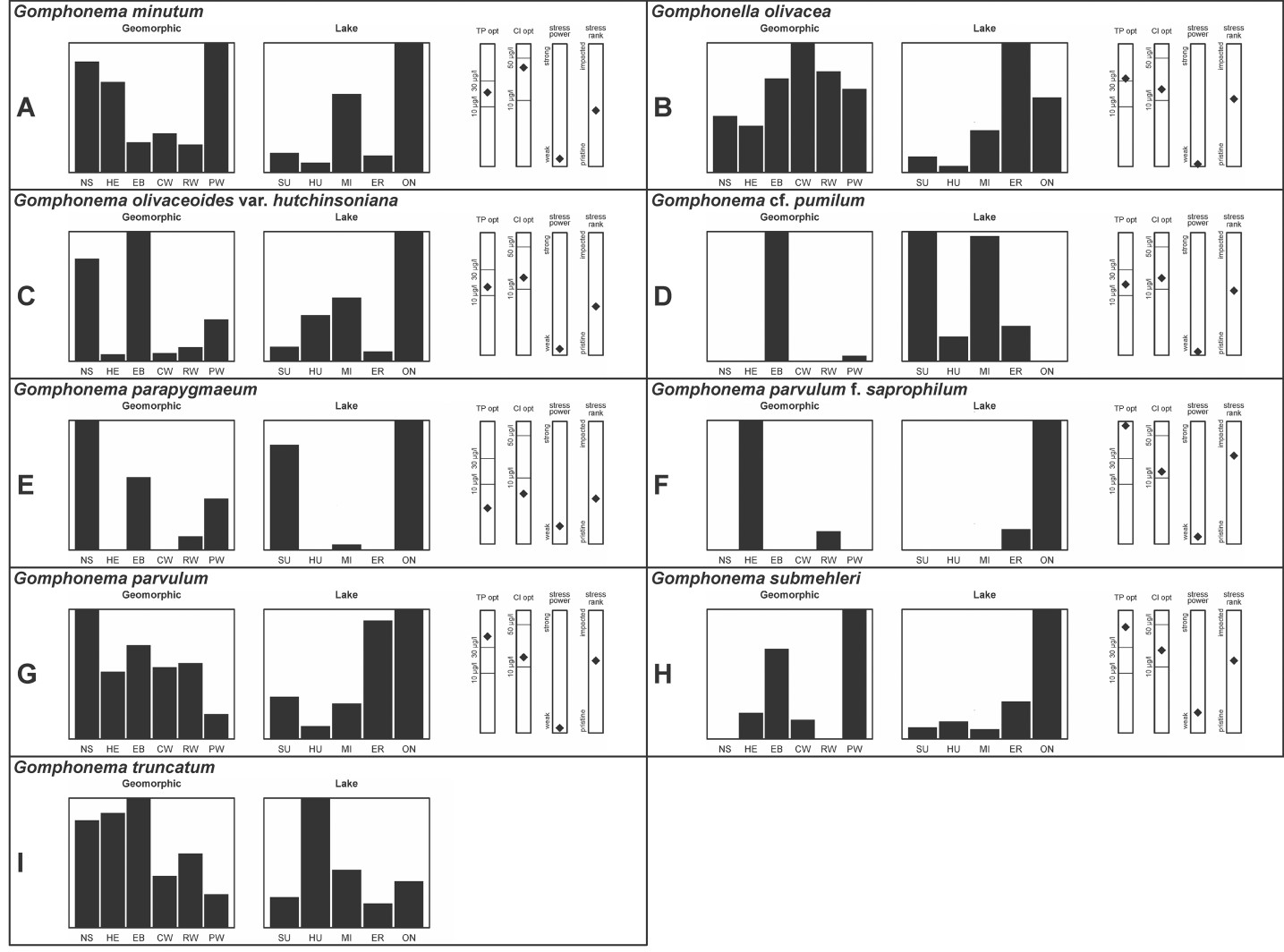

**Figure 5** (A–I) Part 4: geomorphic habitat distribution, lake specificity and environmental characteristics for several of the more common taxa in the United States Great Lakes coastlines.

log-transformed environmental data: 1–521 μg TP/L (average 20 μg/L); 0.33–120.74 mg Cl/L (average 13.39 μg/L). Two additional bars, 'stress power' and 'stress rank,' depict the relative ability of a taxon to track stress and whether that taxon reflects low or high stress. These results were achieved by evaluating relationships between each individual taxon and agricultural, industrial and urban development stressors quantified for each sample location. Briefly, the U.S. coastline of the Great Lakes was divided into 762 segments, each consisting of a shoreline reach and associated watershed (*i.e.*, a 'segment-shed'). Each segment-shed was summarized using 207 geographic information system (GIS)-based environmental variables that included anthropogenic activities (*e.g.*, agricultural activities, urban density, industrial polluters) (*Danz et al., 2005*). For example, 26 agricultural variables (including pesticide runoff and leaching, cropland area, nitrogen and phosphorus exports, percent of county treated for various pests and livestock inventories) comprised an agricultural category. Principal components analysis (PCA) within each category of

environmental variation was used to reduce dimensionality and derive comprehensive gradients for agriculture, atmospheric deposition, point source pollution and urbanization. For each common taxon, species relative abundance data were regressed against the set of comprehensive watershed-level predictors using multiple linear regression and evaluated using the coefficient of determination (R2). The 'stress power' value for a taxon represents its R2 position on the gradient of the R2 values for all common taxa. Taxa with higher values along this gradient had more acute relationships with watershed stressors, whether they reflected high or low stress, and so those taxa are assumed to be better indicators of conditions. To derive 'stress rank,' a canonical correspondence analysis (CCA) was performed including the species assemblages and the comprehensive stressor variables. A primary gradient of stress, largely driven by agricultural activities, was derived by the distillation of these data, similar to that achieved by *Reavie (2007)*. A taxon's stress rank was taken as its score relative to that stressor gradient, standardized by the range of scores for all common taxa.

*Amphora* Ehrenberg

*Amphora aequalis* Krammer (Figs. 6T–6V)
This taxon agrees with *Krammer*'s *(1980)* depiction, although it was sometimes indistinguishable from *A. inariensis* as described in the same article. In general, using LM the puncta should be visible in *A. aequalis*, especially on the dorsal valve, but this was not always certain. *A. inariensis* has equal conopeum development of the ventral and proximal dorsal regions while *A. aequalis* has a ribbed dorsal distal region, but neither of these features are easily visible using light microscopy. Krammer gives valve length range 17–33 µm; width range 3–6 µm; dorsal stria density 14–18/10 µm. Great Lakes specimens have valve length range 17–22 µm, width range 3.5–4.5 µm, dorsal stria density 18–20/10 µm. Environmental characteristics for this species (Fig. 2A) indicate occurrence across the Great Lakes and especially in mesotrophic protected wetlands. The TP optimum is 25 µg/L while the chloride optimum is 12 µg/L. It is a weak indicator of anthropogenic stress.

*Amphora calumetica* (B.W.Thomas) Peragallo (Figs. 7C and 7D)
This distinctive diatom is easily identified by the occurrence of a hyaline crest and its strongly bilobate structure. Great Lakes specimens agree with depictions by *Levkov (2009)*, *Patrick & Reimer (1975)*, *Stoermer & Yang (1971)* and *Edlund & Stoermer (1999)*. Based on combined literature, this species can have valve length range 38–87 µm; width range 8–13 µm; dorsal stria density 12–15/10 µm. Great Lakes specimens have valve length range 39–52 µm, width range 8.5–11 µm, dorsal stria density 15/10 µm.

*Amphora* cf. *eximia* J.R.Carter (Figs. 7I–7M)
Great Lakes specimens generally agree with depictions of this taxon by *Hofmann, Werum & Lange-Bertalot (2011)*, although longer and more finely striate specimens than their maxima were observed. This taxon may be the same as *A. subcostulata Stoermer & Yang (1971)* as previously observed from Lake Michigan, although shortened striae that occasionally occur in the central area of *A. subcostulata* were not observed. For *A. eximia*

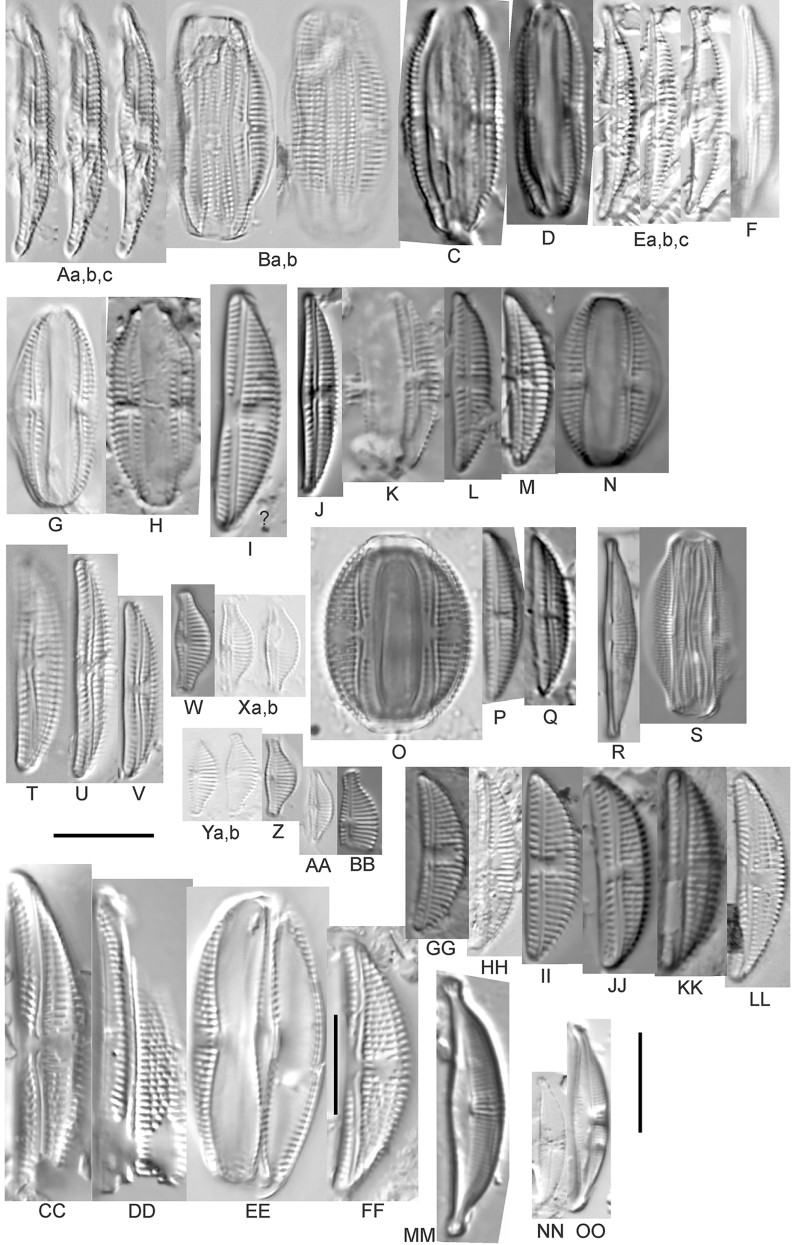

**Figure 6** Diatom light microscope images. (A–H) *Amphora neglecta* Stoermer & J.J.Yang [374, 213, 591, 591, 339, 374, 417, 700]; (I–Q) *Amphora* cf. *inariensis* Krammer [606, 378, 581, 739, 591, 698, 325, 606, 591]; (R and S) *Halamphora coffeaeformis* (C. Agardh) Z.Levkov [225, 462]; (T–V) *Amphora aequalis* Krammer [299, 278, 275]; (W-BB) *Halamphora thumensis* (A.Mayer) Z.Levkov [425, 425, 425, 275, 273, 461]; (CC–FF) *Amphora* cf. *macedoniensis* Nagumo [470, 461, 352, 369]; (GG–LL) *Amphora inariensis* Krammer [588, 361, 425, 739, 696, 325]; (MM) *Halamphora* cf. *bullatoides* M.H.Hohn & Hellerman [591]; (NN and OO) *Halamphora montana* (Krasske) Z.Levkov [299, 470]. Lowercase letters indicate multiple images of the same specimen. Square brackets contain sampling locales (Fig. 1) for each specimen. Scale bars: 10 mm.

*Hofmann, Werum & Lange-Bertalot (2011)* give valve length range 14–26 μm, width range 3.5–5.5 μm, dorsal stria density 18–20/10 μm. Great Lakes specimens have valve length range 12–24 μm, width range 3–5 μm, dorsal stria density 18–24/10 μm. One specimen

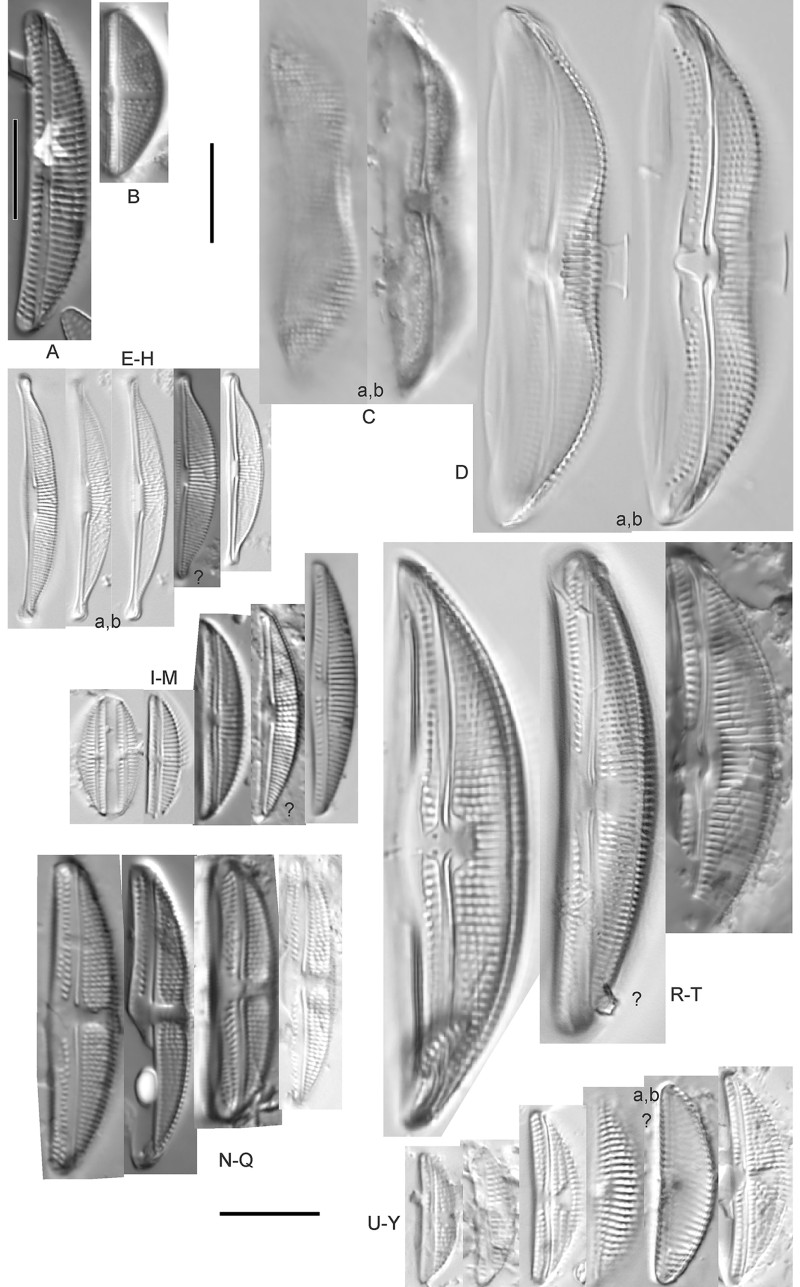

**Figure 7** **Diatom light microscope images.** (A) *Amphora* sp. 104 [461]; (B) *Amphora* sp. 106 [411]; (C and D) *Amphora calumetica* (B.W.Thomas) Peragallo [325, 388]; (E–H) *Halamphora oligotraphenta* (Lange-Bertalot) Z.Levkov [273, 273, 273, 273]; (I–M) *Amphora* cf. *eximia* J.R.Carter [417, 213, 591, 735, 225]; (N–Q) *Amphora sibirica* Skvortzow & Meyer [739, 356, 700, 374]; (R–T) *Amphora ovalis* (Kützing) Kützing [192, 110, 365]; (U–Y) *Amphora* cf. *libyca* Ehrenberg [299, 299, 299, 361, 68]. Lowercase letters indicate multiple images of the same specimen, and a question mark indicates a specimen with taxonomic uncertainty as described in the text. Square brackets contain sampling locales (Fig. 1) for each specimen. Scale bars: 10 mm.                    

(Fig. 7L) may require further taxonomic separation because of clearly punctate striae and no visible ventral striae.

*Amphora inariensis* Krammer (Figs. 6GG–6LL)
This taxon agrees with *Krammer (1980)*, although I was sometimes unable to distinguish it from *A. aequalis* as described in the same article. In general, unlike *A. aequalis* the puncta should not be visible using light microscopy, particularly in the dorsal section of *A. inariensis*, but this was not always obvious. Krammer gives valve length range 10–28 μm, width range 3–6 μm, dorsal stria density 15–20/10 μm. Great Lakes specimens have valve length range 16–21 μm, width range 4–6 μm, dorsal stria density 15–20/10 μm. Environmental characteristics for this species (Fig. 2C) indicate occurrence across the Great Lakes and in all geomorphic habitats. The TP optimum is 24 μg/L while the chloride optimum is 16 μg/L. It is a weak indicator of anthropogenic stress.

*Amphora* cf. *inariensis* Krammer (Figs. 6I–6Q)
Several ambiguous specimens of *Amphora* were difficult to confirm. These specimens were similar to *Krammer & Lange-Bertalot*'s *(1986)* depiction of *A. inariensis* but also had similarities to smaller forms of *A. ovalis* var. *affinis* (Kützing) Van Heurck (*Patrick & Reimer, 1975*), with the exception of *A.* cf. *libyca* having a relatively straight raphe. Usually specimens had an open central area, but sometimes there was a barely visible row of puncta in line with the tips of the dorsal striae. While having similarities to *Amphora ovalis* var. *affinis*, these specimens tended to be shorter than the minimum length described by *Patrick & Reimer (1975)*. Great Lakes specimens have valve length range 16–24 μm, width range 4–5.5 μm, dorsal stria density 16–20/10 μm. Specimen Fig. 6I is especially wide and may be a different species.

*Amphora* cf. *libyca* Ehrenberg (Figs. 7U–7Y)
This uncertain taxon appears to be similar to several possible species descriptions. Superficially these Great Lakes specimens agree with *Krammer & Lange-Bertalot*'s *(1986*; their pl. 149:3–11) and *Krammer*'s *(1980)* depiction of *A. libyca*, although our specimens were typically much smaller than their suggested minimum size, with finer striae density. Based largely on the arrangement of dorsal striae and raphe structure, this taxon could also be *Amphora rotunda* Skvortzow (*Stoermer & Yang, 1971*), *Amphora copulata* (Kützing) *Schoeman & Archibald (1986)* or *Amphora ovalis* v. *affinis* (Kützing) van Heurck (*Patrick & Reimer, 1975*). For *A. libyca Krammer & Lange-Bertalot (1986)* give valve length range 20–80 μm, width range 5.5–35 μm, dorsal stria density 11–15/10 μm. Great Lakes specimens have valve length range 13–22 μm, width range 4–6 μm, dorsal stria density 19–22/10 μm. Environmental characteristics for this species (Fig. 2B) indicate occurrence across the Great Lakes, especially in Lake Erie, and in all geomorphic habitats. The TP optimum exceeds the eutrophic threshold at 35 μg/L while the chloride optimum is relatively high at 36 μg/L. It is a strong indicator of medium levels of anthropogenic stress.

*Amphora* cf. *macedoniensis* Nagumo (Figs. 6CC–6FF)
This taxon somewhat agrees with depictions of *A. macedoniensis* by *Nagumo (2003)*.
During Great Lakes assessments these specimens were identified as a small form of
*A. libyca* Ehrenberg (*Krammer, 1980*). Because of a lack of detailed micrographs and the
large size of Great Lakes specimens, confirming this nomenclature would be premature.
For *A. macedoniensis* the published valve length range is 13–30 µm; width range 3–6 µm;
dorsal stria density 16–18/10 µm. Great Lakes specimens have valve length range
26–~38 µm, width range 6–7 µm, dorsal stria density 16–19/10 µm.

*Amphora neglecta* Stoermer & J.J.Yang (Figs. 6A–6H)
This taxon largely agrees with *Stoermer & Yang*'s *(1971)* depiction, although our specimens
tended to have a higher dorsal striae count. Some specimens intergraded with *A. aequalis*
and *A. inariensis*, though attempts were made to distinguish *A. neglecta* by the occurrence
of rostrate to subcapitate apices. Previously published valve length range 16–30 µm, width
range 3.5–5 µm, dorsal stria density 16–20/10 µm. Great Lakes specimens have valve
length range 19–36 µm, width range 3.5–4.5 µm, dorsal stria density 16–20/10 µm.

*Amphora ovalis* (Kützing) Kützing 1844; (Figs. 7R–7T)
This robust *Amphora* matches depictions by *Patrick & Reimer*'s *(1975)* and *Krammer &
Lange-Bertalot (1986)*, although the dorsal striae near the central area were very faint or
not visible. Published valve length range 35–85 µm; width range 9–17 µm; dorsal stria
density 10–12/10 µm. Great Lakes specimens have valve length range 48–58 µm, width
range 10–15 µm, dorsal stria density 11–13/10 µm, with one questionable specimen having
a density as high as 14/10 µm. Environmental characteristics for this species (Fig. 2D)
indicate occurrence across the Great Lakes and in all geomorphic habitats, especially
embayments and riverine wetlands. The TP optimum just exceeds the eutrophic threshold
at 32 µg/L while the chloride optimum is 13 µg/L. It is a relatively weak indicator of
medium levels of anthropogenic stress.

*Amphora pediculus* (Kützing) Grunow (Figs. 8A–8HHHHH)
Despite being the most photographed diatom from the GLEI project, identifying this small,
widespread *Amphora* by light microscope is problematic. In many cases in the Great Lakes,
larger forms of *A. pediculus* cannot be distinguished from *A. inariensis*. Despite *Krammer*'s
*(1980)* statement that the dorsal raphe in *A. pediculus* is wider than the ventral raphe
(whereas both sides are identical in *A. inariensis*), this distinction was not easily
established. Most often this taxon was distinguished by its usual smaller size. Attempts
were made to distinguish *Amphora perpusilla* (Grunow) Grunow from *A. pediculus*, but
substantial intergrading and difficulty resolving differences (*Krammer, 1980*; *Patrick &
Reimer, 1975*; *Levkov, 2009*) favored a single designation. Though both species are
considered unique based on the latest taxonomic literature, it has long been suggested by
some authors that they are identical (*Schoeman & Archibald, 1978*). However, I
acknowledge that *A. perpusilla* may be an unrecognized member of this complex.
Similarities were also noted with *A. subcostulata* *Stoermer & Yang (1971)*, which may be
present in my complex, but the usual lack of a shortened central stria favored *A. pediculus*.

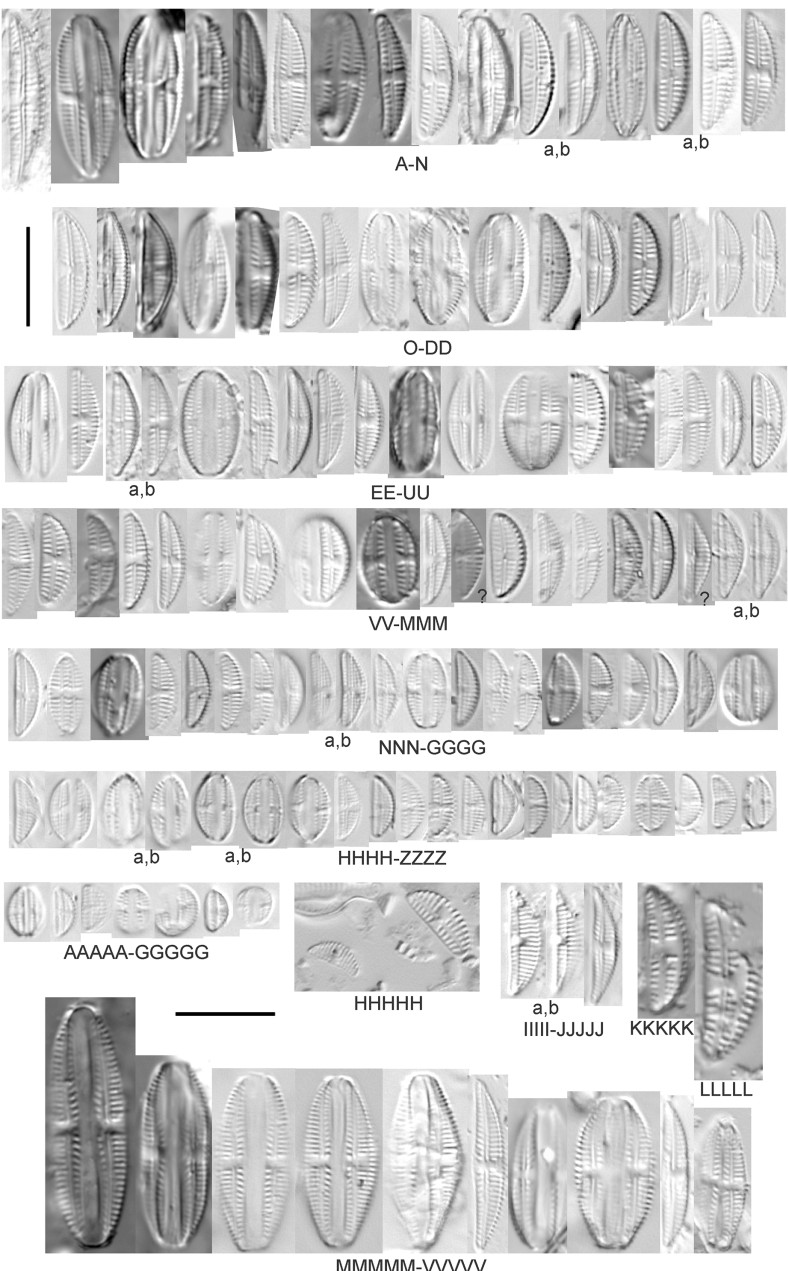

**Figure 8 Diatom light microscope images.** (A–GGGGG) *Amphora pediculus* (Kützing) Grunow [374, 622, 591, 591, 462, 393, 700, 462, 420, 105, 311, 311, 294, 273, 425, 374, 591, 369, 700, 425, 425, 294, 374, 389, 422, 311, 273, 420, 425, 425, 393, 389, 374, 325, 393, 311, 275, 273, 596, 393, 68, 361, 462, 299, 273, 354, 294, 273, 275, 461, 68, 273, 299, 425, 294, 361, 273, 459, 311, 420, 425, 420, 68, 453, 311, 299, 393, 361, 275, 273, 273, 278, 208, 273, 425, 294, 325, 299, 393, 411, 311, 369, 281, 275, 470, 425, 393, 68, 311, 294, 208, 294, 425, 417, 294, 299, 275, 311, 311, 299, 273, 192, 311, 311, 192, 425, 425, 393, 299, 192, 311]; (HHHHH) *Amphora pediculus* (Kützing) Grunow (two specimens showing variation in striae density) [425]; (IIIII and JJJJJ) *Amphora pediculus* (Kützing) Grunow (fine, capitate form) [102, 374]; (KKKKK and LLLLL) *Amphora pediculus* (Kützing) Grunow (coarse form) [247, 325]; (MMMMM–VVVVV) *Amphora pediculus/inariensis* [617, 617, 393, 213, 104, 393, 369, 339, 104, 417]. Lowercase letters indicate multiple images of the same specimen, and a question mark indicates a specimen with taxonomic uncertainty as described in the text. Square brackets contain sampling locales (Fig. 1) for each specimen. Scale bars: 10 mm.

It is highly likely that higher resolution observations would resolve several species from this abundant taxon. Valve length range 4–24 μm; width range 2–5 μm; dorsal stria density 18–28/10 μm. Great Lakes specimens have valve length range 4.5–17 μm, width range 2–4 μm, dorsal stria density 17–28/10 μm. Certain specimens are especially questionable by lacking the dorsal space in the striae (*e.g.*, Fig. 8FFF) or by having slight capitation in the ends (*e.g.*, Figs. 8LLL and 8EEEE). Environmental characteristics for this species (Fig. 2E) indicate occurrence across the Great Lakes and in all geomorphic habitats. The TP optimum indicates a preference for mesotrophic conditions at 24 μg/L while the chloride optimum is 15 μg/L. It is a fair indicator of medium levels of anthropogenic stress.

*Amphora pediculus* (Kützing) Grunow (fine, capitate form) (Figs. 8IIIII–8JJJJJ)
With the lack of a dorsal space in the central area and slightly capitate valve, these specimens may represent an undescribed species.

*Amphora pediculus* (Kützing) Grunow (coarse form) (Figs. 8KKKKK–8LLLLL)
These two examples have very wide valves with low striae densities (14/10 μm).

*Amphora pediculus/inariensis* (Figs. 8MMMMM–8VVVVV)
As described in more detail under *A. pediculus* above, features of both *A. pediculus* and *A. inariensis* are not easily distinguished in larger valves, as illustrated in this larger set of specimens. These specimens have valve length range 13–24 μm, width range 2.5–4 μm, dorsal stria density 18–24/10 μm.

*Amphora sibirica* Skvortzow & Meyer (Figs. 7N–7Q)
This taxon matches *Stoermer & Yang*'s *(1971)* depictions of "*A. siberica*" (orthographic error) from Lake Michigan collections. Great Lakes specimens have a higher dorsal striae count than described by *Nagumo (2003)*. From the literature: valve length range 25–40 μm, width range 5–8 μm, dorsal stria density 16–24/10 μm. Great Lakes specimens have valve length range 25–31 μm, width range 5–7.5 μm, dorsal stria density 20–27/10 μm.

*Amphora* sp. 104 (Fig. 7A)
This rare taxon from an embayment at Caseville (Saginaw Bay, Lake Huron) could not be adequately matched with known species of *Amphora*. Valve shape and raphe structure was like *A. libyca*, although no gap in the central, dorsal striae was observed and the dorsal margin was flatly arched. Valve length range 32 μm, width 7 μm, dorsal stria density 14 μm.

*Amphora* sp. 106 (Fig. 7B)
This rare taxon from a protected wetland in the Cheboygan River outlet (Lake Huron) did not match well with any known species of *Amphora*. Valves had a strong convex dorsal margin and straight ventral margin. The axial area was narrow. Striae were punctate on the dorsal side with a distinct gap in the central area. Ventral striae and raphe characteristics were not well observed. Specimens were similar to *Amphora michiganensis Stoermer & Yang (1971)*, but in our specimens the axial area was closer to the ventral margin. Valve length range 16–17 μm; width range 5–7 μm; dorsal stria density 20/10 μm.

*Halamphora* (Cleve) Z.Levkov

*Halamphora* cf. *bullatoides* M.H.Hohn & Hellerman (Fig. 6MM)
This diatom is similar to *Stoermer & Yang*'s *(1971)* and *Patrick & Reimer*'s *(1975)*
depiction of the species (as *Amphora*). The specimen appears to have a slightly sunken
dorsal central area and a too-high striae density, resulting in intergrading with
*H. montana*. For *H. bullatoides Patrick & Reimer (1975*, as *Amphora*) give valve length
range 17–30 µm, width range 4–6 µm, dorsal stria density 9–14/10 µm. The Great Lakes
specimen has valve length range 26 µm, width 5.5 µm, dorsal stria density 18/10 µm, much
finer near the ends.

*Halamphora coffeaeformis* (C.Agardh) Z.Levkov (Figs. 6R and 6S)
Great Lakes specimens matched depictions of this species (as *Amphora*) by *Archibald &
Schoeman (1984)* and *Patrick & Reimer (1975)*. This taxon intergrades with *A. veneta* but is
generally distinguished by having little or no apparent decrease in striae count at the dorsal
central area. *Stepanek (2011a)* gives valve length range 15–40 µm; width range 5–7 µm; dorsal
stria density 17–21/10 µm at center, 22–24/10 µm near the ends. Great Lakes specimens have
valve length range 18–20 µm, width 4 µm, dorsal stria density 26–32/10 µm, finer at ends.
Environmental characteristics for this species are provided in Fig. 2F. As was typical for
many species of *Halamphora* and *Amphora* in the Great Lakes, *H. coffeaeformis* had
relatively high phosphorus and chloride optima, though it is a weak indicator of a medium
amount of anthropogenic stress. It was observed particularly in Lake Erie protected wetlands.

*Halamphora montana* (Krasske) Z.Levkov (Figs. 6NN and 6OO)
Great Lakes specimens matched depictions of this species (as *Amphora*) by *Krammer &
Lange-Bertalot (1986*; valve length range 9–25 µm, width range 3–6 µm, dorsal stria density
27–40/10 µm). This taxon intergrades with *H. bullatoides* but is generally distinguished by
a dorsal central area that is vacant or nearly free of striae. Specimen valve length 26 µm,
width range 5.5 µm, dorsal stria density 28 and higher/10 µm.

*Halamphora oligotraphenta* (Lange-Bertalot) Z.Levkov (Figs. 7E–7H)
This taxon exhibited great variation in valve shape, and during original GLEI assessments
depictions of *Amphora veneta* Kützing by *Krammer & Lange-Bertalot (1986)* were used for
identification. Similar specimens to those from the Great Lakes are presented by *Germain
(1981)*, *Patrick & Reimer (1975)* and *Krammer & Lange-Bertalot (1986)* as *A. veneta*.
Expression of the ends varied, with some being ventrally deflected and others being
distinctly capitate and/or bulbous (as in *A. bullatoides*). Some intergrading with
*A. coffeaeformis* was observed, but previous descriptions of *A. veneta* were usually
distinguished by a coarser strial density near the central dorsal area. More recent
taxonomic scrutiny separates this species as *H. oligotraphenta* (*Levkov, 2009*). *Stepanek
(2011b)* gives valve length range 22–33 µm; width range 3.9–4.6 µm; dorsal stria density
26–34/10 µm. Great Lakes specimens are relatively small: valve length range 19–25 µm,
width range 3.5–4 µm, dorsal stria density 28–36/10 µm. Environmental characteristics for
this species (Fig. 2G) indicate occurrence across coastal wetlands of the Great Lakes, except

for Lake Superior. The TP optimum is just below the eutrophic threshold of 30 µg/L while the chloride optimum is 10 µg/L. It is a very weak indicator of medium levels of anthropogenic stress.

*Halamphora thumensis* (A.Mayer) Z.Levkov (Figs. 6W–6BB)
Great Lakes specimens are well depicted by *Krammer (1980)* and *Kingston, Lowe & Stoermer (1980)* (as *Amphora*). From the literature valve length range 7–15 µm, width range 3–5 µm, dorsal stria density 20–30/10 µm. Great Lakes specimen valve length 7–11 µm, width range 3–4 µm, dorsal stria density 24–30/10 µm. Environmental characteristics for this species (Fig. 2H) indicate occurrence particularly in Lake Huron, and in all geomorphic habitats especially coastal wetlands. The TP optimum indicates is a mesotrophic indicator with a chloride optimum of 11 µg/L. It is a very weak indicator of medium levels of anthropogenic stress.

*Cymbella* C.Agardh

*Cymbella dorsenotata* Østrup (Figs. 9P and 9Q)
While very similar to *Cymbella neocistula* var. *islandica*, the more lunate shape and presence of multiple stigmata at the central area suggest this is similar to *Cymbella dorsenotata*, though certain specimens (*e.g.*, Fig. 9P; *C.* cf. *dorsenotata*) have a higher striae density. *Krammer (2002)* gives length ~62–170 µm; width range 19–26 µm; stria density 5.5–8/10 µm, but as high as 12/10 µm at the ends. The more certain Great Lakes specimen gives length ~49 µm; width 12 µm; stria density 8/10 µm at the dorsal central area, and up to 12/10 µm at the ends.

*Cymbella affinis* Kützing (Figs. 10A–10E)
Great Lakes specimens of this species match well with *Potapova*'s *(2011)* depiction from Idaho, but not so much with those shown by *Hofmann, Werum & Lange-Bertalot (2011)* and *Krammer (2002)*, which have shortened striae around the central area, dorsal side. *Potapova (2011)* gives valve length range 19–36 µm; width range 6.9–9 µm; stria density 9–12/10 µm in the center valve, 13–14/10 µm at the ends. Great Lakes specimens have valve length range 25–33 µm; width range 7.5–8 µm; stria density 12/10 µm. Based on the striae structure around the central area these specimens may be representatives of *Cymbella affiniformis* Krammer (*Krammer, 2002*) originally identified from Germany, but evaluating associations between European and North America examples is prudent. Environmental characteristics for this species (Fig. 2I) indicate occurrence across the Great Lakes and in all geomorphic habitats, especially nearshore and embayments. The TP optimum is 11 µg/L and the chloride optimum is 11 µg/L. It is a weak indicator of relatively low levels of anthropogenic stress.

*Cymbella* cf. *aspera* (Ehrenberg) Cleve (Fig. 11F)
The shape and size of this specimen is similar to *C. aspera*, among other representatives of the *Cymbella cistula* complex (*Krammer, 2002*), but the striae density is too high, and the lack of clear filiform proximal raphe ends precludes any of those species. The specimen has valve length 93 µm; width 16.5 µm; stria density 17/10 µm.
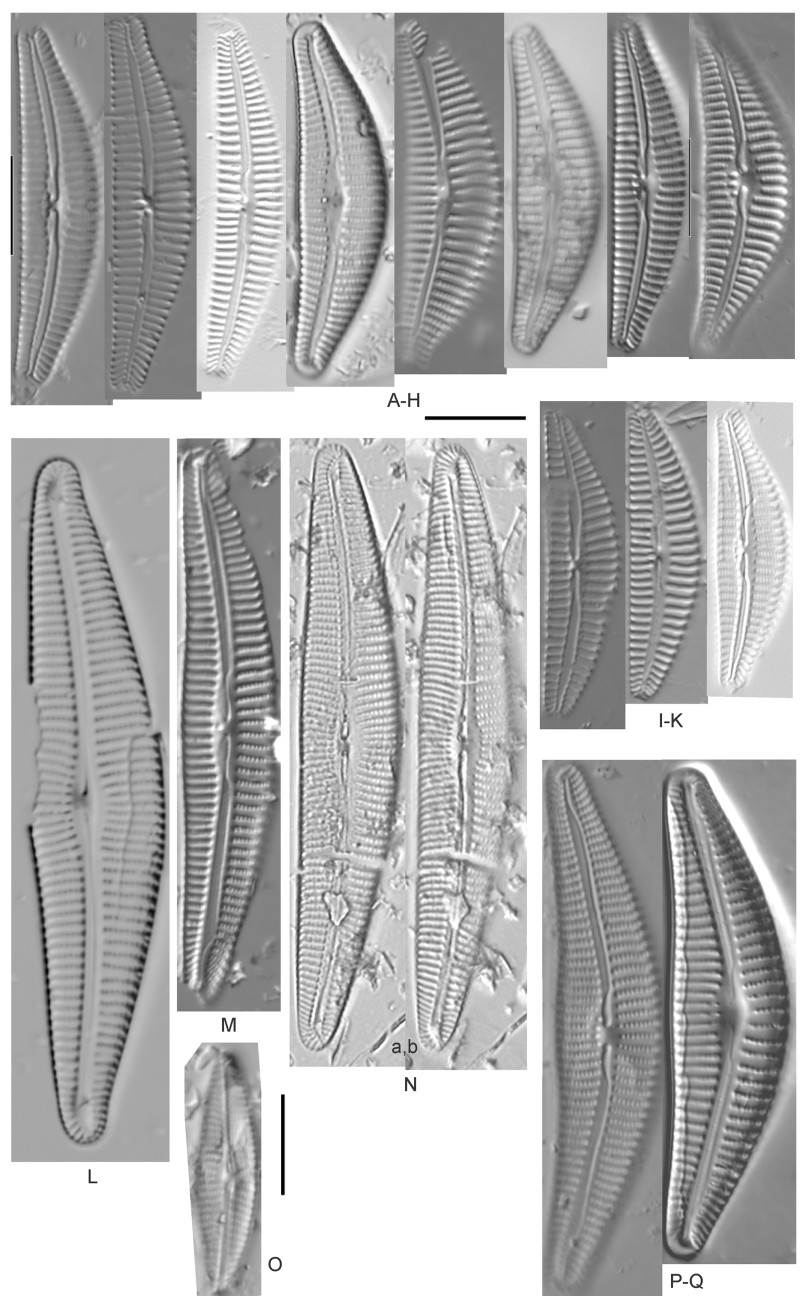

**Figure 9 Diatom light microscope images.** (A–K) *Cymbella parva* (W.Smith) Kirchner [420, 420, 420, 247, 419, 247, 420, 419, 419, 420, 420]; (L) *Cymbella* cf. *lange-bertalotii* Krammer [453]; (M) *Cymbella vulgata* Krammer [420]; (N) *Cymbella* cf. *lange-bertalotii* Krammer [311]; (O) *Cymbella* sp. (Old Woman Creek, Lake Erie) [610]; (P) *Cymbella* cf. *dorsenotata* Østrup [290]; (Q) *Cymbella dorsenotata* Østrup [375]. Lowercase letters indicate multiple images of the same specimen. Square brackets contain sampling locales (Fig. 1) for each specimen. Scale bars: 10 mm.

*Cymbella compacta* Østrup (Figs. 10I–10K)

This diatom is well-described by previous authors such as *Krammer (2002)* and *Bahls (2016a)*. Krammer gives valve length range 28–76 μm; width range 11–15 μm; stria density 10–14/10 μm in the dorsal center, 12–14/10 μm at the ends. Great Lakes specimens give
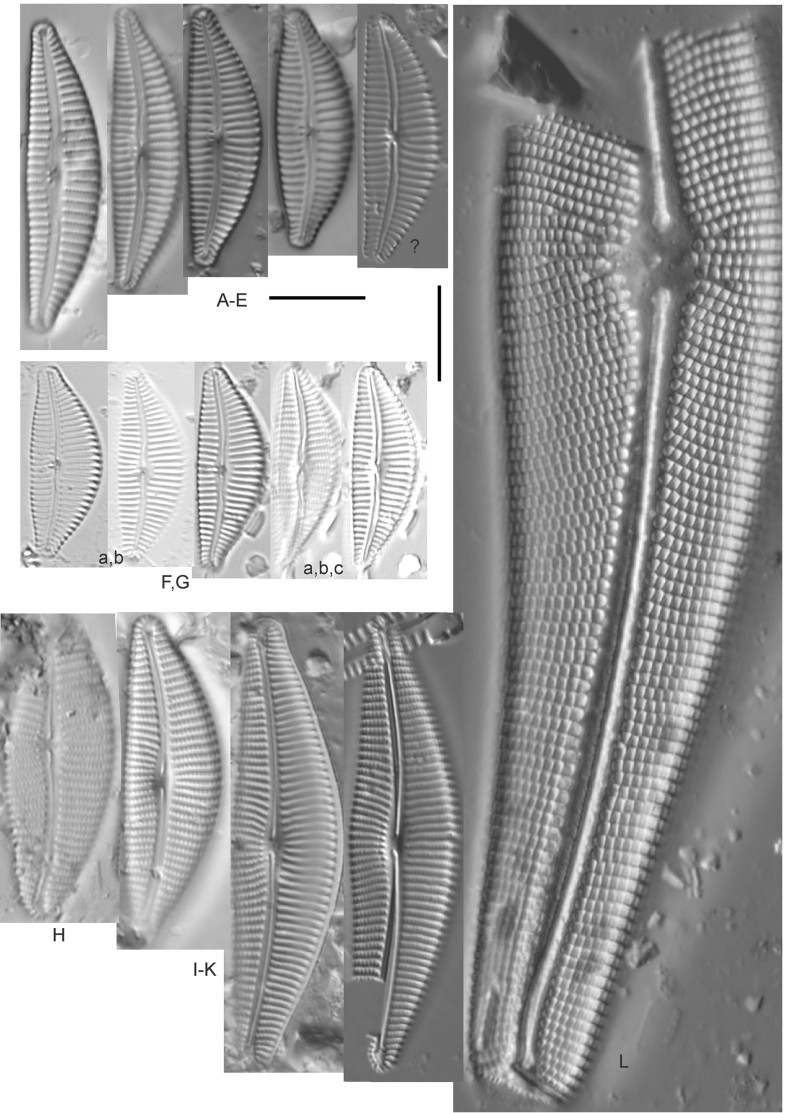

**Figure 10 Diatom light microscope images.** (A–E) *Cymbella affinis* Kützing [247, 393, 470, 417, 420]; (F anf G) *Cymbella* cf. *excisa* Kützing [311, 311]; (H) *Cymbella* cf. *compacta* Østrup [746]; (I–K) *Cymbella compacta* Østrup [369, 350, 420]; (L) *Cymbella mexicana* (Ehrenberg) Cleve [325]. Lowercase letters indicate multiple images of the same specimen, and a question mark indicates a specimen with taxonomic uncertainty as described in the text. Square brackets contain sampling locales (Fig. 1) for each specimen. Scale bars: 10 mm.

valve length range 34–47 µm; width range 10–12 µm; stria density 12–13/10 µm in the dorsal center.

*Cymbella* cf. *compacta* Østrup (Fig. 10H)
Despite being a broken valve, the specimen is a fair fit for the species as described by *Krammer (2002)*, though I may have something that has a higher width to length ratio and less apiculate apices. The Great Lakes specimen has valve length 34 µm; width 12 µm; stria density 12/10 µm.

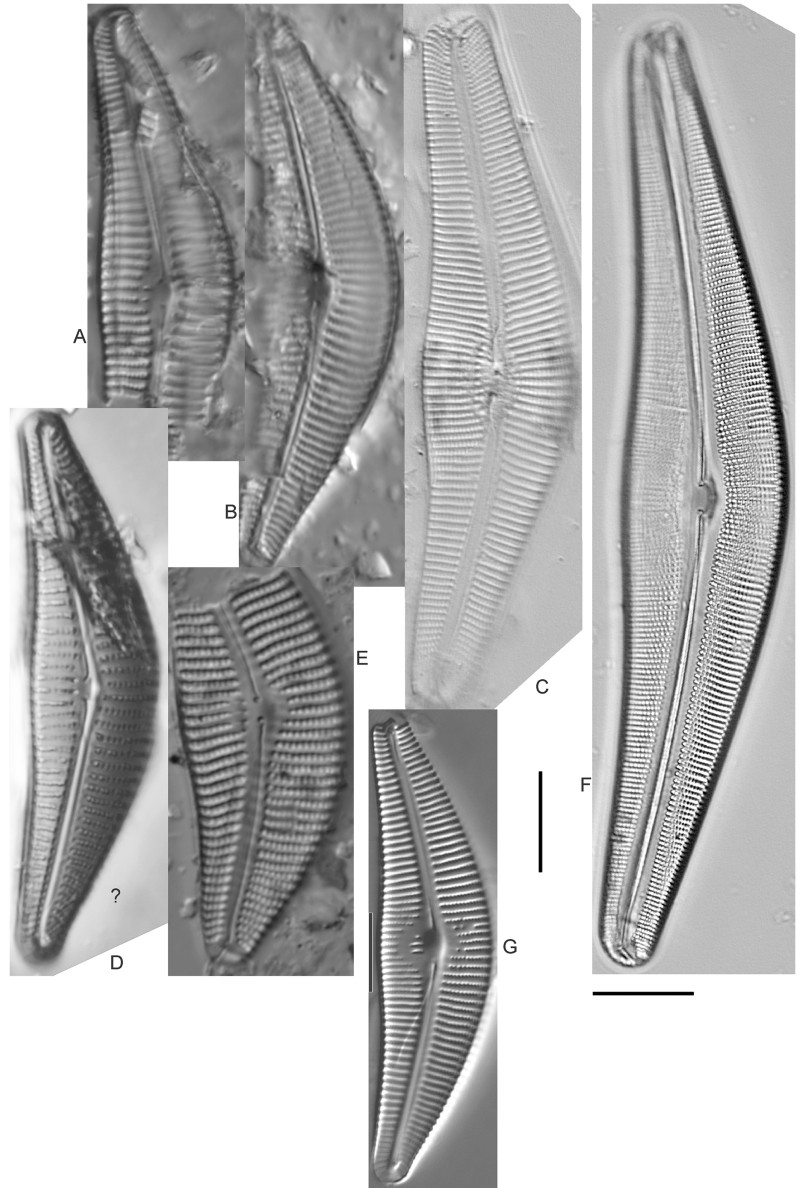

**Figure 11 Diatom light microscope images.** (A–E) *Cymbella neocistula* var. *islandica* Krammer [364, 606, 389, 470, 698]; (F) *Cymbella* cf. *aspera* (Ehrenberg) Cleve [208]; (G) *Cymbella* sp. (Muskegon, Lake Michigan) [378]. A question mark indicates a specimen with taxonomic uncertainty as described in the text. Square brackets contain sampling locales (Fig. 1) for each specimen. Scale bars: 10 mm.

*Cymbella* cf. *excisa* Kützing (Figs. 10F and 10G)

While originally identified as *Cymbella affinis*, the characteristic single, ventral stria connected to the stigma is like observations of this species by *Krammer (2002)*. However, striae densities of Great Lakes specimens exceeded those described by *Krammer (2002)*: valve length range 17–41 µm; width range 6–10.7 µm; stria density 9–13/10 µm in the dorsal center, 12–14/10 µm at the ends. Great Lakes specimens gives valve length range 20–21 µm; width range 7–7.5 µm; stria density 14–15/10 µm in the dorsal center.

*Cymbella* cf. *lange-bertalotii* Krammer (Figs. 9L and 9N)
For *Cymbella lange-bertalotii Hofmann, Werum & Lange-Bertalot (2011)* give valve length range 38–100 μm; width range 10–16 μm; dorsal stria density 8–12/10 μm. Great Lakes specimens have valve length range 60–69 μm; width range 10–14 μm; dorsal stria density 12–14/10 μm. While similar in many aspects, often including the subtle inflection of the raphe proximal ends, the slightly higher striae density in Great Lakes specimens prevents confident association with this species. Further, while the two specimens presented have similar identifying characteristics, the wider valve and slight inflection in the apices of Fig. 9L suggest additional taxonomic splitting may be warranted.

*Cymbella mexicana* (Ehrenberg) Cleve (Fig. 10L)
While larger than that presented by *Krammer (2002)* and *Patrick & Reimer (1975)*, this appears to be a good representative of the species. *Patrick & Reimer (1975)* give length 80–165 μm; width range 24–33 μm; stria density 7–8/10 μm in the dorsal center, becoming 9/10 μm at the ends. The Great Lakes specimen is length ~180 μm; width 31 μm; stria density 6–7/10 μm in the dorsal center.

*Cymbella neocistula* var. *islandica* Krammer (Figs. 11A–11E)
Despite their large, robust features, discerning the many taxa in the "*Cymbella cistula*" complex in the Great Lakes is challenging due to intergrading of species features such as shape and striae density, and a complicated history of taxonomic splitting, combining and renaming. *Cymbella cistula* (Ehrenberg) Kirchner was separated into three new taxa by *Krammer (2002)*, one of them being *Cymbella neocistula* Krammer, a variety of which was later determined to be the same as *Cymbella neocistula* var. *islandica*, which was identified from Lake Michigan as *Cymbella cistula* var. *gibbosa* Brun by *Patrick & Reimer (1975)*. Certain specimens (*e.g.*, Fig. 11D) have the strongly arched dorsal margin like that of *Cymbella perfossalis Krammer (2002)*, though that is only known to be a fossil species from the type locality in Oregon. *Patrick & Reimer (1975)* give valve length range 35–130 μm; width range 11–26 μm; dorsal stria density 6–11/10 μm. Great Lakes specimens have valve length range 50–69 μm; width range 13.5–16.5 μm; stria density 9–11/10 μm. Environmental characteristics for this species (Fig. 2J) indicate occurrence across the Great Lakes and in all geomorphic habitats.

*Cymbella parva* (W.Smith) Kirchner (Figs. 9A–9K)
Great Lakes observations of this species largely matched morphological parameters published by *Hofmann, Werum & Lange-Bertalot (2011)*, *Krammer (2002)* and *Bahls (2016b)*. *Hofmann, Werum & Lange-Bertalot (2011)* give valve length range 15–47 μm; width range 7–10 μm; dorsal stria density 9–11/10 μm, and as high as 13/10 μm at the apices. Great Lakes specimens have valve length range 28–38 μm; width range 7–10 μm; dorsal stria density 10–12/10 μm. Unlike previous descriptions, the characteristic central stigma was sometimes an extension of the middle stria, in some cases accompanied by smaller stigmata on the two adjacent striae. Ventral structure ranged among slightly convex, slightly concave, and flat. Some specimens also had a more constricted central area than previously observed. Though additional splitting of this taxon may be warranted,

these variations may be transient and not enough to negate the identification as *Cymbella parva*. For instance, *Krammer (2002)* presents a derivative species *Cymbella perparva* Krammer that is differentiated as having stronger dorsiventrality, but such a distinction was not apparent and would have been arbitrary in Great Lakes specimens.

*Cymbella vulgata* Krammer (Fig. 9M)
Though originally identified as *Cymbella affinis* Kützing, a more recent split to this more elongate, arcuate species by *Krammer (2002)* provides a comparable taxon. *Hofmann, Werum & Lange-Bertalot (2011)* give valve length range 20–50 µm; width range 7.8–12.7 µm; dorsal stria density 8–12/10 µm. The Great Lakes specimen has valve length range 55 µm; width 9 µm; dorsal stria density 10/10 µm. Although it is slightly longer than previous European observations, this appears to be a suitable identification.

*Cymbella* sp. (Muskegon, Lake Michigan) (Fig. 11G)
This interesting specimen has several structural similarities to *Cymbella neocistula* var. *islandica*, though the striae count is too dense, the characteristic inflection in the proximal raphe ends is not present, and the central area has a unique, circular central area containing a few short, excised striae where stigmata would normally be (as in *Cymbella proxima* Reimer). The specimen has valve length range 46 µm; width 11 µm; stria density 15/10 µm.

*Cymbella* sp. (Old Woman Creek, Lake Erie) (Fig. 9O)
This partly obscured specimen may be *Cymbopleura* but was not adequately identifiable as a known species. The Great Lakes specimen has valve length 24.5 µm; width 6 µm; dorsal stria density 14/10 µm.

*Cymbopleura* (Krammer) Krammer

*Cymbopleura acuta* (M.Schmidt) Krammer (Fig. 12H)
Although originally identified as *Cymbella ehrenbergii* Kützing during GLEI assessments, this specimen is too small to be associated with that species, which has since been combined by *Krammer (2003)* as *C. inaequalis*. *Krammer*'s *(2003)* depiction of *Cymbopleura acuta* appears to be a suitable match, though our specimen has a slightly more rhombic (than lanceolate) shape. *Krammer (2003)* gives valve length range 36–87 µm; width range 16–22 µm; stria density 8–11/10 µm at the central area, and up to 15/10 µm at the ends. The Great Lakes specimen gives valve length 49.5 µm; width 17 µm; stria density 10/10 µm at the dorsal central area, and up to 14/10 µm at the ends.

*Cymbopleura inaequalis* (Ehrenberg) Krammer (Fig. 13A)
This large *Cymbopleura* matches well with depictions by *Krammer (2003)*, who gives valve length range 90–190 µm; width range 32–44 µm; stria density 5–7/10 µm at the central area, and up to 11/10 µm at the ends. The Great Lakes specimen gives valve length 73 µm; width 28 µm; stria density 8/10 µm at the dorsal central area, and up to 13/10 µm at the ends. While smaller than previously observed, the otherwise distinctive shape and features substantiate a correct identification.

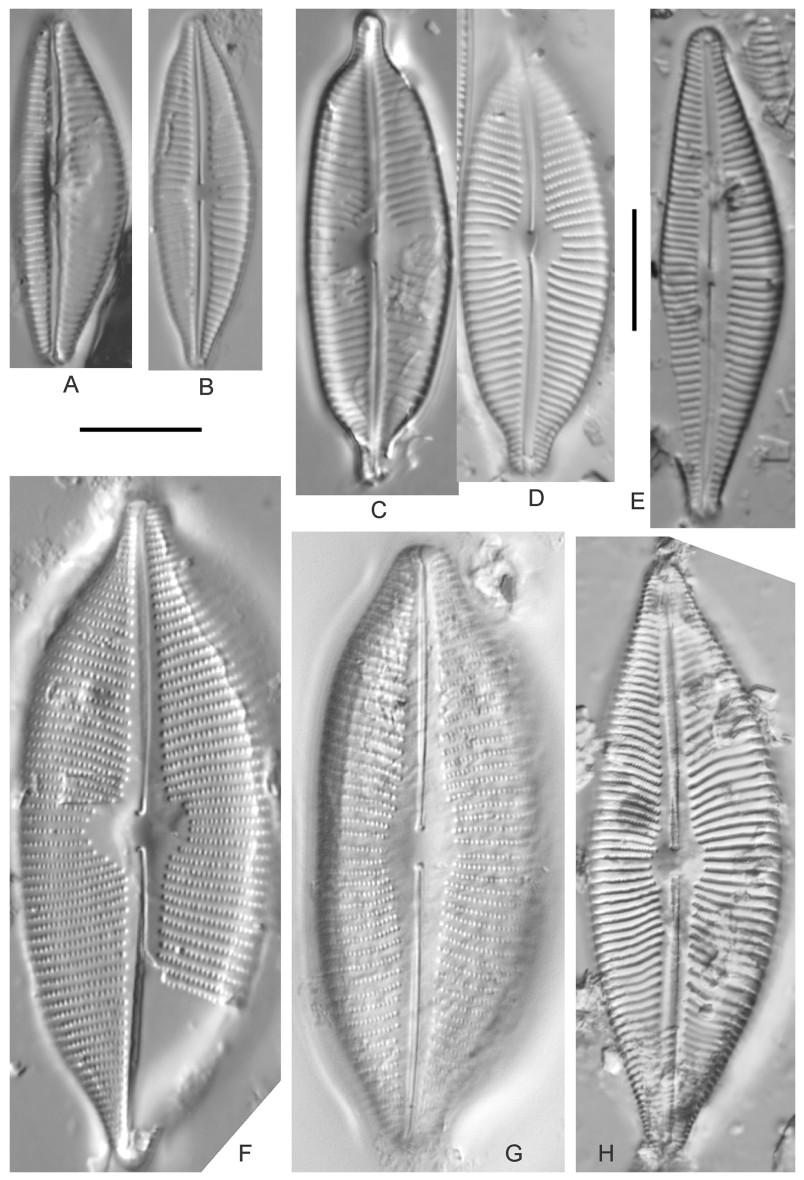

**Figure 12 Diatom light microscope images.** (A) *Cymbopleura* cf. *laeviformis* Krammer [423]; (B) *Cymbopleura* cf. *rupicola* (Grunow) Krammer [462]; (C) *Cymbopleura lanceolata* Krammer [461]; (D) *Cymbopleura* cf. *lanceolata* Krammer [393]; (E) *Cymbopleura* sp. 20 [364]; (F) *Cymbopleura sub-apiculata* Krammer [461]; (G) *Cymbopleura lata* var. *irrorata* Krammer [273]; (H) *Cymbopleura acuta* (M.Schmidt) Krammer [588]. Square brackets contain sampling locales (Fig. 1) for each specimen. Scale bars: 10 mm.                                

*Cymbopleura amphicephala* (Naegeli) Krammer (Figs. 14EE–14HH)
*Krammer (2003)* gives valve length range 22–34 μm; width range 7.2–8.7 μm; stria density 12–15/10 μm, up to 18/10 μm at the ends. The Great Lakes specimens give valve length range 21–29 μm; width range 6.5–8.5 μm; stria density 13–14/10 μm. Environmental characteristics for this species (Fig. 3A) indicate occurrence particularly in Lake Huron and most commonly in embayments. The TP optimum is 14 μg/L while the chloride optimum is relatively high at 43 μg/L. It is a weak indicator of medium levels of anthropogenic stress.

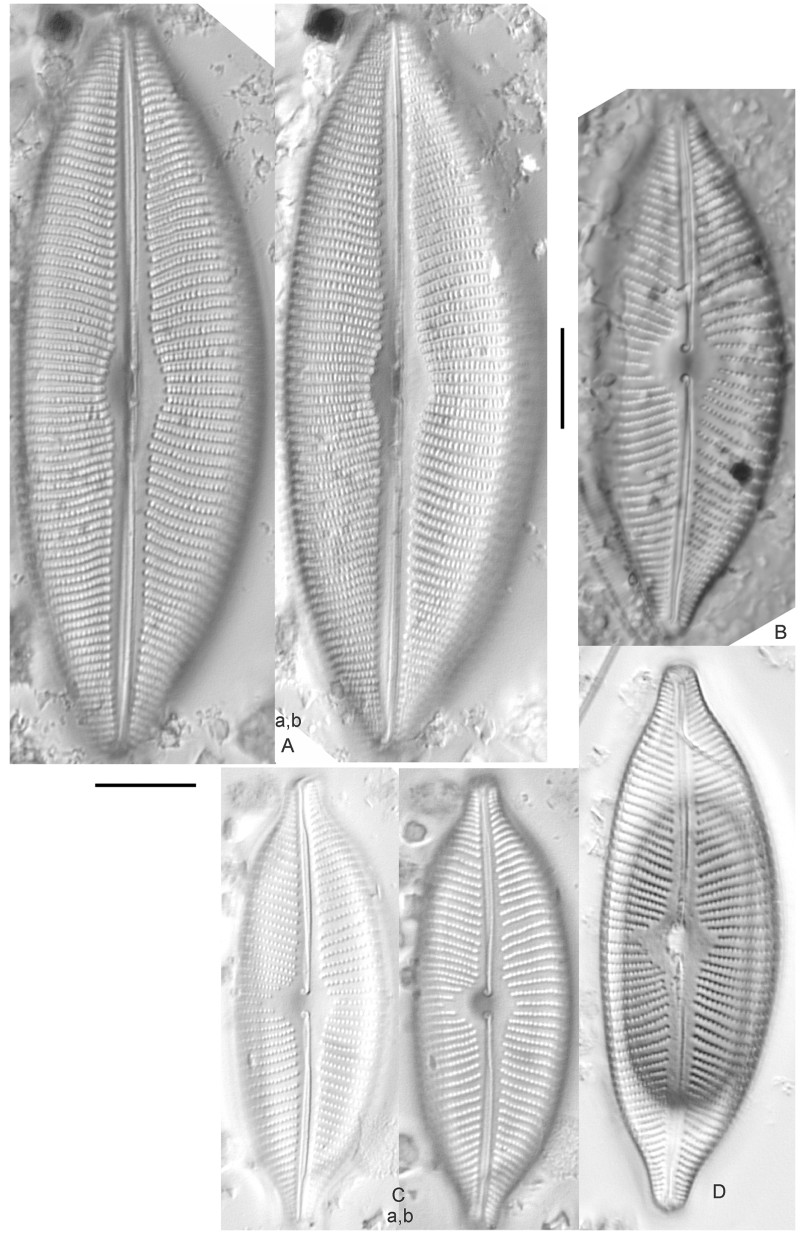

**Figure 13 Diatom light microscope images.** (A) *Cymbopleura inaequalis* (Ehrenberg) Krammer [420]; (B–D) *Cymbopleura subcuspidata* Krammer [581, 325, 375]. Lowercase letters indicate multiple images of the same specimen. Square brackets contain sampling locales (Fig. 1) for each specimen. Scale bars: 10 mm.                                         

*Cymbopleura* cf. *frequens* Krammer (Figs. 14BB–14DD)
While difficult to discern from *Cymbopleura amphicephala*, smaller specimens were a fair match to *Krammer*'s *(2003)* depiction of *Cymbopleura frequens*, with the exception of higher striae densities. The structure of the ends (apiculate/rostrate/capitate) appears to be transient and a poor parameter for separation of these species. *Krammer (2003)* gives valve length range 14–38 μm; width range 6.4–8.8 μm; stria density 11–14/10 μm, up to 15/10 μm at the ends. The Great Lakes specimens give valve length range 21–29 μm; width

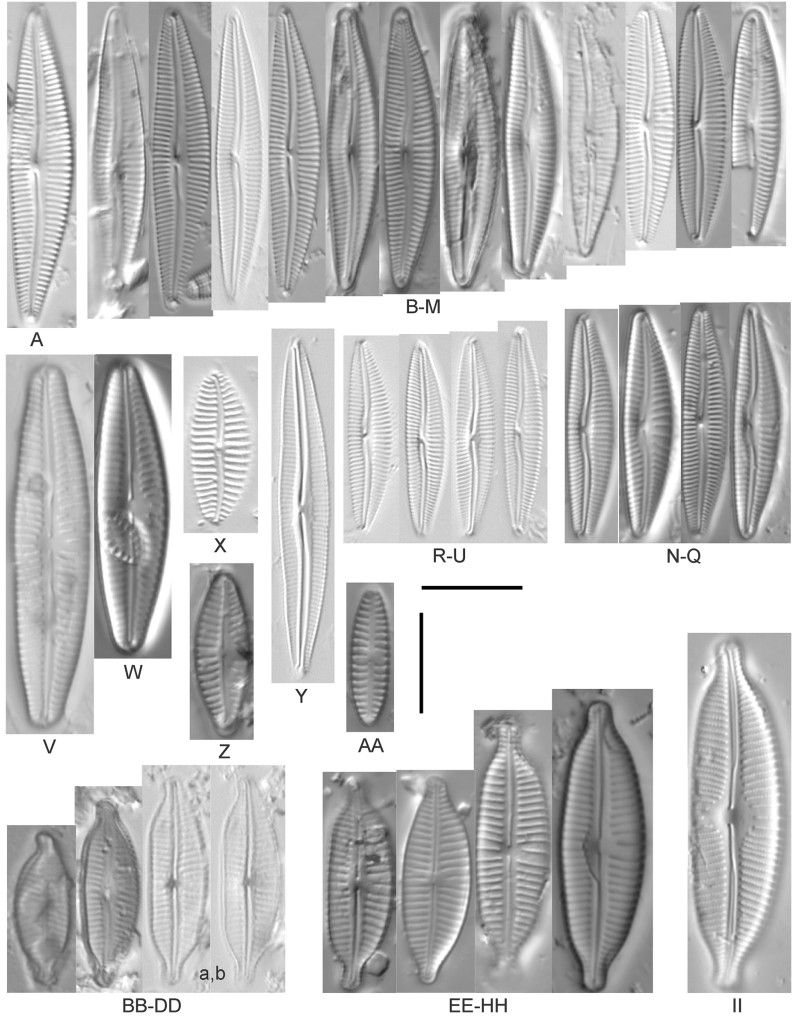

**Figure 14 Diatom light microscope images.** (A) *Delicatophycus* cf. *neocaledonicus* M.J.Wynne [273]; (B–Q) *Delicatophycus neocaledonicus* M.J.Wynne [411, 420, 389, 420, 470, 423, 411, 281, 419, 407, 420, 411, 420, 281, 419, 470]; (R–U) *Delicatophycus* sp. "Bell River 1" [420, 420, 420, 389]; (V and W) *Cymbopleura subaequalis* (Grunow) Krammer [247, 273]; (X) *Cymbopleura/Cymbella* (?) sp. [420]; (Y) *Delicatophycus* sp. "Bell River 2" [420]; (Z) *Cymbopleura/Cymbella* cf. *hustedtii* Krasske [411]; (AA) *Cymbopleura/Cymbella* (?) sp. (Garden Bay) [273]; (BB–DD) *Cymbopleura* cf. *frequens* Krammer [739, 420, 420]; (EE–HH) *Cymbopleura amphicephala* (Naegeli) Krammer [420, 202, 462, 364]; (II) *Cymbopleura naviculiformis* (Auerswald) Krammer [352]. Lowercase letters indicate multiple images of the same specimen. Square brackets contain sampling locales (Fig. 1) for each specimen. Scale bars: 10 mm.

range 6.5–8.5 μm; stria density 18–19/10 μm. No other taxa in the *Cymbopleura amphicephala* group have such high striae densities (*Krammer, 2003*), so this may be an undescribed species.

*Cymbopleura/Cymbella* cf. *hustedtii* Krasske (Fig. 14Z)
This small, possible example of this species is difficult to confirm without additional specimens. *Krammer (2002)* gives valve length range 13–26 μm; width range 5.7–8 μm;

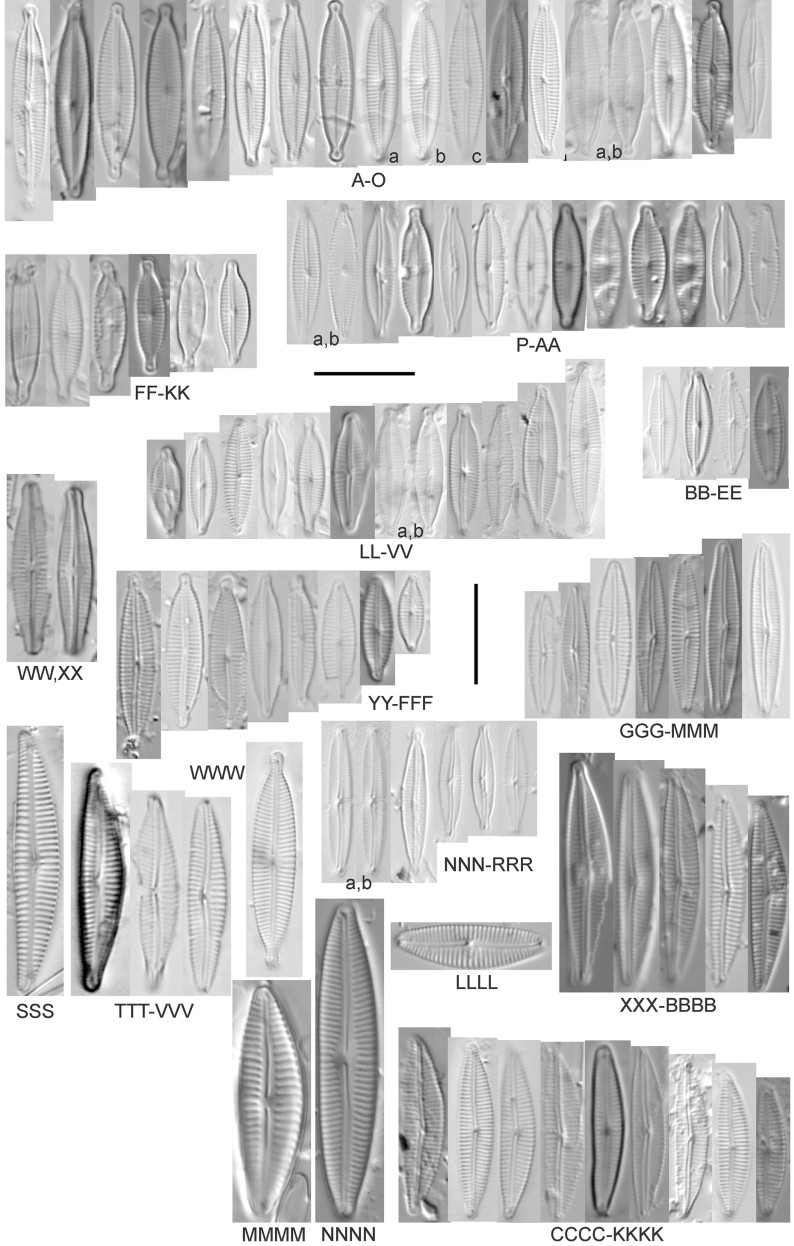

**Figure 15 Diatom light microscope images.** (A–EE) *Encyonopsis subminuta/krammeri* [29, 364, 247, 417, 247, 200, 200, 208, 419, 325, 273, 420, 290, 446, 420, 273, 407, 470, 420, 273, 420, 407, 459, 420, 459, 420, 420, 311, 311, 420, 417]; (FF–KK) *Encyonopsis subminuta/krammeri* (capitate + dorsiventral) [420, 420, 364, 202, 375, 273]; (LL–VV) *Encyonopsis* cf. *krammeri* "very fine" [356, 311, 311, 425, 420, 462, 420, 419, 278, 281, 208]; (WW–XX) *Encyonopsis subminuta* "broadly rounded form" [225, 225]; (YY–FFF) *Encyonopsis subminuta* Krammer & E.Reichardt [407, 88, 364, 247, 247, 247, 446, 311]; (GGG–MMM) *Encyonopsis thumensis* Krammer [420, 420, 420, 420, 420, 407, 389]; (NNN-RRR) *Encyonopsis* cf. *krammeri* "very fine and narrow" [389, 29, 420, 420, 420]; (SSS) *Encyonema* cf. *neogracile* Krammer [200]; (TTT–VVV) *Encyonema* sp. (?) [180, 180, 419]; (WWW) *Encyonopsis descripta* (Hustedt) Krammer [200]; (XXX–KKKK) *Encyonema* cf. *evergladianum* Krammer [419, 423, 411, 407, 420, 411, 273, 419, 407, 417, 407, 420, 273, 423]; (LLLL) *Encyonema* sp. 103 [420]; (MMMM) *Cymbopleura kuelbsii* v. *nonfasciata* Krammer [470]; (NNNN) *Cymbopleura incertiformis* Krammer [423]. Lowercase letters indicate multiple images of the same specimen. Square brackets contain sampling locales (Fig. 1) for each specimen. Scale bars: 10 mm.

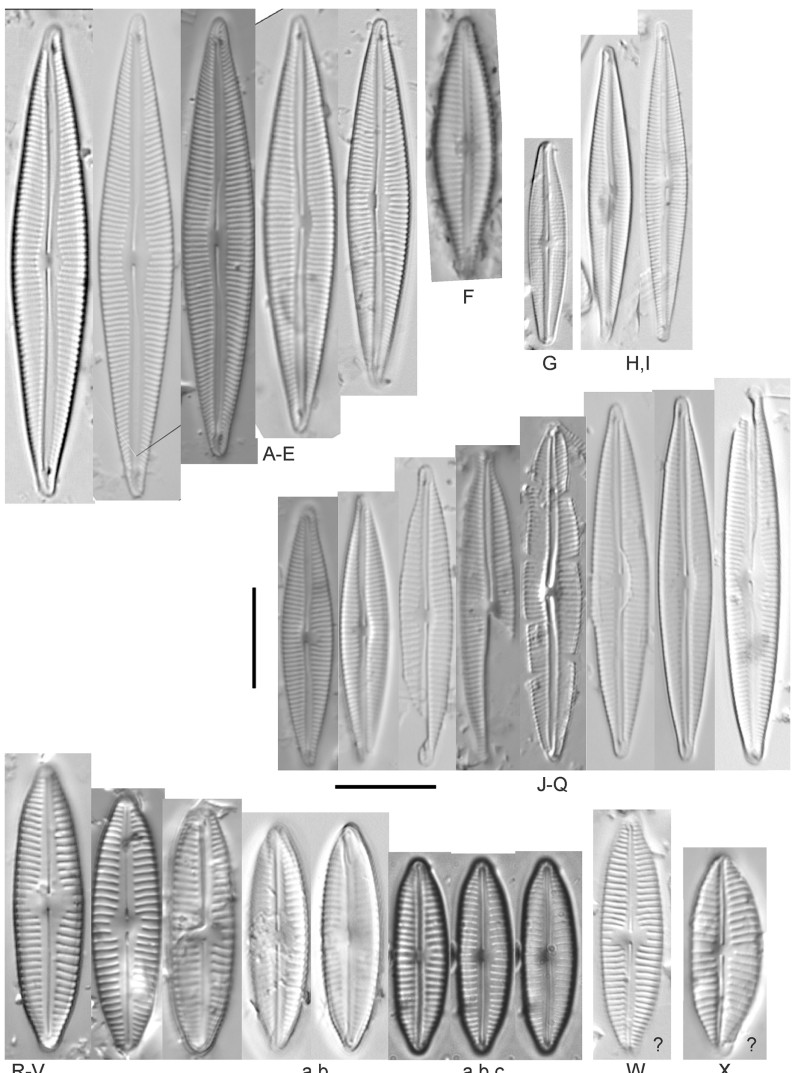

**Figure 16 Diatom light microscope images.** (A–E) *Encyonopsis montana* L.L.Bahls [423, 247, 423, 247, 29]; (F) *Encyonopsis* cf. *montana* L.L.Bahls [698]; (G) *Encyonopsis* cf. *cesatii* (Rabenhorst) Krammer "Ford River" [281]; (H and I) *Encyonopsis* cf. *cesatii* (Rabenhorst) Krammer "Peterson Creek" [200, 200]; (J–Q) *Encyonopsis cesatii* (Rabenhorst) Krammer [423, 423, 247, 423, 419, 247, 247, 200]; (R–V, W? and X?) *Cymbopleura* cf. *incertiformis* Krammer [247, 420, 411, 213, 247, 273, 461]. Lowercase letters indicate multiple images of the same specimen, and a question mark indicates a specimen with taxonomic uncertainty as described in the text. Square brackets contain sampling locales (Fig. 1) for each specimen. Scale bars: 10 mm.

stria density 11–16/10 µm. The Great Lakes specimen has valve length 16.5 µm; width 5.5 µm; stria density 14/10 µm.

*Cymbopleura incertiformis* Krammer (Fig. 15NNNN)

This specimen matches well with *Krammer*'s *(2003)* description for the species. *Krammer (2003)* gives valve length range 24–60 µm; width range 6.5–8.5 µm; stria density 15–19/10 µm at the central area, up to 22/10 µm at the ends. The Great Lakes specimen has valve length 31.5 µm; width 6.5 µm; stria density 18/10 µm at the central area, 22/10 µm at the ends.

*Cymbopleura* cf. *incertiformis* Krammer (Figs. 16R–16V)

These specimens were originally identified as *Cymbella incerta* (Grunow) Cleve based on the broad depiction of that species by *Krammer & Lange-Bertalot (1986)*. Further scrutiny of the valve shape and comparison with additional specimens for *Cymbopleura incerta* provided by *Krammer (2003)* indicate that no published descriptions adequately characterize these Great Lakes specimens. Comparison can be made with *Cymbopleura frequens* Krammer and its varieties (*Krammer, 2003*), and *Cymbopleura hybrida* (Grunow ex Cleve) Krammer (*Bahls, 2015*), but those species have a coarser striae density and *frequens* has a more rostrate shape in the valve ends. Comparison is also made with *Cymbopleura incertiformis* Krammer (valve length range 29.5–60.7 μm, width range 7.0–9.4 μm, stria density 14–17/10 μm at the valve center and 20–24/10 μm at the ends on the dorsal side; *Bahls, 2014a*), which tends to be longer, sometimes has no central area, and has a noticeably oblique raphe in approximately two-thirds of the axial area; features not observed in Great Lakes specimens. Valves of *Cymbopleura* cf. *incertiformis* are lanceolate, very slightly dorsiventral to symmetric along the long axis, subtly triundulate with blunt, apiculate apices; valve length range 19–28 μm; width range 5–7 μm; stria density 14–18/10 μm. The axial area (sternum) is narrow and linear-lanceolate, narrower at the poles, widening gradually then merging with an irregular to circular or oval central area. The central area comprises a third to half of the valve width. The raphe is very weakly lateral and is filiform at the proximal ends, which are expanded and not visibly tipped toward either side. The distal raphe ends are deflected dorsally, a feature of *Cymbopleura* that is confirmed in specimen Fig. 16Va. Striae are weakly radiate at the poles and central area, approaching parallel elsewhere. Similar specimens were observed from multiple locations around the Great Lakes, supporting the view that this may be an undescribed species. The specimen with relatively high striae density and a narrower axial area (Fig. 16W), and the more dorsiventral specimen (Fig. 16X) are presented as additional variations.

*Cymbopleura kuelbsii* v. *nonfasciata* Krammer (Fig. 15MMMM)

While *Krammer (2003)* presents type specimens from France, this Great Lakes specimen appears to be the same taxon. *Krammer (2003)* gives valve length range 23–37 μm; width range 6–8.5 μm; dorsal stria density 10–13/10 μm at the center, up to 17 at the ends. The Great Lakes specimen has valve length 23 μm; width 7.5 μm; stria density 12/10 μm at the center and ~17/10 μm at the ends.

*Cymbopleura* cf. *laeviformis* Krammer (Fig. 12A)

*Krammer (2003)* gives valve length range 27–46 μm; width range 8.5–10.7 μm; stria density 11–13/10 μm at the central area, and up to 15/10 μm at the ends. The Great Lakes specimen gives valve length 28 μm; width 9 μm; stria density 12/10 μm (dorsal central area), 14/10 μm (ends). Other features such as a lacking central area and slightly radiate striae appear characteristic, but additional specimens are needed to confirm taxonomy.

*Cymbopleura lanceolata* Krammer (Figs. 12C and 12D)

*Krammer (2003)* gives valve length range 30–60 μm; width range 11–14 μm; stria density 10–11/10 μm at the central area, and up to 15/10 μm at the ends. The Great Lakes

specimens give valve length 39 µm; width 11.5 µm; stria density 10/10 µm at the dorsal central area, and up to 13/10 µm at the ends. The protracted apical character further confirms this identification. The ends of the *C.* cf. *lanceolata* specimen (Fig. 12D) do not exhibit the usual protracted, subrostrate character, but otherwise measurements are nearly identical.

*Cymbopleura lata* var. *irrorata* Krammer (Fig. 12G)
The subtle central area and bluntly rounded valve ends characterize this *Cymbopleura* and slightly higher striae density differentiates it from very similar varieties such as *Cymbopleura lata* var. *truncata Krammer (2003)*. *Krammer (2003)* gives valve length range 40–60 µm; width range 17–19 µm; stria density 10–12/10 µm. The Great Lakes specimen gives valve length 49.5 µm; width 19 µm; stria density 10/10 µm at the dorsal central area, and up to 12/10 µm at the ends.

*Cymbopleura naviculiformis* (Auerswald) Krammer (Fig. 14II)
This species can be differentiated from *Cymbopleura amphicephala* by having a more distinct, circular central area and puncta that are discernable under LM. *Krammer (2003)* gives valve length range 26–50 µm; width range 9–13 µm; stria density 12–14/10 µm, up to 18/10 µm at the ends. The Great Lakes specimen has valve length 35 µm; width 10 µm; stria density 13/10 µm, 16/10 µm at the ends.

*Cymbopleura* cf. *rupicola* (Grunow) Krammer (Fig. 12B)
*Krammer (2003)* gives valve length range 20–60 µm; width range 6–11.4 µm; stria density 12–14/10 µm at the central area, and up to 15–16/10 µm at the ends. The Great Lakes specimen gives valve length 29 µm; width 8.5 µm; stria density 13/10 µm (dorsal central area), 15/10 µm (ends). The lack of curvature in the raphe and presence of a small central area prevents certain association with *Krammer*'s *(2003)* depiction of the species.

*Cymbopleura subaequalis* (Grunow) *Krammer, 2003* (Figs. 14V and 14W)
Great Lakes specimens of this species are well depicted by *Krammer (2003)* from European samples. *Krammer (2003)* gives valve length range 20–54 µm; width range 7–9.4 µm; stria density 10–14/10 µm. The Great Lakes specimens give valve length range 29–36 µm; width range 7–8 µm; stria density 13/10 µm.

*Cymbopleura subapiculata Krammer, 2003* (Fig. 12F)
While very similar to *C. subcuspidata*, the narrowly protracted apices differentiate this species. *Krammer (2003)* gives valve length range 49–81 µm; width range 21–27 µm; stria density 8–10/10 µm at the central area, and up to 15/10 µm at the ends. The Great Lakes specimen gives valve length 55 µm; width 21.5 µm; stria density 10/10 µm at the dorsal central area, and up to 13/10 µm at the ends.

*Cymbopleura subcuspidata* Krammer (Figs. 13B–13D)
For this species *Krammer (2003)* gives valve length range 49–100 µm; width range 19–25 µm; stria density 8–11/10 µm at the central area, and up to 15/10 µm at the ends. The Great Lakes specimens give valve length range 45–54 µm; width range 16–18 µm; stria density 11/10 µm

at the dorsal central area, and up to 14/10 μm at the ends. While smaller than previous measurements, all other features indicate this is a correct identification.

*Cymbopleura* sp. 20 (Fig. 12E)
A matching description for this likely *Cymbopleura* could not be found in the literature. The Great Lakes specimen has valve length 40.5 μm; width 10.5 μm; stria density 13/10 μm. The ends are slightly capitate and no central area is present.

*Cymbopleura/Cymbella* (?) sp. (Fig. 14X)
This diatom may be a small valve that poorly represents more typical larger specimens. The specimen has valve length 15.5 μm; width 6 μm; stria density 16/10 μm.

*Cymbopleura/Cymbella* (?) sp. (Garden Bay) (Fig. 14AA)
I present this small, narrow, linear specimen from Garden Bay (Michigan), Lake Michigan with little identifying information as its generic placement is uncertain. *Cymbella*, *Cymbopleura* and *Encyonema* are possible genera. The specimen has valve length 4 μm; width 4 μm; stria density 14/10 μm.

*Delicatophycus* M.J.Wynne
The genus name *Delicata* Krammer was deemed invalid because it is a technical term and was recently replaced by *Delicatophycus* by Wynne (2019).

*Delicatophycus neocaledonicus* M.J.Wynne (Figs. 14B–14Q)
During the GLEI assessment most of these widespread specimens were identified using older protocols as *Cymbella delicatula* Kützing. Subsequent coverage of further taxonomic refinement by Krammer (2003) allowed for new identifications. Based on thorough consideration of species parameters, the majority of former "*Cymbella delicatula*" specimens appear to be *Delicata neocaledonica* Krammer. Despite widespread observation of *Delicata delicatula* (Kützing) Krammer in North America (*e.g.*, Bahls, 2017a, 2017b), including in the Great Lakes (Kingston, 1980), that identification is not appropriate for these specimens. This identification is justified by non-protracted ends and appropriate length in Great Lakes examples, though the size range of the species is slightly expanded. The inflection of the proximal raphe ends is variable, though the same variation is presented by Krammer (2003; their pl. 134:34–42). Krammer (2003) gives valve length range 20–31 μm; width range 4.4–5.2 μm; stria density 18–20/10 μm. The Great Lakes specimens give valve length range 19.5–30 μm; width range 4–5.5 μm; stria density 18–20/10 μm. If correct, this identification appears to be the first observation outside of the type locality of New Caledonica. Expansion of the critical dimensions of *Delicata montana* Bahls (2017c, 2019) is also a possibility, but that species has a lower width limit of 5.1 μm and a slightly lower striae density. Environmental characteristics for this species (Fig. 3B) indicate occurrence in the upper Great Lakes, especially in Lake Huron, and especially in embayments. The TP optimum suggests this is an oligotrophic indicator with a low chloride optimum of 7 μg/L. It is a fair indicator of relatively low levels of anthropogenic stress.

*Delicatophycus* cf. *neocaledonicus* M.J.Wynne (Fig. 14A)
I hesitate to confirm this identification as *D. neocaledonicus* due to a too-wide valve: The Great Lakes specimen is 6.5 μm, while the maximum for this species (*Krammer, 2003*) is purportedly 5.2 μm.

*Delicatophycus* sp. "Bell River 1" (Figs. 14R–14U)
This may be a small, finely striate form of *D. neocaledonicus* observed largely in a single sample from Bell River (near Presque Isle, Lake Huron). Great Lakes specimens give valve length range 19.5–21 μm; width range 4–4.5 μm; stria density 24–26/10 μm.

*Delicatophycus* sp. "Bell River 2" (Fig. 14Y)
This specimen, also from Bell River, has many of the characters of *Delicatophycus neocaledonicus* (shape, size, raphe structure) but has a too-fine striae count (23/10 μm) and a wider axial area than other small, uncertain *Delicatophycus* from Bell River.

*Encyonema* Kützing

*Encyonema brevicapitatum* Krammer (Fig. 17Y)
Though the striae density in our specimen slightly exceeds the range described by *Krammer (1997a)* for "Morphotyp 1," the subrostrate apices and flat ventral outline are characteristic for this species. *Krammer (1997a)* gives valve length range 12–25 μm; width range 4.5–6 μm; stria density 12–17/10 μm. The Great Lakes specimen has valve length 13.5 μm; width 4.5 μm; stria density 18/10 μm.

*Encyonema auerswaldii* Rabenhorst (Figs. 18A–18M)
This species was incorrectly reported as *Encyonema cespitosum* Kützing during original GLEI assessments. The higher puncta (areolae) density in Great Lakes specimens indicates that the Great Lakes contains *E. auerswaldii*, which agrees with observations that this species is common in freshwater systems across the USA (*Lowe, 2015*). In Great Lakes specimens the presence of slightly subrostrate, bent apices and variation in the width of the axial area appear to be transient features. For *Encyonema cespitosum* var. *comensis Krammer (1997a)* reports ventrally bent ends (Table 1), a feature that sometimes appeared in Great Lakes specimens of *E. auerswaldii*. That character appears inconsistent in Krammer's photographic examples, so I do not designate any varieties based on this feature. Careful observation of the puncta density is required to separate this species from similar (but larger, coarsely punctate) species (*e.g.*, *Encyonema kamtschaticum* Krammer, *E. cespitosum* Kützing, *Encyonema sinicum* Krammer). Environmental characteristics for this species (Fig. 3C) indicate occurrence particularly in Lake Ontario, and in nearshore and embayment habitats. The TP optimum is 17 μg/L while the chloride optimum is 12 μg/L. It is a weak indicator of medium levels of anthropogenic stress.

*Encyonema* cf. *auerswaldii* Rabenhorst (Fig. 18N)
This specimen adheres to features of the species except for a higher striae density (18/10 μm) and erratic strial lengths on the ventral portion of the valve.

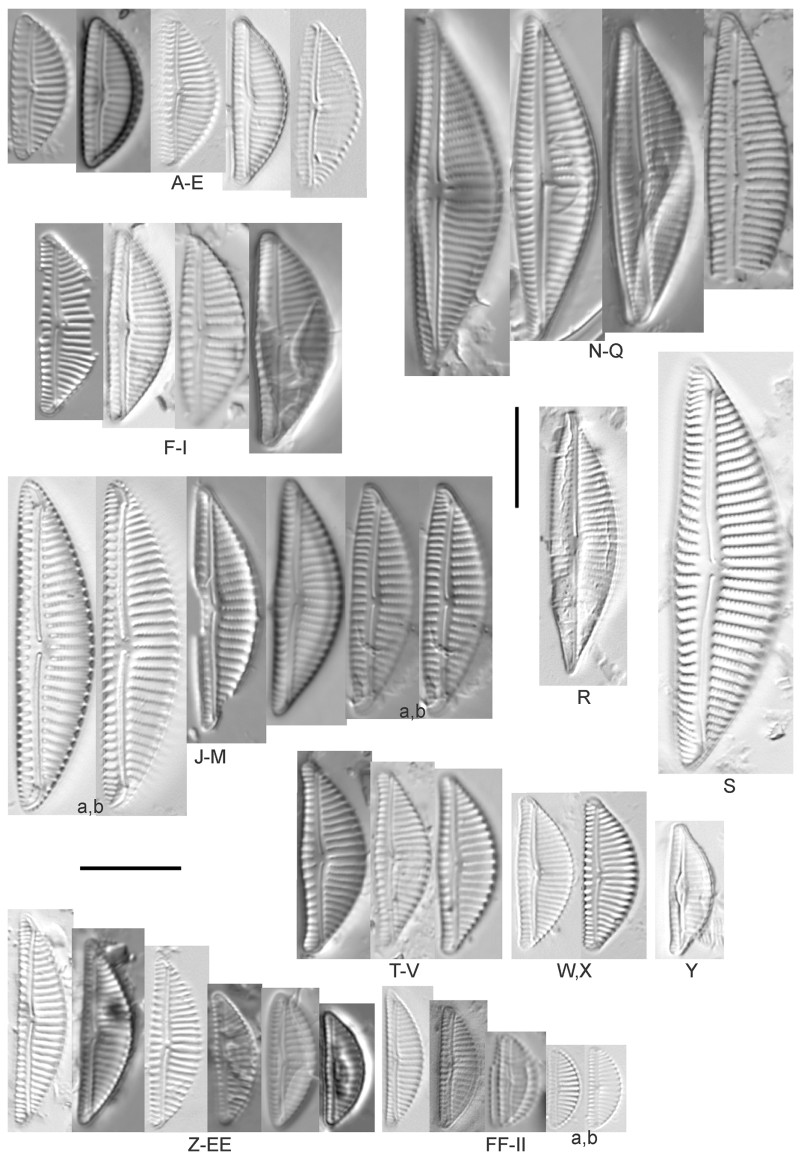

**Figure 17 Diatom light microscope images.** (A–M) *Encyonema silesiacum* (Bleisch in Rabenhorst) D. G.Mann [420, 407, 311, 375, 389, 352, 102, 247, 375, 208, 461, 407, 461]; (N–Q) *Encyonema* cf. *eligense* (Krammer) D.G.Mann [462, 423, 202, 364]; (R) *Encyonema* sp. "Baraga, Lake Superior" [192]; (S) *Encyonema* cf. *silesiacum* (Bleisch in Rabenh.) D.G.Mann [200]; (T–V) *Encyonema lange-bertalotii* Krammer [470, 247, 247]; (W and X) *Encyonema ventricosum* (C.Agardh) Grunow [299, 311]; (Y) *Encyonema brevicapitatum* Krammer [192]; (Z–EE) *Encyonema minutum* (Hilse in Rabenhorst) D.G. Mann [84, 746, 420, 746, 375, 746]; (FF–II) *Encyonema* cf. *minutum* (Hilse in Rabenhorst) D.G.Mann [325, 375, 746, 311]. Lowercase letters indicate multiple images of the same specimen. Square brackets contain sampling locales (Fig. 1) for each specimen. Scale bars: 10 mm.

*Encyonema* cf. *elginense* (Krammer) D.G.Mann (Figs. 17N–17Q)
*Krammer*'s *(1997a)* concept of this species is a fair fit, though Great Lakes specimens are slightly narrower. *Krammer (1997a)* gives valve length range 26–63 µm; width range 10–17 µm; stria density 8–13/10 µm. Great Lakes specimens have valve length range 30–36 µm; width range 8–9 µm; stria density 11–12/10 µm.

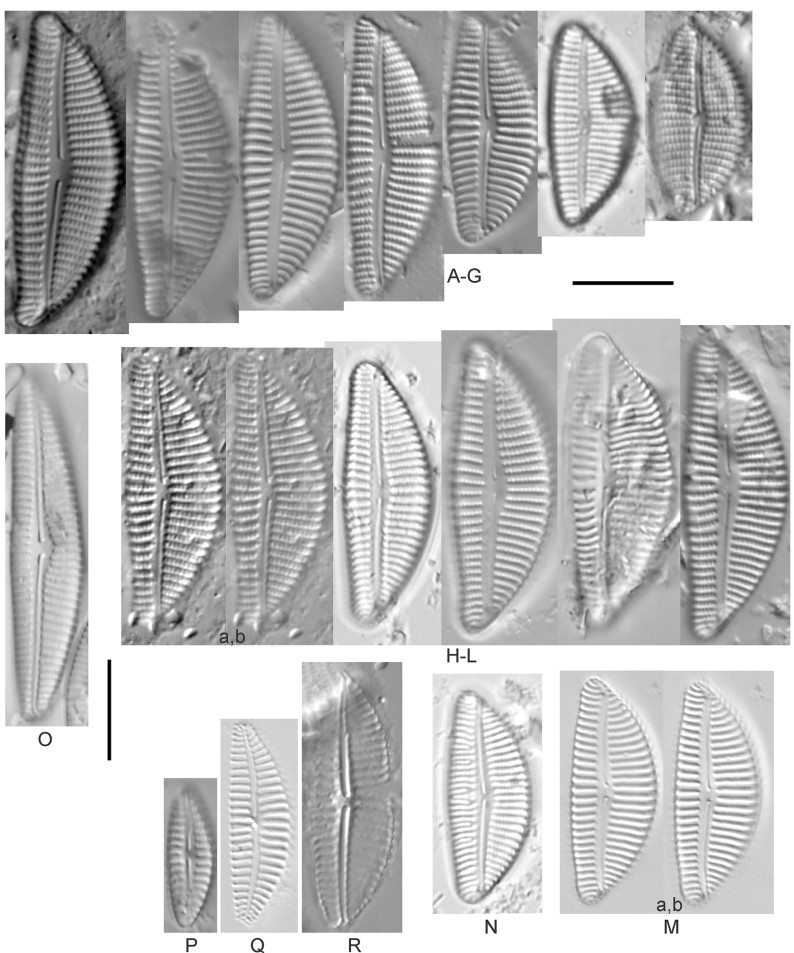

**Figure 18 Diatom light microscope images.** (A–M) *Encyonema auerswaldii* Rabenhorst. [606, 588, 470, 352, 311, 761, 735, 350, 311, 462, 374, 470, 273]; (N) *Encyonema* cf. *auerswaldii* Rabenhorst. [311]; (O) *Encyonema norvegicum* (Grunow) A.Mayer [247]; (P) *Encyonema* sp. "Lake Huron" [423]; (Q) *Encyonema/Cymbella* sp. [420]; (R) *Encyonema* cf. *holmenii* (Foged) Krammer [419]. Lowercase letters indicate multiple images of the same specimen. Square brackets contain sampling locales (Fig. 1) for each specimen. Scale bars: 10 mm.

**Table 1 Critical dimensions and features of *E. auerswaldii* and similar species based on existing literature and new observations from the Great Lakes.**

| | | Length (µm) | Width (µm) | Striae/10 µm | Puncta/ 10 µm | Feature |
|---|---|---|---|---|---|---|
| *Encyonema cespitosum* | *Krammer (1997a)* | 22–57 | 9.8–15 | 9–12 | 17.5–21 | Coarser puncta density |
| *Encyonema cespitosum* var. *comensis* | *Krammer (1997a)* | 22–48 | 10.5–13.4 | 10–11 | 17–20 | Ventral bent ends (not consistent) |
| *Encyonema auerswaldii* | *Krammer (1997a)* | 15–37 | 8–12 | 9–14 | 20–25 | Finer puncta density |
| | *Patrick & Reimer (1975)* as *Cymbella prostrata* var. *auerswaldii* | 15–32 | 8–12 | 10–12, to 14 at ends | 18 | Finer puncta density |
| | *Lowe (2015)* | 22.6–32.9 | 9–10.3 | 10–12, to 15 at ends | ~25 | Finer puncta density |
| | Great Lakes | 19–31.5 | 9–11 | 10–14 | 20–25 | |

*Encyonema* cf. *evergladianum* Krammer (Figs. 15XXX–15KKKK)
As for other attempts to fit specimens into previously published accounts of similar species, striae densities are frequently higher in Great Lakes diatoms (*Reavie & Andresen, 2020*). *Krammer*'s *(1997b)* description of *Encyonema evergladianum* Krammer (recently recommended for transfer to *Encyonopsis* by *Kociolek et al. (2021)*) was determined to be a close fit to these specimens during GLEI assessments, but on further scrutiny they likely represent at least one undescribed species, or greatly increase the permissible striae density for *E. evergladianum*. The shape of Great Lakes valves is a good match for that species, though the presence of a very small central area also differs. Torch Lake, which is connected to Grand Traverse Bay (northeastern Lake Michigan) was noted to lack *E. evergladianum* in favor of a similar but more slender-valved taxon (*Kociolek et al., 2021*), further suggesting that adjacent specimens from the Great Lakes represent a different species. *Krammer (1997b)* gives valve length range 13–28 µm; width range 3.7–5 µm; stria density 21–23/10 µm. Great Lakes specimens have valve length range 13.5–22.5 µm; width range 3.5–4.5 µm; stria density 22–28/10 µm. Environmental characteristics for this species (Fig. 3D) indicate occurrence particularly in lakes Superior and Michigan, and in coastal and protected wetland habitats. The TP optimum is 6 µg/L while the chloride optimum is 5 µg/L. It is a fair indicator of relatively low levels of anthropogenic stress.

*Encyonema* cf. *holmenii* (Foged) Krammer (Fig. 18R)
The curved structure of the raphe suggests this is an *Encyonema* similar to *E. holmenii* (*Krammer, 1997b*), but poor preservation prevents further refinement of taxonomy. The specimen has valve length 26.5 µm; width 8.5 µm; stria density 12/10 µm.

*Encyonema kamtschaticum* Krammer (Figs. 19C and 19D)
While very similar to *E. reimeri*, the higher striae density and ventral inflation are distinctive for this species. This species is not easily distinguished from *Encyonema temperei* Krammer as presented by *Spaulding (2010a)*, but the Great Lakes specimens have slightly greater shortening of the central area stria on the dorsal side of the valve. *Krammer (1997a)* gives valve length range 26–64 µm, width range 10–17 µm, stria density 9–11/10 µm. The Great Lakes specimens have valve length range 30.5–41 µm; width range 14–15 µm; stria density 11–12/10 µm.

*Encyonema leibleinii* (C.Agardh) Silva et al. (Figs. 19F and 19G)
This distinctive diatom was originally identified as *Encyonema prostratum* (Berkeley) Kützing, but more recent examination of type material (*Silva et al., 2013*) led to determination that the name *E. leibleinii* has priority over *E. prostratum*. *Alexson (2014)* gives valve length range 40–65 µm; width range 16–23 µm; stria density 7–10/10 µm. The Great Lakes specimens have valve length range 45–47 µm; width range 15.5–17 µm; stria density 10/10 µm. The two specimens presented have valve outlines that may suggest further taxonomic refinement is possible, but otherwise their critical dimensions and features are similar.

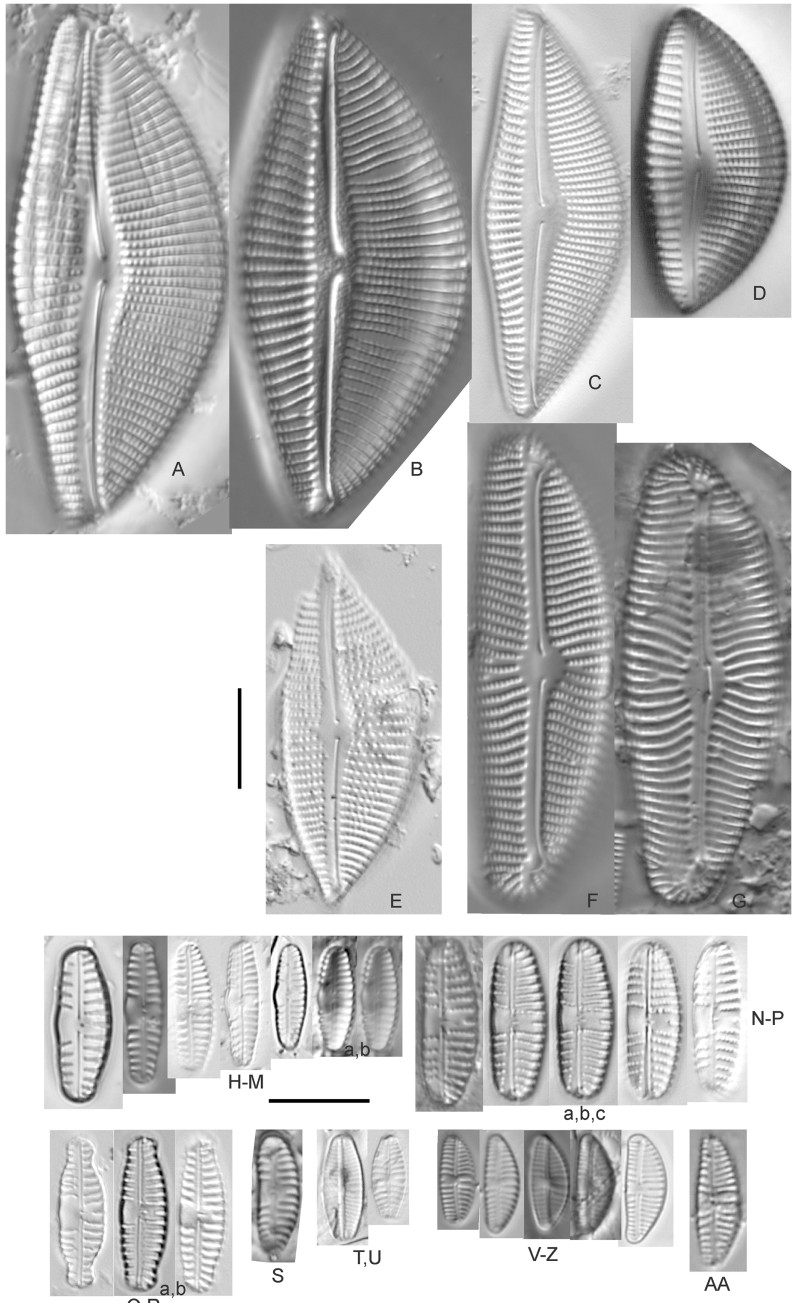

**Figure 19 Diatom light microscope images.** (A and B) *Encyonema reimeri* S.A.Spaulding, J.R.Pool & S. I.Castro [420, 419]; (C and D) *Encyonema kamtschaticum* Krammer [273, 311]; (E) *Encyonema triangulum* (Ehrenberg) Kützing [43]; (F and G) *Encyonema leibleinii* (C.Agardh) Silva et al. [356, 735]; (H–M) *Reimeria sinuata* Kociolek & Stoermer [459, 88, 299, 208, 743, 350]; (N–P) *Reimeria uniseriata* S. E.Sala, J.M.Guerrero & Ferrario [350, 313, 743]; (Q and R) *Reimeria sinuata* f. *antiqua* (Grunow) Kociolek & Stoermer [299, 273]; (S) *Reimeria* sp. (Lake Ontario) [743]; (T and U) *Encyonopsis/Reimeria* (?) [200, 200]; (V–Z) *Encyonema reichardtii* (Krammer) D.G.Mann [311, 407, 743, 735, 88]; (AA) *Encyonema* sp. (Riley's Bay, Lake Michigan) [311]. Lowercase letters indicate multiple images of the same specimen. Square brackets contain sampling locales (Fig. 1) for each specimen. Scale bars: 10 mm.

*Encyonema minutum* (Hilse in Rabenhorst) D.G.Mann (Figs. 17Z–17EE)
This species can be difficult to distinguish from *Encyonema silesiacum*, but under LM the puncta in *Encyonema minutum* are barely or not visible. *Krammer (1997a)* gives valve length range 7–23 μm; width range 4.2–6.9 μm; stria density 15–19/10 μm. Great Lakes specimens have valve length range 11.5–21 μm; width range 4.5–5.5 μm; stria density 14–18/10 μm. Environmental characteristics for this species (Fig. 3E) indicate occurrence particularly in Lake Ontario, and in nearshore habitats. The TP optimum is 15 μg/L while the chloride optimum is 17 μg/L. It is a weak indicator of medium levels of anthropogenic stress.

*Encyonema* cf. *minutum* (Hilse in Rabenhorst) D.G.Mann (Figs. 17FF–17II)
These small specimens are narrower and more finely striate than *E. minutum* as depicted by *Krammer (1997a)*, though these Great Lakes examples may represent a reduction in cell size due to vegetative cell division. Great Lakes specimens have valve length range 8.5–14 μm; width range 3.5–4 μm; stria density 18–22/10 μm.

*Encyonema* cf. *neogracile* Krammer (Fig. 15SSS)
While the shape and structure of the valve face are a good match for *E. neogracile*, the striae density is too high for confident placement. *Krammer (1997a)* gives valve length range 16–50 μm; width range 4.7–6.6 μm; stria density 12–15/10 μm. The Great Lakes specimen has valve length 26 μm; width 5 μm; stria density 18/10 μm.

*Encyonema norvegicum* (Grunow) A.Mayer (Fig. 18O)
This species is well described by *Graeff (2012)*, who gives valve length range 25.0–43.0 μm; width range 7.0–8.0 μm; dorsal stria density 12–13/10 μm at the center, up to 15 at the ends; ventral striae: 13–15/10 μm at the center, up to 17 at the ends. The Great Lakes specimen has valve length 35 μm; width 7 μm; stria density 12/10 μm at the center and ~15/10 μm at the ends.

*Encyonema reichardtii* (Krammer) D.G.Mann (Figs. 19V–19Z)
Mainly found in nordic-alpine areas and northern latitudes, the Pennsylvania specimen *Cymbella brehmii* Hustedt (*Patrick & Reimer, 1975*) has been brought into synonymy with *E. reichardtii* by Mann (*Round, Crawford & Mann, 1990*). Specimens are generally small with convex dorsal and weakly convex ventral sides. Asymmetric Fig. 19Z most likely represents an abnormal valve. Raphe branches are slightly flexed ventrally. The central area is formed by the shortening of the middle striae with the ventral stria being the shortest and the dorsal stria having additional space between it and its neighboring striae. *Krammer (1997b)* provides valve length range 6.7–14.5 μm, width range 3.2–4 μm, stria density 18–22/10 μm and *Patrick & Reimer (1975)* provide valve length range 11–16 μm, width range 4–5 μm, and dorsal stria range 12–14/10 μm, ventral stria range 14–16/10 μm. Great Lakes specimens have valve length range 9.5–13.5 μm, width range 3.5–4.5 μm, stria density 16–21/10 μm. US literature suggests our specimens should have coarser striae density, while in fact they are finer like European examples. *Patrick & Reimer (1975)* emphasize the ventral flex of the raphe (prominent in my specimens) as a character to use

**Table 2** Critical dimensions and features of *E. silesiacum* and similar species based on existing literature and new observations from the Great Lakes.

| | | Length (μm) | Width (μm) | Striae/10 μm | Puncta/10 μm | Feature |
|---|---|---|---|---|---|---|
| *Encyonema silesiacum* | *Patrick & Reimer (1975)*, as *Cymbella minuta* var. *silesiaca* | 18–40 | 7–9 | 11–14 | 26–28 | Apices not rostrate |
| | *Krammer (1997a)* | 14–44 | 5.9–11 | 11–14 | 24–32 | Apices not rostrate |
| | *Bahls (2016c)* | 19.6–48.7 | 6.0–10.8 | 12–14 (17 at apices) | 28–32 | Apices not rostrate |
| | Great Lakes (Figs. 17A–17M) | 14.5–33 | 5.5–8.5 | 11–17 | 30–33 | Apices slightly or not rostrate |
| *Encyonema lange-bertalotii* | *Krammer (1997a)* Morphotype 1 | 19–38 | 7.5–11 | 14–16 | 27–30 | Substrate apices |
| | *Krammer (1997a)* Morphotype 2 | 16–38 | 6.2–8.5 | 14–16 | 27–30 | Substrate apices |
| | *Bahls (2016d)* | 17.6–31.7 | 6.6–8.3 | 14–16 | 30–35 | Substrate apices |
| | Great Lakes (Figs. 17T–17V) | 18–19 | 5.5–6.5 | 13–16 | ~30–35 | Substrate apices |
| *Encyonema ventricosum* | *Krammer (1997a)* Morphotype 3 | 9–21 | 4.5–6.9 | 12–19 | 33–39 | Substrate apices |
| | Great Lakes (Figs. 17W and 17X) | 15.5 | 5.5 | 16–18 | >35 | Substrate apices |

to separate their *Cymbella (Encyonema) brehmii* from the similar *Cymbella (Encyonema) hustedtii*. Environmental characteristics for this species (Fig. 3F) indicate occurrence particularly in lakes Superior and Ontario, and in high-energy habitats. The TP optimum is 11 μg/L while the chloride optimum is 6 μg/L. It is a fairly weak indicator of medium levels of anthropogenic stress.

*Encyonema reimeri* S.A.Spaulding, J.R.Pool & S.I.Castro (Figs. 19A and 19B)
My specimens fit well with features described by *Spaulding (2010b)*, who gives valve length range 30–89 μm, valve width range 16–26.5 μm, stria density 10/10 μm. Great Lakes specimens have valve length range 50–52 μm; valve width range 21–23 μm; stria density 9–10/10 μm.

*Encyonema silesiacum* (Bleisch in Rabenh.) D.G.Mann, *Encyonema lange-bertalotii* Krammer (Figs. 17T–17V), *Encyonema ventricosum* (C.Agardh) Grunow (Figs. 17W and 17X)
*Krammer (1997a)* presents a highly variable concept for *E. silesiacum*. Distinguishing this species from the similarly highly variable *Encyonema elginense* (Krammer) D.G.Mann (*Krammer, 1997a*) is difficult, though *E. elginense* generally lacks a central stigma and North American examples suggest the subtle "shoulder" at the valve ends (*e.g.*, Fig. 17P) may be characteristic (*Bahls, Boynton & Johnston, 2018*). The conspicuousness of the central stigma in *E. silesiacum* is transient in Great Lakes specimens, so its presence is not always useful for identification. This taxon was part of a confusing complex of similar species that were often difficult to distinguish in Great Lakes samples. In Table 2 I compare three taxa that were often included with *E. silesiacum* during sample assessments, and

though challenges remain I have attempted to discern taxonomy of available photographs. Unfortunately, features described in the literature for these species were frustratingly transient, such as a subtle subrostrate character in the apices of *E. silesiacum*, a feature that is more characteristic of *E. lange-bertalotii*. Also, separation of *E. ventricosum* from both of these species is largely based on a higher stria density, but given otherwise similar characteristics with these species I am not convinced this separation is warranted. Further, potential varieties of *E. silesiacum* as presented by *Krammer (1997a*; *distinctepunctata, altensis, lata, Encyonema persilesiacum*) were not easily discerned and may be incorporated into our Great Lakes definition of this species. Environmental characteristics for *E. silesiacum* (Fig. 3G) indicate widespread occurrence across the Great Lakes, but especially in nearshore locations in Lake Superior. The TP optimum is 19 µg/L while the chloride optimum is 8 µg/L. It is a fair indicator of medium levels of anthropogenic stress. Environmental characteristics for *E. ventricosum* (Fig. 3H) indicate it is mainly found in the upper lakes and is a strong indicator of medium levels of anthropogenic stress.

*Encyonema silesiacum* (Bleisch in Rabenh.) D.G.Mann (Figs. 17A–17M)
This is a more typical example of the species as depicted by *Krammer (1997a)*, who gives valve length range 14–48 µm; valve width range 5.9–11 µm; stria density 11–14/10 µm. The Great Lakes specimen has valve length 33 µm; width 8 µm; stria density 10/10 µm.

*Encyonema* cf. *silesiacum* (Bleisch in Rabenhorst) D.G.Mann (Fig. 17S)
Although *Encyonema silesiacum* has comprised a broad definition from a set of largely European examples (*Krammer, 1997a*), this specimen is relatively large and coarsely striate for the species, and it has a clear dorsal inflection of the raphe at the proximal ends. Other similar European concepts include *Encyonema jemtlandicum* var. *venezolana* Krammer and *Encyonema vulgare* Krammer, but additional specimens from the Great Lakes are needed for confirmation. Puncta are also relatively coarse. The Great Lakes specimen has valve length 40.5 µm; width 11.5 µm; stria density 10/10 µm at the central area, 12/10 µm at the ends.

*Encyonema triangulum* (Ehrenberg) Kützing (Fig. 19E)
This distinctive diatom is well matched with previous descriptions of the species from North America. *Bahls (2012)* gives valve length range 37–72 µm; valve width range 16–23 µm; dorsal stria density 8–10/10 µm at the center, 11–12/10 µm at the ends; ventral stria density 9–12/10 µm center, 12–14/10 µm ends. The Great Lakes specimen has valve length 36 µm; width 14 µm; stria density 10–14/10 µm.

*Encyonema* sp. "Baraga, Lake Superior" (Fig. 17R)
While similar in lanceolate shape to our specimens of *E.* cf. *elginense*, this specimen is relatively small with a finer striae density. The Great Lakes specimen has valve length 23 µm; width 7 µm; stria density 16/10 µm.

*Encyonema* sp. "Lake Huron" (Fig. 18P)
This rare, lanceolate and probable *Encyonema* from Lake Huron did not belong to any species I observed in the existing literature, though it may be a very small example in the

asexually declining size series of a species such as *E. norvegicum*, which has a similar stria density.

*Encyonema* sp. (?) (Figs. 15TTT–15VVV)
These specimens are similar to *Encyonema* cf. *evergladianum*, but the inflection of the proximal raphe ends to the ventral side of the valve distinguishes them as possibly *Cymbella* or *Delicata*. The specimens have valve length range 19–23 μm; width range 4–4.5 μm; stria density 22/10 μm.

*Encyonema* sp. 103 (Fig. 15LLLL)
While placed alongside other *Encyonema*, this ambiguous specimen may be *Cymbella* or *Delicata*. The specimen has valve length 15.5 μm; width 4.5 μm; stria density 24/10 μm.

*Encyonema*/*Cymbella* sp. (Fig. 18Q)
The curved raphe separates this specimen from other diatoms in the *E. silesiacum* complex. It is likely a *Cymbella* with similarities to *Cymbella laevis* Naegeli (pl. 191 in *Krammer (2002)*), but the central stria on the ventral side possibly connected to a stigma is distinctive and unique. The specimen has valve length range 20.5 μm; width 6 μm; stria density 11/10 μm at the center and ~15/10 μm at the ends.

*Encyonopsis* Krammer

*Encyonopsis cesatii* (Rabenhorst) Krammer (Figs. 16J–16Q)
These original identifications during the GLEI assessment maintain this species name. While some specimens could be identified as *Encyonopsis cesatiformis* Krammer (*Bahls, 2014b*), the wider valves expected for that species were not observed, so the presence of *E. cesatiformis* in the Great Lakes is questionable. Valves tended to be more lanceolate than the more rhombic *Encyonopsis montana*. *Hofmann, Werum & Lange-Bertalot (2011)* give valve length range 18–60 μm; width range 4–6 μm; stria density 18–22/10 μm. Great Lakes specimens have valve length range 26–38 μm; width range 4.5–6 μm; stria density 17–22/10 μm. Environmental characteristics for this species (Fig. 3I) indicate it occurs in the upper Great Lakes in all coastal habitat types. The TP optimum is 10 μg/L while the chloride optimum is 3 μg/L. It is a good indicator of medium levels of anthropogenic stress.

*Encyonopsis* cf. *cesatii* (Rabenhorst) Krammer "Ford River" (Fig. 16G)
This specimen from near the Ford River outlet (Green Bay, Lake Michigan) generally matched valve parameters for *Encyonopsis cesatii* with the exception of a very high striae density of 28/10 μm.

*Encyonopsis* cf. *cesatii* (Rabenhorst) Krammer "Peterson Creek" (Figs. 16H and 16I)
These narrow valves from near the Peterson Creek outlet (south shore, Lake Superior) largely take on the character of smaller *Encyonopsis cesatii*, but the higher striae density of 26–28/10 μm indicates they may be an undescribed species or variety.

*Encyonopsis descripta* (Hustedt) Krammer (Fig. 15WWW)
This specimen matches well with the species depiction by *Krammer (1997b)*. *Krammer (1997b)* gives valve length range 15–33 μm; width range 4–6.5 μm; stria density 18–21/10 μm. The Great Lakes specimen has valve length 21.5 μm; width 4.5 μm; stria density 21/10 μm.

*Encyonopsis* cf. *krammeri* "very fine" (Figs. 15LL–15VV)
These specimens largely follow depictions of this species by *Hofmann, Werum & Lange-Bertalot (2011)* and *Krammer (1997b)*, and "cf." is applied because of striae densities that exceed the upper limit of 30/10 μm for *Encyonopsis krammeri*. Although *Encyonopsis angusta* Krammer has sufficiently dense striae (32–35/10 μm; *Krammer, 1997b*), the strongly capitate ends present in that species are not consistent in Great Lakes examples. Great Lakes specimens have valve length range 9–17.5 μm; width range 2.8–3.5 μm; stria density 31–36/10 μm. Clearly there remains variation in the prominence of the "shoulders" and capitation at the valve ends, which may warrant further taxonomic splitting. Environmental characteristics for this species (Fig. 3J) indicate it occurs mainly in the upper Great Lakes, especially in riverine wetlands. The TP optimum is 7 μg/L while the chloride optimum is 10 μg/L. It is a weak indicator of relatively low levels of anthropogenic stress.

*Encyonopsis* cf. *krammeri* "very fine and narrow" (Figs. 15NNN–15RRR)
As for *Encyonopsis* cf. *krammeri*, these specimens have higher than previously published striae densities, but they also represent distinctly narrow valves, a character which may specify a unique species. Great Lakes specimens have valve length range 10.5–15 μm; width ~2.5 μm; stria density 31–36/10 μm. Environmental characteristics for this species (Fig. 4A) indicate it occurs across the Great Lakes except in Erie, and in all coastal habitat types. The TP optimum is 15 μg/L while the chloride optimum is 11 μg/L. It is a relatively weak indicator of medium levels of anthropogenic stress.

*Encyonopsis montana* L.L.Bahls (Figs. 16A–16E)
Although specimens were originally identified as *Encyonopsis cesatiformis* Krammer and *Encyonopsis cesatii* (Rabenhorst) Krammer, the strongly radiate striae at the ends distinguished them under this newer name. *Bahls (2013a)* gives valve length range 37–75 μm; width range 7.0–8.8 μm; stria density 19–21/10 μm at the valve center, 21–22/10 μm near the apices. Great Lakes specimens have valve length range 37–47 μm; width range 6-8 μm; stria density 18–21/10 μm at the valve center, ~23/10 μm near the apices.

*Encyonopsis* cf. *montana* L.L.Bahls (Fig. 16F)
This uncertain specimen from a riverine wetland at Buck Pond, Lake Ontario, has a vague similarity to *E. montana* but it is too short (26 μm), narrow (6.5 μm) and coarsely striate (16/10 μm). Strong strial radiation at the ends also seems to be lacking and there may be a central stigma, which is not typical for the genus.

*Encyonopsis subminuta* Krammer & E.Reichardt (Figs. 15YY–15FFF)
This species identification is held for specimens with a striae density of 24/10 μm; low enough to indicate proper association with the species. However, given the difficulties described for the *E. subminuta/krammeri* complex, many specimens in that complex could also be *E. subminuta.* Given overlap in measurements (size, shape, character of the ends and central area, and especially striae density) and transitions in these characters among species, LM assessments of Great Lakes assemblages are challenging. It is likely many of these specimens could be identified as *Encyonopsis minuta* Krammer & E.Reichardt, but transition between these two species makes this difficult. For instance, a comparison of the two species using the associated diatoms.org pages (*Bahls, 2013b*; *Kociolek, 2011a*) states that "*Encyonopsis subminuta* has larger, more elliptic and more symmetric (less dorsiventral) valves"; features that appear to be transient within species and transitional between the two species. Overlap of lengths and striae densities also prevents distinction. Further, despite use of the name during GLEI assessments, the purportedly ubiquitous *Encyonopsis microcephala* is not applied to any Great Lakes specimens due to (a) rare observation of specimens with the required lower striae density, and (b) the inability to distinguish it from *Encyonopsis subminuta* based on published valve characteristics and LM limitations. Given that European concepts are used for these names, it is clear that more detailed, high-resolution studies are needed to better define these species for the Great Lakes. For *Encyonopsis subminuta*, *Hofmann, Werum & Lange-Bertalot (2011)* give valve length range 10–25 μm; width range 3.4–4.5 μm; stria density 12–14/10 μm, up to 23–26/10 μm at the ends. The Great Lakes specimens have valve length range 8–16 μm; width range 3–4 μm; stria density 24/10 μm. Environmental characteristics for this species (Fig. 4B) indicate it occurs mainly in the upper Great Lakes in most habitat types except riverine wetlands. The TP optimum is 8 μg/L while the chloride optimum is 10 μg/L. It is a fair indicator of medium levels of anthropogenic stress.

*Encyonopsis subminuta* "broadly rounded form" (Figs. 15WW and 15XX)
This taxonomic assignment matches that used for *Encyonopsis subminuta*, though these specimens were characterized by broadly rounded ends and a distinct widening of the axial area in the center of the valve. Great Lakes specimens have valve length range 8–16 μm; width range 3–4 μm; stria density 24/10 μm.

*Encyonopsis subminuta/krammeri* (Figs. 15A–15KK)
Inspection of dozens of specimens from many locations throughout the upper (Superior, Huron, Michigan; locations 29–486; Fig. 1) coastal Great Lakes reveals that previous taxonomic accounts for these small *Encyonopsis* are inadequate to consistently identify species. Given frustrating transition in valve characters that are supposed to be unique to each species—rostrate to distinctly capitate ends, dorsiventrality, striae density, small to non-existent central areas—I eventually decided to create an ambiguous complex that clearly requires additional high-resolution scrutiny. For instance, a subset of these specimens (Figs. 15FF–15KK) shows a combination of greater capitation and dorsiventral difference, similar to *Encyonopsis microcephala* (Grunow) Krammer, which may support

separation. Because it is described as a defining character for *Encyonopsis krammeri*, a maximum striae density of 30/10 µm was arbitrarily selected as the cutoff for assignment to this complex. Higher striae densities are not presented in previous literature on these taxa, so another category (*Encyonopsis* cf. *krammeri* "very fine") was created for these Great Lakes-specific forms. A lower cutoff of 25/10 µm was also chosen to separate this complex from likely specimens of *Encyonopsis subminuta*. Though largely European concepts (*Krammer, 1997b*) were used in naming, special consideration is clearly needed for the Great Lakes. Great Lakes specimens in this complex have valve length range 8–20.5 µm; width range 3–4 µm; stria density 25–30/10 µm. Environmental characteristics for this species (Fig. 4C) indicate it occurs mainly in the upper Great Lakes in all coastal habitat types except protected wetlands. The TP optimum is 15 µg/L while the chloride optimum is 9 µg/L. It is a weak indicator of medium levels of anthropogenic stress.

*Encyonopsis thumensis* Krammer (Figs. 15GGG–15MMM)
Though specimens in this taxonomic assignment graded into *Encyonopsis subminuta* and the ambiguous *Encyonopsis subminuta*/*krammeri* complex, they were characterized by a subtle rostrate character at the apices. *Krammer (1997b)* gives valve length range 9.5–18 µm; width range 3.5–4 µm; stria density 23–26/10 µm. The Great Lakes specimens have valve length range 12–17.5 µm; width range 3–3.5 µm; stria density 24–28/10 µm. Using LM, I see no adequate way to distinguish these specimens from *Encyonopsis vandamii* Krammer as presented by *Krammer (1997b)*. As for most of these small Great Lakes *Encyonopsis*, this name should be considered tentative until SEM or genetic comparisons can be made against examples from the type locality, in this case Bavaria.

*Encyonopsis/Reimeria* (?) (Figs. 19T and 19U)
This valve is placed alongside specimens of *Reimeria* due to similarities and uncertainty of the genus assignment. *Stoermer & Yang (1969)* reported this taxon from the Great Lakes as *incertae sedis Cymbella* sp. 11. The specimens illustrated have fine striae, making them outliers based on the reported measurements. The similar *Encyonopsis beheri* (Foged) Krammer & Metzeltin is much larger (*Krammer (1997b)* gives valve length range 19–32 µm, width range 7.4–9.7 µm, stria density 12–15/10 µm). Great Lakes specimen measurements give valve length range 9–12 µm, width range 3.5–4 µm, stria density 20–24/10 µm.

*Encyonema* sp. (Riley's Bay, Lake Michigan) (Fig. 19AA)
This appears to be an unknown *Encyonema* with similarities to *Reimeria*. The construction of the central area is similar to *E. reichardtii* but has an isolated stigma in the center. The raphe is not flexed ventrally as is *E. reichardtii*, though the ventral side is weakly convex. Valve length 13.3 µm, width range 4.4 µm, stria density 17.5/10 µm are from a single specimen and combined with the morphological characters provide the only evidence for a new species.

 

*Reimeria* Kociolek & Stoermer
*Reimeria* observations in the Great Lakes largely match with previously described species, but some detailed descriptions are provided due to ambiguities in previous literature.

*Reimeria sinuata* Kociolek & Stoermer (Figs. 19H–19M)
Valves are linear with ends that are broadly rounded. The central area usual possesses a unilateral swelling on the ventral side with the dorsal side being tumid. The striae are slightly radiate to parallel on the dorsal side and more strongly radiate to parallel on the ventral side. The ends of the ventral side have a small 'blank' area lacking striae and instead filled with the apical porefield (*Schoeman & Archibald, 1979*) that extends to the mantle (*Kociolek & Stoermer, 1987*). The raphe is straight throughout the valve, becoming flexed toward the dorsal side at the ends of the valve, although one specimen exhibited a weakly arched raphe (Fig. 19J). There is a central stigma located just to the dorsal side or between the proximal raphe ends. Figures 19L and 19M have a slightly bilateral symmetry unlike Figs. 19H–19K, so it is possible that Figs. 19L,M represent a different species of *Reimeria*. The critical dimensions of forms identified during this project exhibit expanded variability. *Kociolek & Stoermer (1987)* give length 12–20 µm, breadth 4–5 µm and striae 10–12/10 µm (10–16/10 µm presented by *Potapova (2009)*). These new Great Lakes specimens are length 10.5–20.5 µm, breadth 3.5–6 µm and striae 12–16/10 µm. Environmental characteristics for this species (Fig. 4D) indicate it occurs mainly in Lake Ontario, especially in high-energy environments. The TP optimum is 15 µg/L while the chloride optimum is 12 µg/L. It is a fair indicator of medium levels of anthropogenic stress.

*Reimeria uniseriata* S.E.Sala, J.M.Guerrero & Ferrario (Figs. 19N–19P)
Though these specimens do not show a clear lateral swelling that is typical for the genus, they have coarsely punctate striae as well illustrated for this species by *Levkov & Ector (2010)* who provide valve length range 15–40, width range 4–9, stria density 7–14/10 µm. Great Lakes specimens are length 15.5–20.5 µm, breadth 5.5–6 µm and striae 10–12/10 µm.

*Reimeria sinuata* f. *antiqua* (Grunow) Kociolek & Stoermer (Figs. 19Q and 19R)
Though *Krammer & Lange-Bertalot (1986)* do not recognize the form *antiqua*, stating that the circumscription of varieties by outline is not feasible (translated), *Kociolek & Stoermer (1987)* recognize *R. sinuata* f. *antiqua* and describe the unique, fine structure with SEM. The ends are distinctly capitate. The striae are radiate throughout the valve, appear as solid lines in the LM but are doubly punctate in the SEM. A central pore (stigma) is located off-center externally but penetrating the valve on an angle, thus when focusing through the valve the stigma appears like a dash. The raphe is straight curving toward the dorsal side at the distal end. The critical measurements for the specimens observed fall well within those given by *Patrick & Reimer (1975*, as *Cymbella sinuata* f. *antiqua* (Grunow) Reimer): valve length range 11–40 µm, width range 3.5–9 µm, stria density 9–14/10 µm. Great Lakes specimens are valve length range 15–15.5 µm, width 4.5 µm, stria density 12–14/10 µm.

*Reimeria* sp. (Lake Ontario) (Fig. 19S)
This appears to be an undescribed form of *Reimeria* and is possibly teratogenic. The shape is reniform with the ventral side central area slightly concave. The raphe appears to be straight with the distal ends not resolved. Striae puncta are not resolved with the LM and are perpendicular to the raphe at the center, becoming radial at the ends. The central area is somewhat unilateral but similar to other species of *Reimeria*. The isolated stigma is characteristically placed midway between the proximal raphe ends. The critical dimensions are valve length 12.5 µm, width 4 µm, stria density 12/10µm.

*Gomphonema* Ehrenberg
*Gomphonema* and similar taxa are represented in Figs. 20–25. Though not detected in any samples analyzed as part of the GLEI project, the well-known gomphonemoid diatom *Didymosphenia geminata* (Lyngbye) M.Schmidt is well presented by other authors (*Stoermer, Yu-Zao & Ladewski, 1986* (Lake Superior north shore)). While this mat-forming species is often abundant, its presence in tributaries well upstream from the GLEI sediment and epiphyton samples likely prevented its detection in coastal samples.

*Gomphonema acuminatum* Ehrenberg (Fig. 20M)
As depicted by *Thomas et al. (2009)*, *G. acuminatum* has a bulge in the wider portion of the valve that is wider than that for the central area. *Hofmann, Werum & Lange-Bertalot (2011)* give valve length range 17.5-–57 µm, width range 6.7–10.8 µm, stria density 9–15/10 µm. The Great Lakes specimen is valve length 35.5 µm, width 10 µm, stria density 12/10 µm. Environmental characteristics for this species (Fig. 4E) indicate it occurs across the Great Lakes except in Lake Erie and was not observed in coastal and protected wetlands. The TP optimum reflects eutrophic conditions at 34 µg/L while the chloride optimum is 33 µg/L. It is a weak indicator of medium levels of anthropogenic stress.

*Gomphonema* cf. *acuminatum* Ehrenberg (Fig. 20N)
Identified as *Gomphonema acuminatum* "v10" during initial assessments, this specimen is part of a complex of similar taxa including *Gomphonema interpositum* E.Reichardt and *Gomphonema coronatum* Ehrenberg, which have tapered poles in the wider part of the valve. The pinch between valve bulges is not as expressed in this specimen when compared to *G. acuminatum* presented by others, though one example provided by *Hofmann, Werum & Lange-Bertalot (2011*; their Figs. 93: 9) is a reasonable fit. While also similar to *Gomphonema brebissonii* Kützing (*Kociolek, 2011b*), my specimen has a too-high width-to-length ratio for that species. For *G. acuminatum Hofmann, Werum & Lange-Bertalot (2011)* give valve length range 17.5–57 µm, width range 6.7–10.8 µm, stria density 9–15/10 µm. The Great Lakes specimen has valve length 26 µm, width 9.5 µm, stria density 12/10 µm.

*Gomphonema amoenum* Lange-Bertalot (Fig. 22E)
Though the poles of specimens shown by *Hofmann, Werum & Lange-Bertalot (2011)* are more strongly tapered to a point, this is a good match for this Great Lakes diatom. *Hofmann, Werum & Lange-Bertalot (2011)* give valve length range 30–65 µm, width range

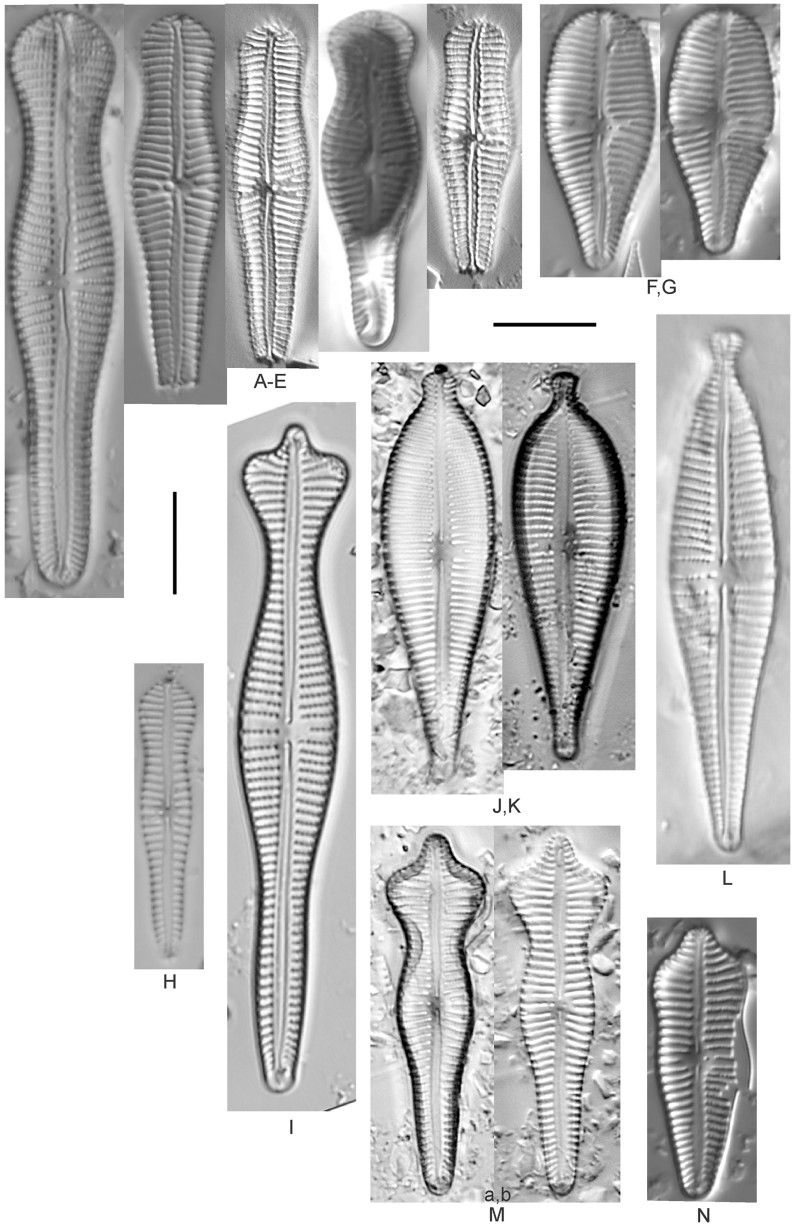

**Figure 20 Diatom light microscope images.** (A–E) *Gomphonema truncatum* Ehrenberg [325, 420, 420, 375, 420]; (F and G) *Gomphonema italicum* Kützing [470, 470]; (H) *Gomphonema angusticephalum* E. Reichardt & Lange-Bertalot [109]; (I) *Gomphonema coronatum* Ehrenberg [247]; (J and K) *Gomphonema sphaerophorum* Ehrenberg [68, 68]; (L) *Gomphonema pseudosphaerophorum* Kobayasi [290]; (M) *Gomphonema acuminatum* Ehrenberg [68]; (N) *Gomphonema* cf. *acuminatum* Ehrenberg [470]. Lowercase letters indicate multiple images of the same specimen. Square brackets contain sampling locales (Fig. 1) for each specimen. Scale bars: 10 mm.

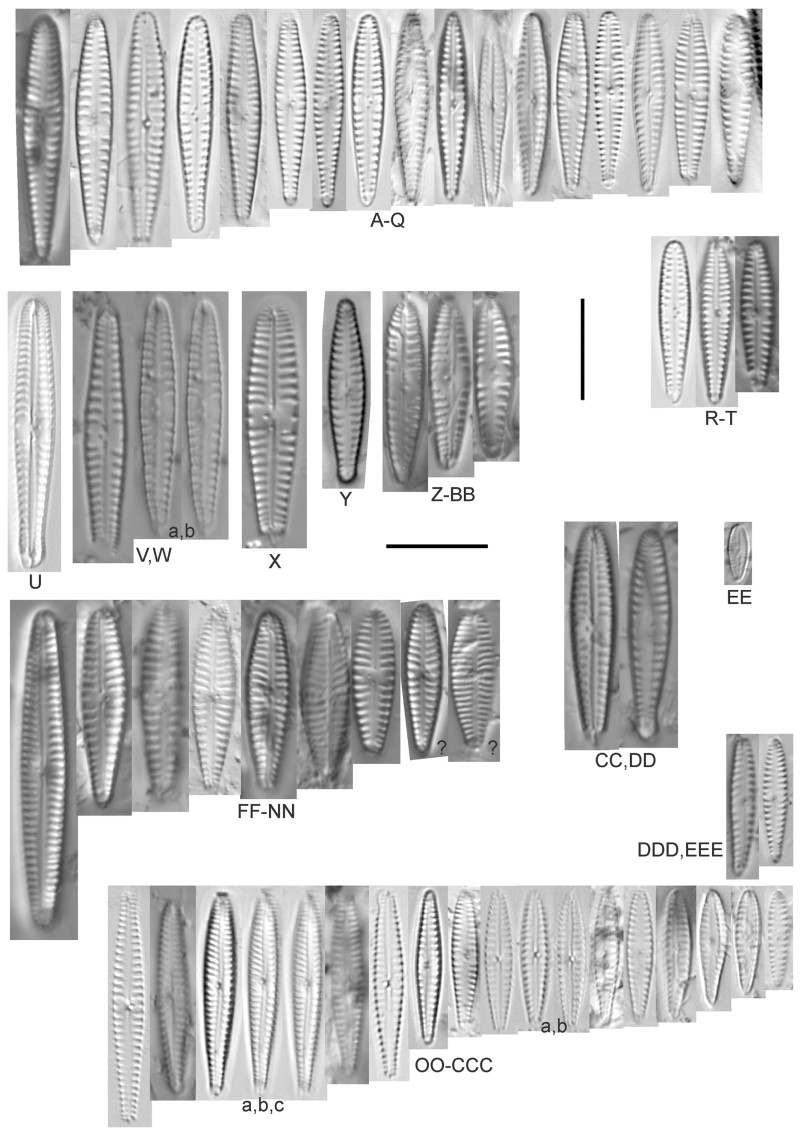

**Figure 21 Diatom light microscope images.** (A–T) *Gomphonema* cf. *pumilum* (Grunow) E.Reichardt & Lange-Bertalot/cf. *pygmaeoides* You & Kociolek/*parapygmaeum* Jüttner & Kociolek [746, 375, 311, 102, 375, 375, 389, 102, 375, 311, 375, 375, 375, 375, 375, 389, 375, 102, 311, 486]; (U) *Gomphonema* sp. (Big Sable) [389]; (V and W) *Gomphonema* cf. *kobayasii* Kociolek & Kingston [364, 364]; (X) *Gomphonema* cf. *rexlowei* Liu & Kociolek [407]; (Y) *Gomphonema* sp. (Grand Lake outlet, Lake Huron) [417]; (Z–BB) *Gomphonema kobayasii* Kociolek & Kingston [364, 364, 364]; (CC and DD) *Gomphonema* sp. (Dunes State Park) [364, 364]; (EE) *Gomphosphenia fontinalis* Lange-Bertalot, Ector & Werum [68]; (FF–NN) *Gomphonema minutum* (C.Agardh) C.Agardh [746, 470, 364, 68, 746, 606, 470, 746, 470]; (OO–EEE) *Gomphonema* cf. *pseudotenellum* Lange-Bertalot [420, 746, 311, 746, 420, 311, 375, 389, 311, 105, 375, 375, 375, 375, 375, 364, 104]. Lowercase letters indicate multiple images of the same specimen, and a question mark indicates a specimen with taxonomic uncertainty as described in the text. Square brackets contain sampling locales (Fig. 1) for each specimen. Scale bars: 10 mm.

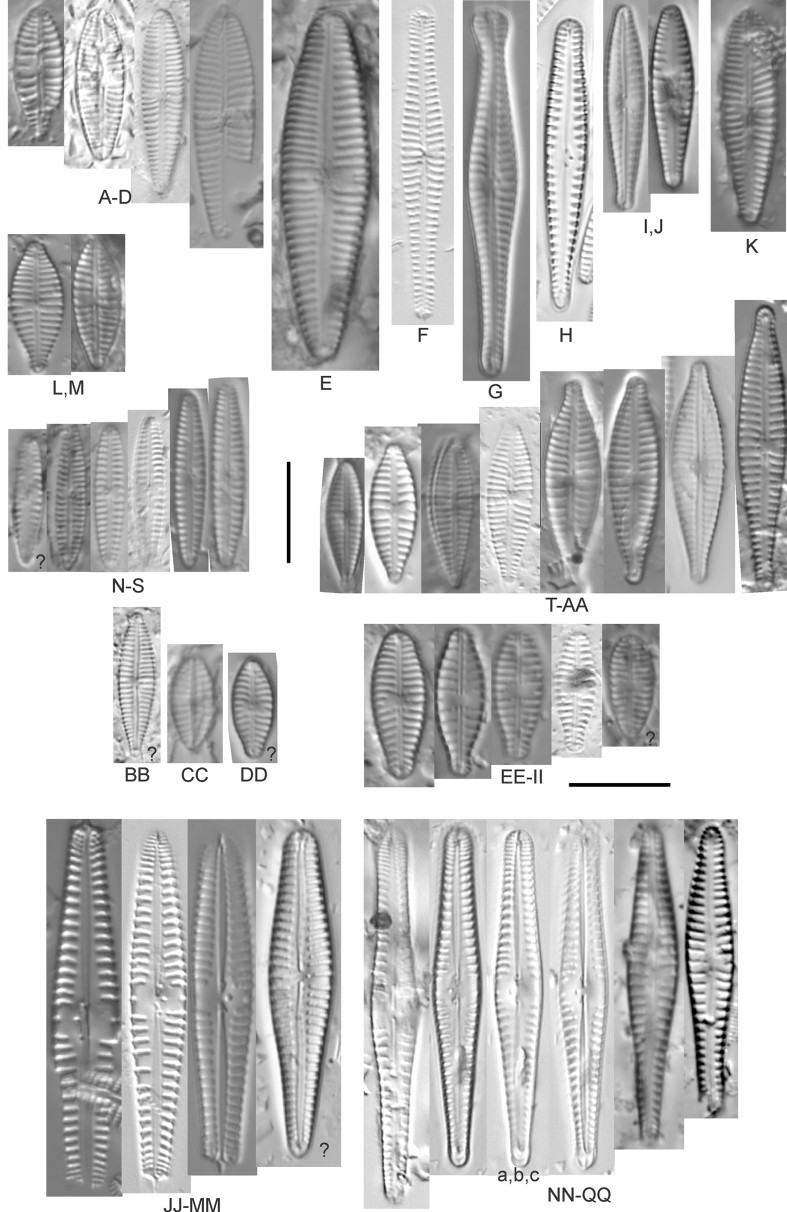

**Figure 22  Diatom light microscope images.** (A–D) *Gomphonema* cf. *parvulum* (Kützing) Kützing [719, 105, 299, 325]; (E) *Gomphonema amoenum* Lange-Bertalot [610]; (F) *Gomphonema* cf. *subtile* Ehrenberg [420]; (G) *Gomphonema subtile* v12 [423]; (H) *Gomphonema* cf. *minusculum* Cleve-Euler [102]; (I and J) *Gomphonema cuneolus* E.Reichardt [425, 425]; (K) *Gomphonema* cf. *clavatulum* E.Reichardt [568]; (L and M) *Gomphonema parvulum* f. *saprophilum* Lange-Bertalot & E.Reichardt/*Gomphonema himalayaense* Jüttner [470, 719]; (N–S) *Gomphonema* cf. *micropumilum* E.Reichardt [746, 743, 746, 68, 202, 746]; (T–BB) *Gomphonema parvulum* (Kützing) Kützing [743, 369, 743, 68, 389, 375, 247, 748, 105]; (CC) *Gomphonema* cf. *parvulum* (Mona Lake) [378]; (DD) *Gomphonema parvulum* (Kützing) Kützing [746]; (EE–II) *Gomphonella calcifuga* (Lange-Bertalot & E.Reichardt) Tuji [735, 281, 281, 68, 743]; (JJ–LL) *Gomphonema intricatum* Kützing (wider) [420, 420, 420]; (MM–QQ) *Gomphonema intricatum* Kützing [84, 43, 29, 610, 364]. Lowercase letters indicate multiple images of the same specimen, and a question mark indicates a specimen with taxonomic uncertainty as described in the text. Square brackets contain sampling locales (Fig. 1) for each specimen. Scale bars: 10 mm.

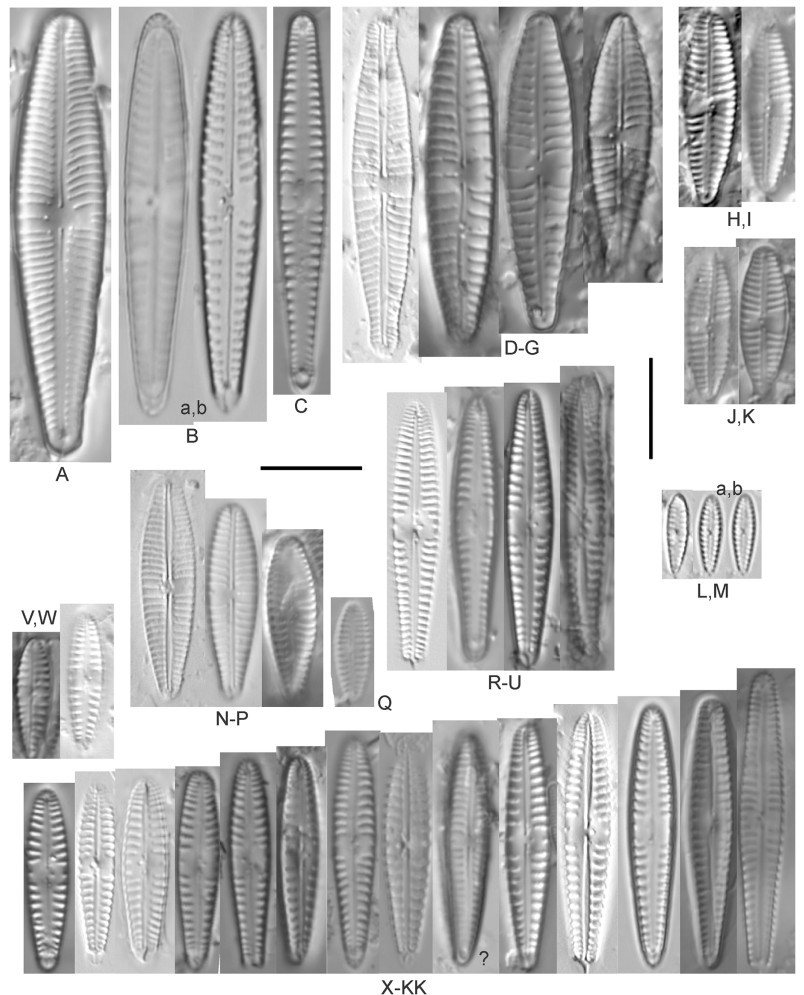

**Figure 23 Diatom light microscope images.** (A) *Gomphonema* cf. *angustatum* (Kützing) Rabenhorst [369]; (B) *Gomphonema* cf. *paludosum* E.Reichardt [299]; (C) *Gomphonema* "v16" [375]; (D–G) *Gomphonema micropus* Kützing [84, 588, 365, 581]; (H–J) *Gomphonema tergestinum* (Grunow) Fricke [350, 350, 350]; (K) *Gomphonema tergestinum* (Grunow) Fricke (? finer) [350]; (L and M) *Gomphonema* sp. (Riley's Bay) [311, 311]; (N–P) *Gomphonema innocens* E.Reichardt [325, 247, 470]; (Q) *Gomphonema* cf. *innocens* E.Reichardt [746]; (R–U) *Gomphonema minusculum* Cleve-Euler [420, 364, 364, 364]; (V–KK) *Gomphonema parapygmaeum* Jüttner & Kociolek [743, 420, 446, 420, 419, 364, 364, 446, 364, 420, 364, 446, 420, 247, 375, 407]. Lowercase letters indicate multiple images of the same specimen, and a question mark indicates a specimen with taxonomic uncertainty as described in the text. Square brackets contain sampling locales (Fig. 1) for each specimen. Scale bars: 10 mm.

9–16 μm, striae 10–11/10 μm. The Great Lakes specimen has valve length 34.5 μm, width 9 μm, stria density 11/10 μm.

*Gomphonema* cf. *angustatum* (Kützing) Rabenhorst (Fig. 23A)

With no obvious rostration at the ends, this specimen looks like some variations of the species as presented by *Hofmann, Werum & Lange-Bertalot (2011)*. For *G. angustatum Hofmann, Werum & Lange-Bertalot (2011)* give valve length range 16–48 μm, width range 5.3–6.7 μm, stria density 10–14/10 μm. *Patrick & Reimer (1975)* give valve length range

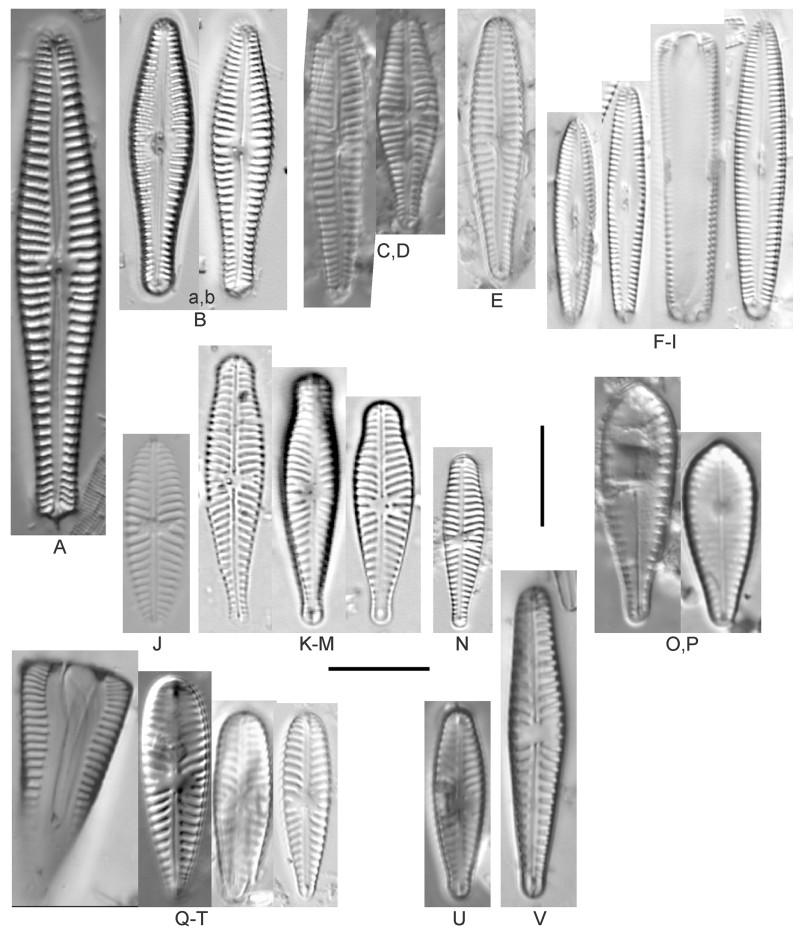

**Figure 24 Diatom light microscope images.** (A and B) *Gomphonema submehleri* Kociolek & Stoermer [420, 29]; (C and D) *Gomphonema* cf. *subclavatum* Grunow [735, 719]; (E) *Gomphonema leptocampum* Kociolek & Stoermer [375]; (F–I) *Gomphonema* cf. *caperatum* Ponader & Potapova [281, 281, 281, 281]; (J) *Gomphonella* sp. (Howard, Wisconsin) [304]; (K–M) *Gomphosinica capitata* Kociolek, You & Wang [51, 51, 51]; (N) *Gomphosinica* cf. *capitata* Kociolek, You & Wang [29]; (O,P) *Gomphosphenia* (cf.?) *grovei* (M.Schmidt) Lange-Bertalot [365, 369]; (Q–T) *Gomphonella olivacea* (Hornemann) Rabenhorst [225, 470, 369, 281]; (U) *Gomphonema* (*Gomphonella?*) *olivaceoides* var. *hutchinsoniana* R.M.Patrick [746]; (V) *Gomphonella olivacea* (Hornemann) Rabenhorst var. 1 UMD [225]. Lowercase letters indicate multiple images of the same specimen. Square brackets contain sampling locales (Fig. 1) for each specimen. Scale bars: 10 mm.

12–45 µm, width range 5–9 µm, stria density 9–13/10 µm, though their depiction has distinct tapering at the ends. Further, LM examples in the literature do not exhibit visible areolae in the striae, unlike my specimen. The Great Lakes specimen is valve length 45 µm, width 9 µm, stria density 13/10 µm. Environmental characteristics for this species (Fig. 4F) indicate it occurs across the Great Lakes except for Lake Erie and in lower-energy wetland environments. The TP optimum reflects eutrophic environments at 54 µg/L while the chloride optimum is 27 µg/L. It is a relatively weak indicator of high levels of anthropogenic stress.

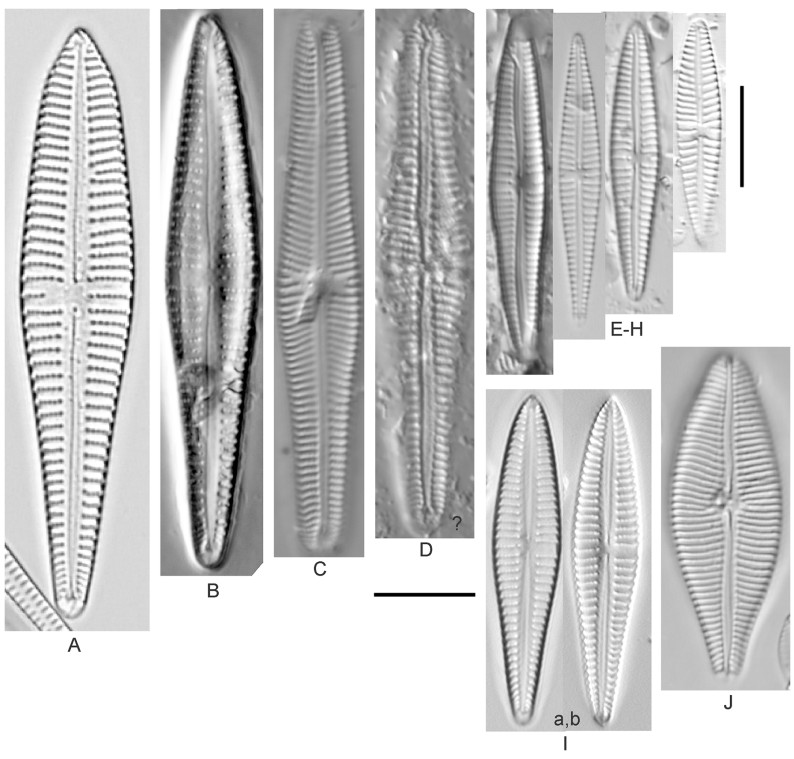

**Figure 25 Diatom light microscope images.** (A) *Gomphonema* cf. *insigne* Gregory [104]; (B) *Gomphonema* cf. *gracile* Ehrenberg [375]; (C and D) *Gomphonema* sp. (Green Bay) [304, 290]; (E–H) *Gomphonema hebridense* Gregory [364, 109, 290, 84]; (I) *Gomphonema* cf. *pseudoaugur* Lange-Bertalot [375]; (J) *Gomphoneis* cf. *eriense* (Grunow) Skvortzov [311]. Lowercase letters indicate multiple images of the same specimen, and a question mark indicates a specimen with taxonomic uncertainty as described in the text. Square brackets contain sampling locales (Fig. 1) for each specimen. Scale bars: 10 mm.

*Gomphonema angusticephalum* E.Reichardt & Lange-Bertalot (Fig. 20H)

Referred to as *Gomphonema interpositum* var. 1 during initial assessments, this specimen is similar to specimens presented by *Bahls, Boynton & Johnston (2018)* from western Canada, which they call "*G. sp.* (cf. *G. interpositum*) (cf. *G. montanum* PH)," though my specimen does not have their deflected proximal raphe ends and has a slightly narrower southern pole. It also has similarities to "*Gomphonema* sp. (cf. *G. capitatum*)" (*Bahls, Boynton & Johnston, 2018*), but the central area formed by shortened striae is smaller in my specimen. A much better identification is *Gomphonema angusticephalum* as identified from Ireland by *Reichardt (1999)*. For *G. angusticephalum Reichardt (1999)* gives valve length range 19.5–49 μm, width range 4–6.4 μm, stria density 10–14/10 μm. The Great Lakes specimen is valve length 28.5 μm, width 5.5 μm, stria density 13/10 μm.

*Gomphonema* cf. *caperatum* Ponader & Potapova (Figs. 24F–24I)

Originally identified as *Gomphonema* sp. 1 during GLEI assessments, this diatom looks like *G. caperatum* (*Ponader et al., 2017*) but has a finer striae density. Also, the stigma was not always visible in our specimens. For *G. caperatum Bishop (2017)* gives valve length range 22.4–31.9 μm, width range 4.4–5.2 μm, stria density 14–16/10 μm. The Great Lakes

specimens have valve length range 21–30 µm, width range 4.5–6 µm, stria density 18–20/ 10 µm.

*Gomphonema* cf. *clavatulum* E.Reichardt (Fig. 22K)
While similar to the species described by *Reichardt (1999)*, this name is attributed due to fair adherence to known valve features and the presence of rounded ends (lack of capitation present in many similar *Gomphonema* species). *Reichardt (1999)* gives valve length range 17.8–38.6 µm, width range 4.7–5.7 µm, stria density 10.5–14/10 µm. The Great Lakes specimen is valve length 21.5 µm, width 5.5 µm, stria density 14-15/10 µm. In the Great Lakes specimen striae density is slightly higher and the ratio of length to width appears to be lower than that in Reichardt's examples (their pl. 25: 1–13).

*Gomphonema coronatum* Ehrenberg (Fig. 20I)
This species is largely distinguished by being longer than *G. acuminatum* and having similar width of the central and upper bulges (*Thomas et al., 2009*; *Hofmann, Werum & Lange-Bertalot, 2011*). *Hofmann, Werum & Lange-Bertalot (2011)* give valve length range 42–108 µm, width range 8.5–14 µm, stria density 8–12/10 µm. The Great Lakes specimen is valve length 65 µm, width 10 µm, stria density 10/10 µm.

*Gomphonema cuneolus* E.Reichardt (Figs. 22I and 22J)
Though I believe this is the first observation of this species in North America, it matches well with that described by *Reichardt (1997)* from Germany. While similar in many ways to *Gomphonema consector* M.H.Hohn & Hellerman, these specimens are too wide. For *G. cuneolus Reichardt (1997)* gives valve length range 11.5–27 µm, width range 3.7–4.5 µm, stria density 10–13(16)/10 µm. The Great Lakes specimen has valve length range 18–20 µm, width 4 µm, stria density 13/10 µm.

*Gomphonema* cf. *gracile* Ehrenberg (Fig. 25B)
While slightly wider, this matches well with *Karthick & Kociolek*'s *(2012)* description. This species cannot be distinguished from *Gomphonema spiculoides* H.P.Gandhi (*Karthick & Kociolek, 2012*) without a closer look at helictoglossa using SEM. Further, this is not easily separable from *Gomphonema affine* Kützing (*Reichardt, 1999*; valve length range 36–88 µm, width range 9–13.6 µm, stria density 8–12/10 µm). For *G. gracile Karthick & Kociolek (2012)* give valve length range 24.5–43.5 µm, width range 6.7–8.4 µm, striae parallel at the center and slightly radiate toward the poles, 11–16/10 µm. *Reichardt*'s *(2015)* thorough reevaluation of *G. gracile* lectotypes indicates that this may be more appropriately named *Gomphonema naviculoides* W.Smith, but additional specimens are needed. Further, critical measurements are a good fit for *Gomphonema kezlyae Pardhi et al. (2020)*; I cannot adequately separate my specimens from theirs. The Great Lakes specimen has valve length 54.5 µm, width 9.5 µm, stria density 11/10 µm.

*Gomphonema hebridense* Gregory (Figs. 25E–25H)
*Hofmann, Werum & Lange-Bertalot (2011)* summarize two species, *G. hebridense* and *Gomphonema auritum* A.Braun, that may represent these specimens from the Great Lakes. Assignment to *G. hebridense* is largely due to the better fit with published striae density for

the species. For *G. hebridense Hofmann, Werum & Lange-Bertalot (2011)* give valve length range 20–70 μm, width range 4.5–8 μm, stria density 12–18/10 μm. Great Lakes specimens have valve length range 22.5–35 μm, width range 4.5–6 μm, stria density 14–16/10 μm. Environmental characteristics for this species (Fig. 4G) indicate it occurs across the Great Lakes in all coastal habitat types. The TP optimum reflects eutrophic conditions at 47 μg/L while the chloride optimum is 15 μg/L. It is a weak indicator of medium levels of anthropogenic stress.

*Gomphonema innocens* E.Reichardt (Figs. 23N–23P)
While similar to *G. parvulum* in shape and asymmetric central area, the wider, subtly capitate ends are a good match with examples of this species provided by *Reichardt (1999)*. *Reichardt (1999)* gives valve length range 9–31 μm, width range 5.2–7 μm, stria density 14–18/10 μm, while Great Lakes specimens are valve length range 16–22 μm, width range 5–5.5 μm, stria density 14–15/10 μm.

*Gomphonema* cf. *innocens* E.Reichardt (Fig. 23Q)
Due to its small size, a lack of visible features makes identification difficult. The striae density and arrangement around the central area suggest it may be a valve of *G. innocens*, but it is narrower than the range published by *Hofmann, Werum & Lange-Bertalot (2011)*. The Great Lakes specimen is valve length 9 μm, width 3.5 μm, stria density 18/10 μm.

*Gomphonema* cf. *insigne* Gregory (Fig. 25A)
Valve features largely match that for *G. insigne* as depicted by *Reichardt (1999)*, with the exception that the Great Lakes specimen has a distinctly subrostrate, triangular head-pole, which is sometimes observed in the more finely striate *Gomphonema insigniforme* E.Reichardt & Lange-Bertalot. For *G. insigne Reichardt (1999)* gives valve length range 20–84 μm, width range 8.6–12.8 μm, stria density 6.5–10/10 μm. The Great Lakes specimens has valve length 58 μm, width 11.5 μm, stria density 10/10 μm.

*Gomphonema intricatum* Kützing (Figs. 22MM–22QQ)
This is similar to *Gomphonema vibrio* Ehrenberg as presented by *Jüttner et al. (2018)* and *Hofmann, Werum & Lange-Bertalot (2011)*, but striae are finer than for *G. vibrio*. This taxon is likely the same as *G. intricatum* as identified from Lake Superior by *Kociolek & Stoermer (1991)*, who indicate valve length range 19–65 μm, width range 4–9 μm, stria density 10–15/10 μm. The Great lakes specimens have valve length range 28–37 μm, width range 5–7 μm, stria density 10–13/10 μm. As noted in the plate, there may be a wider form of this species, and the central area was not always as large as depicted in previous literature. The specimen marked with a question mark is uncertain.

*Gomphonema intricatum* Kützing (wider) (Figs. 22JJ–22LL)
As previously identified from Lake Superior by *Kociolek & Stoermer (1991)*, the large, square central area, hooked proximal raphe ends and linear-clavate shape define this species. Though still ambiguously associated with other species in the *G. pumilum* complex, this appears to be a correct assignment for the Great Lakes. *Kociolek & Stoermer (1991)* give valve length range 19–65 μm, width range 4–9 μm, stria density 10–13/10 μm

in the middle and as high as 20 at the poles. Great Lakes specimens are valve length range 33–40 µm, width range 5.5–7 µm, stria density 10–12/10 µm in the middle.

*Gomphonema italicum* Kützing (Figs. 20F and 20G)
These distinctive, clavate valves with broadly rounded head poles are a good match with the species as described by *Reichardt (2001)*. Reichardt gives valve length range 19–53.5 µm, width range 9.3–14.0 µm stria density 10–16/10 µm. My Great Lakes specimens are valve length range 24–27 µm, width range 10.5–11 µm, stria density 13/10 µm. Aside from slightly finer striae densities in my specimens, I see little way of separating this species from *Gomphonema laticollum* E.Reichardt as characterized from Lake Baikal by *Kulikovskiy et al. (2015)*.

*Gomphonema kobayasii* Kociolek & Kingston (Figs. 21Z–21BB)
These specimens match well with this species, particularly as depicted by *Kociolek (2011c)*. They give valve length range 10–30 µm, width range 3.5–5 µm, stria density 9–12/10 µm. The Great Lakes specimens have valve length range 16–19 µm, width 4 µm, stria density 12–13/10 µm. Environmental characteristics for this species (combined with *G.* cf. *kobayasii*) are provided in Fig. 4H. It was observed in Lake Michigan in high energy and protected wetland habitats. The TP optimum is 8 µg/L while the chloride optimum is 68 µg/L. It is a weak indicator of medium levels of anthropogenic stress.

*Gomphonema* cf. *kobayasii* Kociolek & Kingston (Figs. 21V and 21W)
Though initially identified as *Gomphonema* cf. *bavaricum*, Great Lakes specimens are narrower than that described by *Reichardt & Lange-Bertalot (1991)* for that species. Our narrower valves suggested similarity to *G. kobayasii* as described by *Kociolek & Kingston (1999)* from United States rivers, but those valves have a more lanceolate shape, whereas these specimens taper more consistently from the central area to the ends. Otherwise, primary features of Great Lakes specimens (valve length 24 µm, width 4 µm, stria density 10–12/10 µm) adhere to previous descriptions of the species. Environmental characteristics for this species (combined with *G. kobayasii*) are provided in Fig. 4H. This taxon has one of the highest chloride optima, reflecting the high chloride concentrations at Dunes State Park (site 364, Lake Michigan) where several specimens were observed.

*Gomphonema leptocampum* Kociolek & Stoermer (Fig. 24E)
Occasionally misidentified as *Gomphonema olivaceum* var. *calcarum* (Cleve) Van Heurck, which has no central stigma, this species was originally identified from an epilithic sample from Lake Superior (*Kociolek & Stoermer, 1991*). *Kociolek & Stoermer (1991)* give valve length range 23–53 µm, width range 6–8 µm, stria density 12–16/10 µm. The specimen has valve length 26.5 µm, width 6 µm, stria density 15/10 µm.

*Gomphonema* cf. *micropumilum* E.Reichardt (Figs. 22N–22S)
Largely identified as *G. pumilum* during original assessments, these specimens have a striae density that is too high for that species. They match some of the valve features presented by *Reichardt (1997)*, but the nearly parallel-sided valve shape, lack of even subtle capitation, and sometimes higher striae density suggest that this is an undescribed species.

For *G. micropumilum Reichardt (1997)* gives valve length range 10–22 µm, width 3.5 µm, stria density 11.5–14–18/10 µm. Great Lakes specimens are valve length range 13–18 µm, width 3.5 µm, stria density 16–20/10 µm. One specimen (Fig. 22N) has a wider axial area and may be a different, undescribed species. Environmental characteristics for this species (Fig. 4I) indicate it occurred mainly in nearshore collections from Lake Ontario. The TP optimum is 16 µg/L while the chloride optimum is 18 µg/L. It is a weak indicator of medium levels of anthropogenic stress.

*Gomphonema micropus* Kützing (Figs. 23D–23G)
These specimens largely match those depicted by *Reichardt (1999)* for *G. micropus.* Though similar, *G. micropus* does not have the more parallel sides of *Gomphonema sarcophagus* W.Gregory. My observations are a more coarsely striate version of the species, but still within Reichardt's published range. Though type material originated from Europe, *Reichardt (1999)* indicates occasional observation of this species in North America. *Reichardt (1999)* gives valve length range 19–44 µm, width range 6.3–9 µm, stria density 11–14/10 µm (as low as 8 in the center and 16 at the ends). Great Lakes specimens are valve length range 25.5–33 µm, width range 6–7 µm, stria density 8–13/10 µm. Environmental characteristics for this species (Fig. 4J) indicate it occurs mainly in lakes Huron and Michigan, especially in embayments. The TP optimum is 16 µg/L while the chloride optimum is 18 µg/L. It is a weak indicator of medium levels of anthropogenic stress.

*Gomphonema minusculum* Cleve-Euler (Figs. 23R–23U)
Considered synonymous with *G. superiorensis* from Lake Superior (*Kociolek & Stoermer, 1991*), *Reichardt (1997)* gives valve length range 14.0–32.7 µm, width range 2.8–4.6 µm, stria density 12–16/10 µm. Given the prevalence of similar taxa throughout the Great Lakes it is debatable whether *G. parapygmaeum* and *G. minusculum* are different species. Instead they may represent subtle valve shape variations of a single taxon.

*Gomphonema* cf. *minusculum* Cleve-Euler (Fig. 22H)
Major valve features (shape, length, width, striae density) match that for *G. minusculum* as detailed by *Reichardt (1997)*, but the regular arrangement of shortened striae around the central area in the Great Lakes specimen (unlike the abruptly shortened stria on each side of the central area in *G. minusculum*) prevents full assignment to that species. For *G. minusculum Reichardt (1997)* gives valve length range 14–32.7 µm, width range 2.4–4.6 µm, stria density 12–16/10 µm. The Great lakes specimen has valve length 29 µm, width 4.5 µm, stria density 14/10 µm.

*Gomphonema minutum* (C.Agardh) C.Agardh (Figs. 21FF–21NN)
The specific taxonomic characteristics of this species are difficult to confirm due to wide variations in length/width ratio and striae density among Great Lakes specimens, but these observations match European examples depicted by *Hofmann, Werum & Lange-Bertalot (2011)*, including the slight deflection in the proximal raphe ends to one side of the valve. Several Great Lakes *Gomphonema* species in the same plate are similar and may grade into this taxon, but it is generally separated by its striae density. For *G. minutum Hofmann,*

*Werum & Lange-Bertalot (2011)* give valve length range 10–35 µm, width range 4–8 µm, stria density 8–18/10 µm. Great Lakes specimens have valve length range 15–31.5 µm, width range 3.5–5 µm, stria density 12–17/10 µm. The specimen noted with a question mark is uncertain due to its small central area and relatively high ratio of width to length. Environmental characteristics for this species (Fig. 5A) indicate it occurs in all coastal habitats across the Great Lakes, especially in lakes Michigan and Ontario. The TP optimum is 19 µg/L while the chloride optimum is 35 µg/L. It is a weak indicator of medium levels of anthropogenic stress.

*Gomphonema* cf. *paludosum* E.Reichardt (Fig. 23B)
LM-visible areolae, striae density and elongate shape make this similar to *G. paludosum* as depicted by *Reichardt (1999)*; valve length range 17.5–59.5 µm, width range 5.3–7.1 µm, stria density 8.5–12/10 µm. However, the open central area with distinctly shortened striae on both sides differentiates this specimen; valve length 40 µm, width 6.5 µm, stria density 9/10 µm.

*Gomphonema parapygmaeum* Jüttner & Kociolek (Figs. 23V–23KK)
These lanceolate valves fit the species as depicted by *Jüttner et al. (2018)*, who give valve length range 13.5–25.0 µm, width range 3.0–4.0 µm, stria density 12–16/10 µm. Several species in the *G. pumilum* complex are easily confused with this. The main feature distinguishing these specimens is the lack of tapering at the ends, which turns an otherwise similar diatom into *G. minusculum* as depicted by *Reichardt (1997)* from Germany. Using LM I see no reliable way to distinguish this species from *G. pumilum* var. *rigidum* E. Reichardt & Lange-Bertalot as depicted by *Reichardt (1997)*, which has valve length range 12–36 µm, width range 3–5.3 µm, stria density 11.5–14/10 µm. Reichardt indicates it may be distinguishable by its rectangular central area, but this feature appears to be transient. Great Lakes specimens are valve length range 12–29 µm, width range 3.5–5 µm, stria density 12–14/10 µm. The specimen marked by a question mark has a higher striae density that stands out among other specimens, but it is still within the published range. Environmental characteristics for this species (Fig. 5E) indicate it occurs mainly in lakes Superior and Ontario in a variety of coastal habitats. The TP optimum reflects oligotrophic conditions at 3 µg/L while the chloride optimum is 5 µg/L. It is a weak indicator of medium levels of anthropogenic stress.

*Gomphonema parvulum* (Kützing) Kützing (Figs. 22T–22BB and 22DD)
Several Great Lakes specimens fit within the broad *G. parvulum* complex. Narrower, less lanceolate forms matched well with *Hofmann, Werum & Lange-Bertalot*'s *(2011)* depiction of the nominate form, though some of my observations were slightly narrower. *Hofmann, Werum & Lange-Bertalot (2011)*, give valve length range 10–36 µm, width range 5–8 µm, stria density 7–20/10 µm, a wide range of striae density that captures several Great Lakes specimens. Great Lakes specimens are valve length range 10–28 µm, width range 4–6 µm, stria density 12–18/10 µm, with one specimen (Fig. 22BB) noted to be as high as 20/10 µm (marked with a question mark). Another 8-µm-long specimen (Fig. 22DD) is below published lengths and is placed next to *G. calcifuga* for comparison. Environmental

characteristics for this species (Fig. 5G) indicate it occurs across the Great Lakes in all coastal habitats. The TP optimum reflects eutrophic conditions at 47 μg/L while the chloride optimum is 15 μg/L. It is a weak indicator of medium levels of anthropogenic stress.

*Gomphonema* cf. *parvulum* (Kützing) Kützing (Figs. 22A–22D)
These valves with less prominent capitation and may represent an undescribed species, but they are otherwise similar to *G. parvulum*.

*Gomphonema* cf. *parvulum* (Mona Lake) (Fig. 22CC)
Such a small specimen is difficult to identify, as it may be at the end of a diminution series and is missing the diagnostic features of larger valves. It may be a very small example of *G. parvulum*. Valve length 9.5 μm, width 4.5 μm, stria density 16/10 μm.

*Gomphonema parvulum* f. *saprophilum* Lange-Bertalot & E.Reichardt/*Gomphonema himalayaense* Jüttner (Figs. 22L and 22M)
For *G. himalayaense* *Jüttner et al. (2018)* give valve length range 9.5–29.0 μm, width range 5.0–7.0 μm, stria density 14–16/10 μm. Great Lakes specimens are valve length 13 μm, width 5 μm, stria density 16/10 μm. Using LM there appears to be no way to distinguish smaller valves of *G. parvulum* from *G. himalayaense*, so these specimens from the Great Lakes remain ambiguous. Environmental characteristics for this species (Fig. 5F) indicate it occurs largely in high-energy habitats from Lake Ontario. In concord with a name indicating tolerance of polluted environments (*Hofmann, Werum & Lange-Bertalot, 2011*), *G. parvulum* f. *saprophilum* has one of the highest phosphorus optima (114 μg/L) for the Great Lakes diatoms. It is a weak indicator of impacted conditions.

*Gomphonema* cf. *pseudoaugur* Lange-Bertalot (Fig. 25I)
Though previously identified from the Great Lakes, our specimen is smaller and has a more finely pointed end on the wider part of the valve compared to specimens of *G. pseudoaugur* presented by *Kociolek & Stoermer (1991*; their Figs. 71–74), suggesting I may have something different. *Kociolek & Stoermer (1991)* give valve length range 40–76 μm, width range 10–12 μm, stria density 10–12/10 μm and distinctly punctate. The Great Lakes specimen has valve length 32.5 μm, width 6.5 μm, stria density 13/10 μm.

*Gomphonema pseudosphaerophorum* Kobayasi (Fig. 20L)
*Gomphonema sphaerophorum* Ehrenberg as identified from The Great Lakes by *Kociolek & Stoermer (1991)* appears to include *G. pseudosphaerophorum* within that species, and I now recognize two unique species. *Kociolek & Stoermer (1991)* show one specimen (their Fig. 44) that appears to be transitional into *G. sphaerophorum*. Future observation of additional specimens may indicate a need for further taxonomic splitting. *Polaskey & Vaccarino (2016)* give valve length range 41.9–50.2 μm, width range 8.1–9.4 μm, stria density 9–14/10 μm. The Great Lakes specimen has valve length 51 μm, width 10 μm, stria density 10/10 μm.

*Gomphonema* cf. *pseudotenellum* Lange-Bertalot (Figs. 21OO–21EEE)
Unlike *G. parapygmaeum*, the valve outline tapers gradually in some specimens from the center to the apices, though this is not clearly observed in all Great Lakes specimens. The 'cf.' qualifier is further used because LM images of *G. pseudotenellum* as depicted by *Hofmann, Werum & Lange-Bertalot (2011)* have a barely visible central stria on each side of the central area, whereas in Great Lakes specimens this feature is more clearly apparent. This is not easily distinguishable from smaller specimens of *Gomphonema minusculum* Cleve-Euler (*Hofmann, Werum & Lange-Bertalot, 2011*; *Jüttner et al., 2018*), and they may be conspecific. Many of these may be narrow, fine specimens of a greater complex. For *G. pseudotenellum* *Krammer & Lange-Bertalot (1986)* give valve length range 10–28 μm, width range 2.5–4 μm, stria density 15–22/10 μm. The Great lakes specimens have valve length range 9.5–23 μm, width range 2.5–3.5 μm, stria density 16–20/10 μm.

*Gomphonema* cf. *pumilum* (Grunow) E.Reichardt & Lange-Bertalot/cf. *pygmaeoides* You & Kociolek/*parapygmaeum* Jüttner & Kociolek (Figs. 21A–21T)
Many of these specimens were identified as *Gomphonema* "sp2" during initial GLEI assessments. Without the benefit of SEM, identifying species members of the *G. pumilum* complex is challenging, and given their initial descriptions from largely European and Great Britain locales, whether the same species exist in North America is disputable. Specimens with striae densities that exceed those for the nominate form may be considered *Gomphonema pumilum* var. *elegans* E.Reichardt & Lange-Bertalot (*Reichardt, 1997*; his pl. 1: 1–6, pl. 2: 1–29, pl. 4: 20–23) but a gradient of striae densities is apparent, preventing clear distinction of varieties. While otherwise similar, striae densities greater than 16/10 μm associate specimens with *Gomphonema pygmaeoides*. Some specimens appear to be aligned with *Krammer & Lange-Bertalot (1986)* examples of *Gomphonema angustum* C. Agardh (*e.g.*, their pl. 164). Smaller specimens look like *G. parapygmaeum*, but there is no clear way to diagnostically separate them from larger specimens. *Hofmann, Werum & Lange-Bertalot (2011)* demonstrate another very similar diatom *Gomphonema elegantissimum* E.Reichardt & Lange-Bertalot, which consistently has proximal raphe ends that are slightly deflected to one side of the valve, a feature that appears to occur in many Great Lakes examples. *You, Wang & Kociolek (2015)* originally identified similar specimens as unknown forms like *G. parvulum* and *G. pumilum*. Great Lakes specimens are also more lanceolate than *Gomphonema pseudotenellum* Lange-Bertalot. These somewhat match specimens identified as *Gomphonema pygmaeum* Kociolek & Stoermer from the Great Lakes (*Kociolek & Stoermer, 1991*; holotype from Lake Superior presented), but they do not have the more parallel sides of *G. pygmaeum* as later depicted by *Kociolek (2015)*. Overall, Great Lakes specimens have valve length range 15–25 μm, width range 3–4 μm, stria density 12–18/10 μm. These diatoms are part of the *G. pumilum* complex and should be subjected to greater scrutiny using SEM. Environmental characteristics for this species (Fig. 5D) indicate it occurs across the Great Lakes in embayments. The TP optimum is 16 μg/L while the chloride optimum is 18 μg/L. It is a weak indicator of medium levels of anthropogenic stress.

*Gomphonema* cf. *rexlowei* Liu & Kociolek (Fig. 21X)
Despite many articles resolving similar *Gomphonema* species, Great Lakes specimens such as this do not fit with known species. This is similar to that shown by *Liu, Kociolek & Wang (2013)*, but both striae at the central area are shortened, instead of just one for *G. rexlowei*. This was called *Gomphonema minutum* "v10" during assessment due to similarities in valve shape and strial density, but *G. minutum* has a less prominent central area formed by shortened striae. Another similar species is *Gomphonema juettnerii Karthick et al. (2015)*, but that species has a larger axial area and more evenly distributed striae around the central area. For *G. rexlowei Liu, Kociolek & Wang (2013)* give valve length range 15–35 μm, width range 4.6–6.2 μm, stria density 8–14/10 μm. The Great lakes specimen has valve length 24.5 μm, width 4.5 μm, stria density 13/10 μm.

*Gomphonema sphaerophorum* Ehrenberg (Figs. 20J and 20K)
This generally wider species is similar to *G. pseudosphaerophorum* but has a more asymmetrically inflated "shoulder" end. Specimens were sometimes mistaken for *Gomphonema augur* Ehrenberg but that species does not have an inflated headpole like Great Lakes specimens, although *Thomas et al. (2009)* present examples of what they consider *G. augur* that are not distinguishable from my specimens based on LM assessment. *Polaskey & Bishop (2016)* give valve length range 36.7–55.9 μm, width range 12.8–13.2 μm, stria density 12–16/10 μm. The Great Lakes specimen has valve length range 38–40 μm, width 12 μm, stria density 14/10 μm.

*Gomphonema* cf. *subclavatum* Grunow (Figs. 24C and 24D)
The valve shape is similar to that described for *G. subclavatum* by *Patrick & Reimer (1975)*, but the size is too small and is slightly more finely striate. For *G. subclavatum* Patrick & Reimer give valve length range 35–70 μm, width range 8–10 μm, stria density 9–13/10 μm. The specimens have valve length range 21–27 μm, width 6 μm, stria density 14/10 μm. It also has similarities to *Gomphonema mexicanum* Grunow as depicted by *Reichardt (1997)* but lacks the clearly punctate striae of that species. Further, the two specimens presented for this taxon have different valve shapes, indicating further taxonomic division may be possible.

*Gomphonema submehleri* Kociolek & Stoermer (Figs. 24A and 24B)
While similar to *G. leptocampum*, *G. submehleri* has a wider axial area as in specimen Fig. 24B, a characteristic that is well illustrated by *Kociolek & Stoermer (1991)*, who collected the holotype from a Lake Superior epilithic sample. Specimen Fig. 24A does not carry the wide axial area through the central area, which may reflect a variation within the species. *Kociolek & Stoermer (1991)* give valve length range 25–55 μm, width range 7–11 μm, stria density 8–14/10 μm. The specimens have valve length range 27.5–49 μm, width range 7–8.5 μm, stria density 11–13/10 μm. Environmental characteristics for this species (Fig. 5H) indicate it occurs mainly in embayments and protected wetland habitats in Lake Ontario. The TP optimum reflects eutrophic conditions at 54 μg/L while the chloride optimum is 27 μg/L. It is a weak indicator of medium levels of anthropogenic stress.

*Gomphonema* cf. *subtile* Ehrenberg (Fig. 22F)

The shape of this specimen suggests this identity but the very small central area prevents certain assignment to the species. For *G. subtile* Krammer & Lange-Bertalot give valve length range 24–50 μm, width range 3.5–8 μm, stria density 10–14/10 μm. The Great lakes specimen has valve length 31 μm, width 4.5 μm, stria density 14/10 μm.

*Gomphonema subtile* v12 (Fig. 22G)

Despite some thorough scrutiny of the *G. subtile* complex (*Glushchenko, Kulikovskiy & Kociolek, 2017*), this specimen remains inconclusive and was given a tentative variety. The Great lakes specimen has valve length 36.5 μm, width 5 μm, stria density 13/10 μm.

*Gomphonema tergestinum* (Grunow) Fricke (Figs. 23H–23K)

For *G. tergestinum* *Hofmann, Werum & Lange-Bertalot (2011)* give valve length range 10–32.5 μm, width range 4.7–7 μm, stria density 10–16/10 μm. Great Lakes specimens have valve length range 15–19 μm, width 5 μm, stria density 14–16/10 μm. One specimen (Fig. 23K) stands out as having finer striae than the rest.

*Gomphonema truncatum* Ehrenberg (Figs. 20A–20E)

This species has been previously identified from The Great Lakes by *Kociolek & Stoermer (1991)*. For the Great Lakes they give valve length range 40–75 μm, width range 9–11 μm, and for the species overall *Patrick & Reimer (1975)* give stria density 10–12/10 μm. My Great Lakes specimens are valve length range 25–56 μm, width range 8–10.5 μm, stria density 10–13/10 μm. My observations increase the size range for this species in the Great Lakes. Environmental characteristics for this species (Fig. 5I) indicate it occurs across the Great Lakes in all coastal habitat types.

*Gomphonema* sp. (Big Sable) (Fig. 21U)

Superficially it would be easy to place this specimen within the broad complex of *G. pumilum* and its varieties as depicted by *Reichardt (1997)*, but the slightly capitate ends are unique.

*Gomphonema* sp. (Dunes State Park) (Figs. 21CC and 21DD)

While a likely member of the confusing *G. pumilum* complex, these valves from Lake Michigan have a slightly rostrate southern pole, possibly distinguishing it from other species in the complex.

*Gomphonema* sp. (Grand Lake outlet, Lake Huron) (Fig. 21Y)

While associated with the *G. pumilum* complex during sample assessment, this specimen has more distinctly capitate ends as in *G. angustum* (*Hofmann, Werum & Lange-Bertalot, 2011*), but not having the same central area features as that species.

*Gomphonema* sp. (Green Bay) (Figs. 25C and 25D)

With its distinctly convergent striae around the central area and inflation of the central region of the valve, no adequate match for these specimens could be found in the literature. The intact specimen has valve length 53 μm, width 8.5 μm, stria density 13/10 μm.

One specimen from the same sample is poorly preserved but may represent the same species (marked by a question mark).

*Gomphonema* sp. (Riley's Bay) (Figs. 23L and 23M)
These tiny gomphonemoid diatoms from the eastern shore of Green Bay (Lake Michigan) require higher magnification for proper identification, but they may belong to a small member of the *G. pumilum* complex, such as that presented by *Reichardt (1997)*. Specimens have valve length 8 µm, width 2.5 µm, stria density 20/10 µm.

*Gomphonema* "v16" (Fig. 23C)
While this valve from the *G. pumilum* complex is similar to *Gomphonema parapygmaeum* Jüttner & Kociolek, it is much longer than published ranges for that species. The distinctly elongate shape, wide axial area and square central area prevented assignment to any known taxon. The specimen is valve length 38 µm, width 4.5 µm, stria density 10/10 µm.

*Gomphosphenia* Lange-Bertalot

*Gomphosphenia fontinalis* Lange-Bertalot, Ector & Werum (Fig. 21EE)
Though difficult to confirm because of its small size, this specimen appears to match the species described by *Werum & Lange-Bertalot (2004)* from Germany. They give valve length range 5–8 µm, width range 1.5–2.3 µm, stria density 25–32/10 µm. The Great Lakes specimen has valve length 6 µm, width 2 µm, stria density 32/10 µm. A wide axial area is evident, further suggesting this is a correct identification.

*Gomphosphenia* (cf.?) *grovei* (M.Schmidt) Lange-Bertalot (Figs. 24O and 24P)
I was not able to convincingly separate more than one species from this small complex of species similar to *G. grovei*. Specimen Fig. 24P may be *Gomphosphenia lingulatiformis* (Lange-Bertalot & E.Reichardt) Lange-Bertalot based on a higher striae density, though the presence of ghost striae in my specimen is not observed by others (*Kociolek & Bishop, 2017*) and it is too wide. Ghost striae appear to be an occasional feature for *Gomphonema herrmanniana* Palik (assumed to belong to *Gomphosphenia*; identified in material from China; *Kociolek, Jing-Rong & Stoermer, 1988*) but my specimen has a too-dense striae count. Valve width suggests both specimens may be examples of *G. grovei*, but only specimen Fig. 24O meets other characteristics for that species (*Kociolek, 2011d*).
For *G. grovei Kociolek (2011d)* gives valve length range 15–70 µm, width range 8–11 µm, stria density 10–12/10 µm. The Great Lakes specimens have valve length range 19–25 µm, width range 7–8 µm, stria density 11–14/10 µm.

*Gomphonella* Rabenhorst

*Gomphonella calcifuga* (Lange-Bertalot & E.Reichardt) Tuji (Figs. 22EE–22II)
These specimens were identified as *Gomphoneis olivaceum* var. *minutissimum* (Hustedt) Bukhtiyarova during original GLEI assessments, prior to further taxonomic refinement by several authors. Previously examples of the *Gomphonema olivaceoides* Hustedt complex have been recognized from Lake Huron (*Hustedt, 1950*; *Patrick & Reimer, 1975*), though these smaller individuals indicate a fit with *G. calcifuga* as depicted by *Hofmann, Werum &*

*Lange-Bertalot (2011*; as *Gomphonema calcifugum* Lange-Bertalot & E.Reichardt). Aside from their small size, these specimens are also very similar to *Gomphonema olivaceoides* var. *hutchinsoniana Patrick (1971)*. For *G. olivaceoides Patrick & Reimer (1975)* give valve length range 18–35 µm, width range 5–6 µm, stria density 10–14/10 µm. For *G. calcifugum Hofmann, Werum & Lange-Bertalot (2011)* give valve length range 12–20 µm, width range 3.7–5.5 µm, stria density 12–16/10 µm. Great Lakes specimens are valve length range 14–15 µm, width range 5–5.5 µm, stria density 12–16/10 µm. More capitate ends in some specimens (*e.g.*, specimen Fig. 22FF) suggest further taxonomic differentiation may be possible, though this variability is also presented in a size diminution series by *Hofmann, Werum & Lange-Bertalot (2011)*. Also, a very small specimen (Fig. 22II, valve length 10 µm) appears to have the four-stigmoid feature but may have lost the capitate feature of the ends due to being at the end of a diminution series.

*Gomphonella olivacea* (Hornemann) Rabenhorst (Figs. 24Q–24T)
*Jahn et al. (2019)* recently revised the taxonomic position of this species, and their critical features are a good match with Great Lakes specimens. They give valve length range 14.3–42.2 µm, width range 5.5–8.7 µm, stria density 8.0–15/10 µm. The Great Lakes specimens have valve length range 18.5–25 µm, width range 5.5–7 µm, stria density 12–14/10 µm. Environmental characteristics for this species (Fig. 5B) indicate it occurs mainly in the lower Great Lakes in all coastal habitat types. The TP optimum is 33 µg/L while the chloride optimum is 15 µg/L. It is a weak indicator of medium levels of anthropogenic stress.

*Gomphonella olivacea* (Hornemann) Rabenhorst var. 1 UMD (Figs. 24V)
This specimen has most diagnostic characteristics with the exception of the clavate shape that is typical for the species. Instead, this valve has a slightly inflated central area. For *Gomphonema olivaceum* (Hornemann) Brébisson *Hofmann, Werum & Lange-Bertalot (2011)* gives length 12–42 µm, width range 5.5–9 µm, 8–12/10 µm. The specimen has valve length 31 µm, width 6 µm, stria density 10/10 µm.

*Gomphonema* (*Gomphonella?*) *olivaceoides* var. *hutchinsoniana* R.M.Patrick (Fig. 24U)
For this species-variety *LaLiberte (2017)* gives valve length range 18.4–23.4 µm, width range 5.5–6.1 µm, stria density 11.7–14.3/10 µm. The Great Lakes specimen has valve length 19 µm, width 5.5 µm, 14/10 µm. LaLiberte's specimens from Maine have slightly more capitate poles but otherwise are a good match. A transfer to *Gomphonella* seems suitable for this species, though I could not find a formal record of transfer. Environmental characteristics for this species (Fig. 5C) indicate it occurs mainly in lakes Superior and Ontario typically in high-energy habitats. The TP optimum is 14 µg/L while the chloride optimum is 16 µg/L. It is a weak indicator of medium levels of anthropogenic stress.

*Gomphonella* sp. (Howard, Wisconsin) (Fig. 24J)
This single specimen from a coastal wetland at Howard, Wisconsin (Green Bay, Lake Michigan) has features of *G. olivacea* but has very wide, densely packed striae. The Great Lakes specimen has valve length 19 µm, width 6 µm, stria density 13/10 µm.

*Gomphosinica* Kociolek, You, Wang & Liu

*Gomphosinica capitata* Kociolek, You & Wang (Figs. 24K–24M)
This fits *Kociolek & Stoermer*'s *(1991)* depiction of the *Gomphonema geitleri* (Kociolek & Stoermer) *Kociolek et al. (2015)* holotype from a rock scrape sample at Grand Marais, Minnesota, Lake Superior. Under the recent revision to *Gomphosinica* my specimens match the holotype and isotypes of *G. capitata* from Montana, shown by *Kociolek et al. (2015)* who gives valve length range 15–29 μm, width range 5–6 μm, stria density 12–14/10 μm. The Great Lakes specimens have valve length range 22–30 μm, width range 4.5–6 μm, stria density 12–14/10 μm.

*Gomphosinica* cf. *capitata* Kociolek, You & Wang (Fig. 24N)
This specimen is much like *G. capitata* but has a finer striae density of 18/10 μm.

*Gomphoneis* Cleve

*Gomphoneis* cf. *eriense* (Grunow) Skvortzov (Fig. 25J)
Valve shape suggests this is *G. eriense*, an identification that is supported by its previous identification from the Great Lakes by *Kociolek & Stoermer (1988)*. However, this specimen has a higher striae density than that characterized previously. *Kociolek & Stoermer (1988)* give valve length range 25–67 μm, width range 9.5–13 μm, striae 10–13/10 μm. The Great Lakes specimen is valve length 33 μm, width 11.5 μm, striae 15/10 μm.

## ACKNOWLEDGEMENTS

Diatom photography was supported by A. Kireta, N. Andresen, J. Kingston, G. Sgro, J. Johansen and M. Ferguson. K. Kennedy and A. Bellamy helped with graphic design and document production. David Williams and two anonymous reviewers provided valuable comments on a previous draft of the manuscript. This document has not been subjected to the U.S. Environmental Protection Agency's required peer and policy review and therefore does not necessarily reflect the view of the Agency, and no official endorsement should be inferred.

### Funding

This research was supported by a grant from the U.S. Environmental Protection Agency's Science to Achieve Results (STAR) Estuarine and Great Lakes (EaGLe) program through funding to the Great Lakes Environmental Indicators (GLEI) project, U.S. EPA Agreement EPA/R-8286750. The funders had no role in study design, data collection and analysis, decision to publish, or preparation of the manuscript.

### Grant Disclosures

The following grant information was disclosed by the authors:
U.S. Environmental Protection Agency's Science to Achieve Results (STAR) Estuarine and Great Lakes (EaGLe) program.

Great Lakes Environmental Indicators (GLEI) Project.
U.S. EPA Agreement: EPA/R-8286750.

## Competing Interests

The authors declare that they have no competing interests.

## Author Contributions

- Euan D. Reavie conceived and designed the experiments, performed the experiments, analyzed the data, prepared figures and/or tables, authored or reviewed drafts of the article, and approved the final draft.

## Data Availability

The raw data is available in the Supplemental Files.

## Supplemental Information

Supplemental information for this article can be found online at http://dx.doi.org/10.7717/peerj.14887#supplemental-information.

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
